

# Learning the simplicity of scattering amplitudes

Clifford Cheung[1⋆], Aurélien Dersy[2,3†] and Matthew D. Schwartz[2,3‡]

**1** Walter Burke Institute for Theoretical Physics,
California Institute of Technology, 91125 Pasadena, CA, USA
**2** Department of Physics, Harvard University, 02138 Cambridge, MA, USA
**3** NSF Institute for Artificial Intelligence and Fundamental Interactions

⋆ clifford.cheung@caltech.edu , † adersy@g.harvard.edu , ‡ schwartz@g.harvard.edu

## Abstract

The simplification and reorganization of complex expressions lies at the core of scientific progress, particularly in theoretical high-energy physics. This work explores the application of machine learning to a particular facet of this challenge: the task of simplifying scattering amplitudes expressed in terms of spinor-helicity variables. We demonstrate that an encoder-decoder transformer architecture achieves impressive simplification capabilities for expressions composed of handfuls of terms. Lengthier expressions are implemented in an additional embedding network, trained using contrastive learning, which isolates subexpressions that are more likely to simplify. The resulting framework is capable of reducing expressions with hundreds of terms—a regular occurrence in quantum field theory calculations—to vastly simpler equivalent expressions. Starting from lengthy input expressions, our networks can generate the Parke-Taylor formula for five-point gluon scattering, as well as new compact expressions for five-point amplitudes involving scalars and gravitons. An interactive demonstration can be found at https://spinorhelicity.streamlit.app.



# 1  Introduction

The modern scattering amplitude program involves both the computation of amplitudes as well as the study of their physical properties. Are there better, more efficient, or more transparent ways to compute these objects? The dual efforts to devise powerful techniques for practical calculation and to then use those results to glean new theoretical structures have led to sustained progress over the last few decades. An archetype of this approach appears in the context of QCD, whose Feynman diagrams yield famously cumbersome and lengthy expressions. For example, even for the relatively simple process of tree-level, five-point gluon scattering, Feynman diagrams produce hundreds of terms. However, in the much-celebrated work of Parke and Taylor [1], it was realized that this apparent complexity is illusory. These hundreds of terms at five-point—and more generally, for any maximally helicity violating configuration—simplify to a shockingly compact monomial formula,

$$A(1^+2^+3^+\cdots i^-\cdots j^-\cdots n^+) = \frac{\langle ij\rangle^4}{\langle 12\rangle\langle 23\rangle\cdots\langle n1\rangle}, \tag{1}$$

shown here in its color-ordered form. The simplicity of the Parke-Taylor formula strongly suggests an alternative theoretical framework that directly generates expressions like Eq. (1) without the unnecessarily complicated intermediate steps of Feynman diagrams.

   This essential fact—that on-shell scattering amplitudes are simple and can illuminate hidden structures in theories—has led to new physical insights. Indeed, shortly after [1] it was realized that Eq. (1) also describes the correlators of a two-dimensional conformal field theory [2], which is a pillar of the modern-day celestial holography program [3]. Much later, Witten deduced from Eq. (1) that Yang-Mills theory is equivalent to a certain topological string

theory in twistor space [4], laying the groundwork for a vigorous research program that eventually led to the twistor Grassmanian formulation [5,6] and amplituhedron [7]. Examples like this abound in the amplitudes program—structures like double copy [8,9] and the scattering equations [10–12] were all derived from staring directly at amplitudes, rather than from the top-down principles of quantum field theory.

Progress here has hinged on the existence of *simple* expressions for on-shell scattering amplitudes. We are thus motivated to ask whether there is a more systematic way to recast a given expression from its raw form into its most compact representation. For example, a complicated spinor-helicity expression can often be simplified through repeated application of Schouten identities

$$|1\rangle\langle 23\rangle + |2\rangle\langle 31\rangle + |3\rangle\langle 12\rangle = 0\,, \tag{2}$$

together with total momentum conservation of $n$-point scattering

$$|1\rangle[1| + |2\rangle[2| + \cdots + |n\rangle[n| = 0\,. \tag{3}$$

However, the search space for these operations is expansive and difficult to navigate even with the help of existing computer packages [13, 14], and, to our knowledge, there exists no canonical algorithmic way to inform which operations simplify complicated expressions analytically. This is where recent advances in machine learning (ML) offer a natural advantage.

The role of ML in high-energy physics has grown dramatically in recent years [15]. In the field of scattering amplitudes, much of the work to date has focused on reproducing the numerical output of these amplitudes using neural networks [16–19]. However, recent advances in ML have led to the development of powerful architectures, capable of handling increasingly complex datasets, including those that are purely symbolic. In particular, the transformer architecture [20] has allowed for practical applications across a wide range of topics, including jet tagging [21], density estimation for simulation [22,23], and anomaly detection [24]. The appeal of transformers comes from their ability to create embeddings for long sequences which take into account all of the objects composing that sequence. In natural language processing, where transformers first originated, this approach encodes a sentence by mixing the embeddings of all of the words in the sentence. These powerful representations have been a key driver for progress in automatic summarization, translation tasks, and natural language generation [25–27]. Since mathematical expressions can also be understood as a form of language, the transformer architecture has been successfully repurposed to solve certain interesting mathematical problems. For those problems, the validity of a model's output can often be confirmed through explicit numerical evaluation of the symbolic result, allowing one to easily discard any model hallucinations. From symbolic regression [28] to function integration [29], theorem proving [30], and the amplitudes bootstrap [31], transformers have proven to be effective in answering questions that are intrinsically analytical rather than numerical. In particular, transformers have been adapted to simplify short polylogarithmic expressions [32] and it is natural to expect that the same methodology can be extended to our present task, which is the simplification of spinor-helicity expressions.

A common bottleneck for transformer-based approaches is the length of the mathematical expression that can be fed through the network. Typical amplitude expressions can easily have thousands of distinct terms and processing the whole expression at once quickly becomes intractable. The self-attention operation in a transformer scales quadratically in time and memory with the sequence length and it is therefore most efficiently applied to shorter expressions. For instance, the Longformer and BigBird architectures [33,34] implement reduced self-attention patterns, using a sliding window view on the input sequence and resorting to global attention only for a few select tokens. In the context of simplifying mathematical expressions, it is quite clear that humans proceed similarly: we start by identifying a handful of terms that are likely to combine and then we attempt simplification on this subset. In this

**SciPost** SciPost Phys. 18, 040 (2025)

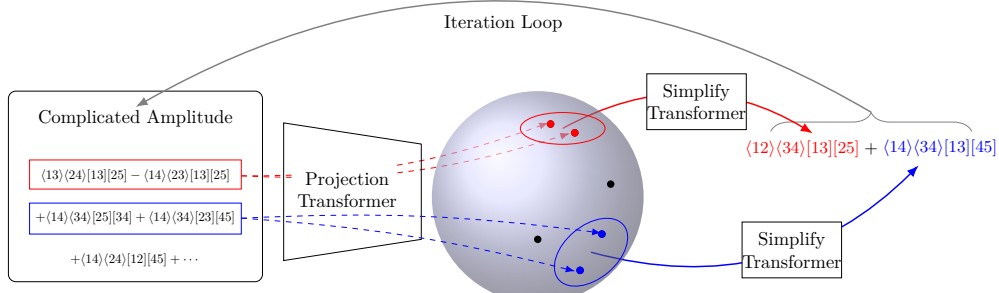

Figure 1: Spinor-helicity expressions are simplified in several steps. To start, individual terms are projected into an embedding space (grey sphere). Using contrastive learning, we train a "projection" transformer encoder to learn a mapping that groups similar terms close to one another in the embedding space. After identifying similar terms we use a "simplify" transformer encoder-decoder to predict the corresponding simple form. After simplifying all distinct groups, this procedure is repeated with the resulting expression, iterating until no further simplification is possible.

paper, we mimic this procedure by leveraging contrastive learning [35–39]. As illustrated in Fig. 1 we train a network to learn a representation for spinor-helicity expressions in which terms that are likely to simplify are close together in the learned embedding space. Grouping nearby terms, we then form a subset of the original expression which is input into yet another transformer network trained to simplify more moderately-sized expressions. By repeating the steps of grouping and simplification we are then able to reduce spinor-helicity expressions with enormous numbers of distinct terms.

Our paper is organized as follows. We begin in Section 2 with a brief review of the spinor-helicity formalism and its role in scattering amplitude calculations. We describe the physical constraints that amplitudes must satisfy, as well as the various mathematical identities that can relate equivalent expressions. In Section 3 we introduce a transformer encoder-decoder architecture adapted to the simplification of moderately-sized spinor-helicity expressions. We describe our procedure for generating training data and discuss the performance of our networks. Afterwards, in Section 4 we present the concept of contrastive learning and describe how it arrives at a representative embedding space. We present an algorithm for grouping subsets of terms that are likely to simplify in lengthier amplitude expressions. We then showcase the performance of our full simplification pipeline on actual physical amplitudes, in many cases composed of hundreds of terms.[1] Finally, we conclude with a brief perspective on the prospects for ML in this area.

## 2 Notation and training data

In this section, we review the mechanics of the spinor-helicity formalism and then describe the generation of training data for our models. Our notation follows [40], though a more detailed exposition can also be found in [41–43] and references within.

---

[1]Our implementation, datasets and trained models are available at https://github.com/aureliendersy/spinorhelicity. This repository also contains a faster local download of our online interactive demonstration, hosted at https://spinorhelicity.streamlit.app. This application reduces amplitudes following the procedure described in Fig. (1) and has the ability to simplify the amplitude expressions quoted in this paper.

## 2.1 Spinor-helicity formalism

The basic building blocks of spinor-helicity expressions are *helicity spinors,* which are two-component objects whose elements are complex numbers. Left-handed spinors transform in the $\left(\frac{1}{2}, 0\right)$ representation of the Lorentz group and are written as $\lambda_\alpha$. Right-handed spinors transform in the $\left(0, \frac{1}{2}\right)$ representation of the Lorentz group and are written as $\tilde{\lambda}^{\dot{\alpha}}$. A general four-momentum transforms in the $\left(\frac{1}{2}, \frac{1}{2}\right)$ representation of the Lorentz group and is written as the two-by-two matrices $p^{\alpha\dot{\alpha}}$ or $p_{\dot{\alpha}\alpha}$. When the four-momentum corresponds to a massless particle, it satisfies the on-shell condition, $p \cdot p = \det(p^{\alpha\dot{\alpha}}) = 0$, and can be written as the outer product of helicity spinors, so $p^{\alpha\dot{\alpha}} = \lambda^\alpha \tilde{\lambda}^{\dot{\alpha}}$. As usual in the study of scattering amplitudes, we generalize to complex four-momenta, so $\lambda^\alpha$ and $\tilde{\lambda}^{\dot{\alpha}}$ are independent objects.

Helicity spinors of the same chirality can be dotted into each other to form the Lorentz invariant, antisymmetric products,

$$\text{Angle brackets:} \quad \langle \lambda \chi \rangle = -\langle \chi \lambda \rangle = \lambda_\alpha \chi_\beta \, \varepsilon^{\alpha\beta} \,, \tag{4}$$

$$\text{Square brackets:} \quad [\lambda \chi] = -[\chi \lambda] = \tilde{\lambda}^{\dot{\alpha}} \tilde{\chi}^{\dot{\beta}} \, \varepsilon_{\dot{\alpha}\dot{\beta}} \,.$$

Here all indices are raised and lowered with the antisymmetric two-index tensors, $\varepsilon^{\alpha\beta}$ or $\varepsilon_{\dot{\alpha}\dot{\beta}}$. The Lorentz invariant product of a pair of four-momenta is

$$p \cdot q = \frac{1}{2} \langle \lambda \chi \rangle [\chi \lambda], \tag{5}$$

where we have also defined $q^{\alpha\dot{\alpha}} = \chi^\alpha \tilde{\chi}^{\dot{\alpha}}$.

For physical processes, we are typically interested in the $n$-point amplitude, which describes a scattering process involving $n$ external particles, here taken to be all incoming for convenience. This object depends on $n$ external massless momenta, which we write as $p_i^{\alpha\dot{\alpha}} = \lambda_i^\alpha \tilde{\lambda}_i^{\dot{\alpha}}$. We use the standard shorthand in which angle and square brackets are labelled by their corresponding external states, so

$$p_i \cdot p_j = \frac{1}{2} \langle ij \rangle [ji]. \tag{6}$$

Note that the antisymmetry of the angle and square bracket imply that $\langle ii \rangle = [ii] = 0$.

The $n$-point scattering amplitude is strongly constrained by the little group, which by definition acts trivially on four-momenta but nontrivially on helicity spinors

$$\lambda_i^\alpha \to z_i \lambda_i^\alpha, \quad \text{and} \quad \tilde{\lambda}_i^{\dot{\alpha}} \to z_i^{-1} \tilde{\lambda}_i^{\dot{\alpha}}, \tag{7}$$

where $z_i$ is an arbitrary complex number. The little group defines the spin representation of each external state. Consequently, the $n$-point scattering amplitude transforms under the little group as

$$\overline{\mathcal{M}}(1^{h_1} 2^{h_2} \cdots n^{h_n}) \to \left( \prod_i z_i^{-2h_i} \right) \overline{\mathcal{M}}(1^{h_1} 2^{h_2} \cdots n^{h_n}), \tag{8}$$

where $h_i$ is the helicity of leg $i$. Hence, the little group strongly constrains the number of powers of each helicity spinor that can appear in every term in the amplitude. Note also that the mass dimension of each helicity spinor is one-half, so each angle or square bracket is mass dimension one.

A general $n$-point amplitude is highly constrained by the little group and dimensional analysis. As we have seen, Eq. (8) restricts the allowed powers of left- and right-handed spinors. Assuming that there is a single coupling constant of fixed mass dimension, then the mass dimension of each term in the amplitude must also be the same. These constraints, together with information from singular kinematic limits, can sometimes be exploited to extract analytic results from numerical calculations [44–47]. Within our ML framework, we will assume that all amplitudes have fixed mass dimension and little group scaling.

## 2.2 Target data set

The goal of this work is to train a computer program that takes as input a complicated spinor-helicity expression $\mathcal{M}$ and then outputs a more minimal form $\overline{\mathcal{M}}$. The ML approach to this problem requires a training set composed of multiple instances of such pairs, $\{\mathcal{M}, \overline{\mathcal{M}}\}$. To build such a set, we randomly generate a simple target expression $\overline{\mathcal{M}}$ and then scramble it using various spinor-helicity identities to obtain a more complicated but mathematically equivalent form $\mathcal{M}$. We then iterate this procedure many times to generate a list of many such pairs $\{\mathcal{M}, \overline{\mathcal{M}}\}$. Following the terminology of [29], this data generation procedure is called *backward generation*, where one starts by generating the target $\overline{\mathcal{M}}$, rather than the input $\mathcal{M}$. In the alternative approach, the *forward generation*, one would instead generate $\mathcal{M}$ and simplify it with an external software to generate the target $\overline{\mathcal{M}}$. Since we lack a clear algorithmic way to maximally simplify an amplitude, we cannot use this approach, and our datasets will be constructed only from backward generation.

As noted earlier, for $\overline{\mathcal{M}}$ to describe a physical scattering amplitude, its terms must all exhibit the same little group scaling and mass dimension. However, to craft a general simplification algorithm, our network will need to be able to simplify subexpressions whose little group scaling and mass dimension differ from the final target. For this reason, the various pairs $\{\mathcal{M}, \overline{\mathcal{M}}\}$ in the same training set will in general exhibit different mass dimensions or external helicity choices.

An efficient mathematical representation of spinor-helicity expressions should be free of notational degeneracies. To eliminate the intrinsic redundancy of antisymmetry in the angle and square brackets, we choose a convention in which all brackets are written with their first entry smaller than the second. Concretely, we send $\langle ji \rangle \to -\langle ij \rangle$ and $[ji] \to -[ij]$ for $i < j$ whenever possible. Furthermore, we rationalize all of our amplitudes and write the numerator in a fully expanded form, yielding

$$\overline{\mathcal{M}} = \frac{1}{\mathcal{D}} \sum_{\ell=1}^{N_{\text{terms}}} \mathcal{N}_\ell \,, \tag{9}$$

where each $\mathcal{N}_\ell$ is a monomial product of angle and square brackets and $\mathcal{D}$ is a common denominator. Since the target amplitudes should be compact, the number of distinct terms $N_{\text{terms}}$ should not be too large. For concreteness, we restrict $0 \le N_{\text{terms}} \le 3$, with $\overline{\mathcal{M}} = 0$ an allowed possibility, so this is our operational definition of "simple".

The precise algorithm for creating a target amplitude $\overline{\mathcal{M}}$ for the training set is as follows. To begin, we randomly generate its first term, i.e., $\mathcal{N}_1/\mathcal{D}$ corresponding to $\ell = 1$, in several steps:

1. We fix the number $n$ of external momenta.

2. We fix the number of numerator $n_{\mathcal{N}}$ and denominator terms $n_{\mathcal{D}}$, which are chosen in the ranges $n_{\mathcal{N}} \in [0, 2n]$ and $n_{\mathcal{D}} \in [1, 2n]$.

3. For each numerator or denominator term $r$ we

   (a) Randomly choose $[ij]$ or $\langle ij \rangle$, where $i < j$ and $i, j \in [1, n]$.

   (b) Raise this bracket to the power $p = \max(1, \tilde{p})$, where $\tilde{p}$ is drawn from the Poisson distribution $\text{Poisson}(\lambda)$, where $\lambda = 0.75$ in our analysis. The resulting expression for the term is then $r = \langle ij \rangle^p$ or $r = [ij]^p$.

4. We combine the brackets and randomize the overall sign to obtain the first term,

$$\frac{\mathcal{N}_1}{\mathcal{D}} = \pm \prod_{a=1}^{n_{\mathcal{N}}} \prod_{b=1}^{n_{\mathcal{D}}} \frac{r_a}{r_b} \,. \tag{10}$$

Next, we add additional terms to the target amplitude that have the same little group scaling and mass dimension as the first term. These additional terms are of the form

$$\mathcal{N}_{\ell>1} = \prod_{j=2}^{n}\prod_{i=1}^{j-1} \langle ij \rangle^{p_{ij}} [ij]^{\tilde{p}_{ij}} \,, \tag{11}$$

where $p_{ij}$ and $\tilde{p}_{ij}$ are the solutions to the system of $n+1$ equations,

$$m(\mathcal{N}_1) = \sum_{j=2}^{n}\sum_{i=1}^{j-1} \left( p_{ij} + \tilde{p}_{ij} \right) \,, \tag{12}$$

$$h_k(\mathcal{N}_1) = \sum_{j=k+1}^{n} \left( p_{kj} - \tilde{p}_{kj} \right) + \sum_{i=1}^{k-1} (p_{ik} - \tilde{p}_{ik}) \,, \tag{13}$$

where we have defined $\mathcal{N}_1$ to have mass dimension $m(\mathcal{N}_1)$ and little group scalings $h_k(\mathcal{N}_1)$ for each external momentum $1 \leq k \leq n$. Here a solution is deemed acceptable only if the coefficients $p_{ij}, \tilde{p}_{ij}$ are non-negative, so the common denominator is unchanged. We then repeat this procedure until we have generated all terms $\mathcal{N}_\ell$ in Eq. (9), thus yielding our final form for $\overline{\mathcal{M}}$.

Note that when adding numerator terms we do not multiply them by random rational numbers. Rather, we instead consider expressions where each numerator term has $\pm 1$ as a relative coefficient. This will be mostly sufficient for the physical amplitudes under consideration.

## 2.3 Input data set

With a set of target amplitudes $\overline{\mathcal{M}}$ in hand, we can now scramble them into more complicated input amplitudes $\mathcal{M}$ so that the inverse map can be learned by the network. This reshuffling is achieved using various mathematical identities that relate equivalent spinor-helicity expressions.

The first mathematical identity that we will use for scrambling is the Schouten identity, which is a consequence of the two-dimensional nature of spinors:

$$\text{Schouten identity}: \quad \begin{cases} \langle ij \rangle \langle kl \rangle = \langle il \rangle \langle kj \rangle + \langle ik \rangle \langle jl \rangle \,, \\ [ij][kl] = [il][kj] + [ik][jl] \,. \end{cases} \tag{14}$$

These relations obviously generate more terms from fewer terms, and they are independent of the number of external legs $n$.

The second identity that we use arises as a consequence of the total momentum conservation: $\sum_{i=1}^{n} p_i^{\alpha\dot\alpha} = \sum_{i=1}^{n} \lambda_i^\alpha \tilde{\lambda}_i^{\dot\alpha} = 0$. Sandwiching this identity between any of the $n$ helicity spinors yields the $n^2$ equations

$$\text{momentum conservation}: \quad \sum_{j=1}^{n} \langle ij \rangle [jk] = 0 \,, \quad \forall i, k \,. \tag{15}$$

When $i \neq k$, this is a linear relation on $n-2$ non-vanishing terms, whereas for $i = k$ it constrains $n-1$ non-vanishing terms. Here we can also take the square of total momentum conservation, $(\sum_{i=1}^{n} p_i^{\alpha\dot\alpha})^2 = 0$, to obtain

$$\text{momentum squared}: \quad \sum_{\substack{i<j \\ (i,j)\in S_1^n}} \langle ij \rangle [ji] = \sum_{\substack{k<l \\ (k,l)\in S_2^n}} \langle kl \rangle [lk] \,, \tag{16}$$

where $S_1^n$ and $S_2^n$ are two disjoint subsets forming a partition of the momenta set $\{p_1, \cdots, p_n\}$. Of course, total momentum conservation and its square are not independent identities. In fact, one can often simplify amplitudes in different ways using different identities. For instance, to simplify the expression $\langle 14 \rangle [14] - \langle 23 \rangle [23]$ in four-point scattering, one can use the squared version of momentum conservation which reads $(p_1 + p_4)^2 = (p_2 + p_3)^2$ and implies $\langle 14 \rangle [14] = \langle 23 \rangle [23]$. Alternatively, one can use momentum conservation, $\langle 12 \rangle [23] = -\langle 14 \rangle [43]$, multiply both sides by $[14]$, and then apply another momentum conservation identity, $\langle 21 \rangle [14] = -\langle 23 \rangle [34]$. So while the various identities are not independent, having some redundancy in operations can often expedite multiple intermediate simplification steps.

To proceed, we allow for the scrambling identities of Eqs. (15-16) to be applied in two slightly different ways. The first method involves selecting a random bracket in the numerator of a spinor-helicity expression and then choosing whether to apply momentum conservation, its squared counterpart, or the Schouten identity. There is no technical reason that requires us to only scramble terms in the numerator, but we do so for the sake of simplicity. Note that the amplitudes we consider will have denominators that are simple products of square and angle brackets. Once a numerator bracket has been selected, we then randomly pick an identity and craft the appropriate replacement rule. For instance, if $\langle 12 \rangle$ is selected in the five-point amplitude, we can generate a substitution following the Schouten identity as

$$\langle 12 \rangle \to \frac{\langle 13 \rangle \langle 25 \rangle - \langle 15 \rangle \langle 23 \rangle}{\langle 35 \rangle} \, . \tag{17}$$

To apply Eq. (17) we must randomly choose two additional external momenta, as required by the form of Eq. (14). Similarly, when applying momentum conservation or its squared cousin, one must randomly select reference helicity spinors, as in Eq. (15), or the subsets $S_1^n$ and $S_2^n$, as in Eq. (16). After applying the substitution in Eq. (17) to all relevant bracket terms in the numerator, we then say that the amplitude is one identity away from its simple form, i.e., it has been *scrambled once*.

To increase the diversity of the generated expressions we implement a second method for applying the scrambling identities. Rather than substituting an existing bracket, we instead allow for multiplication by unity or addition of zero. To multiply by unity we write a trivial fraction using a randomly chosen bracket and scramble its corresponding numerator following the aforementioned procedure. For instance, if we are using the Schouten identity in this scrambling step we send

$$\text{multiplication by unity}: \quad \overline{\mathcal{M}} \to \overline{\mathcal{M}} \, \frac{\langle 12 \rangle}{\langle 12 \rangle} \to \overline{\mathcal{M}} \, \frac{\langle 13 \rangle \langle 25 \rangle - \langle 15 \rangle \langle 23 \rangle}{\langle 12 \rangle \langle 35 \rangle} \, . \tag{18}$$

For the addition of zero we proceed similarly: randomly select a bracket, write $0 = \langle ij \rangle - \langle ij \rangle$, and then scramble one of the two terms. This step is necessary when scrambling target amplitudes that vanish, so $\overline{\mathcal{M}} = 0$. Alternatively, we can also insert this identity into the numerator of a spinor-helicity expression, where we need to multiply it by an appropriate bracket expression so that the little group and mass dimension scalings are preserved. For instance, we can apply the replacement

$$\text{addition of zero}: \quad \overline{\mathcal{M}} + \frac{[34] - [34]}{\mathcal{D}} \mathcal{F} \to \overline{\mathcal{M}} + \frac{[14][35] - [13][45] - [15][34]}{\mathcal{D}[15]} \mathcal{F} \, , \tag{19}$$

where we have scrambled $[34]$ with the Schouten identity. Here $\mathcal{D}$ is the denominator of the original amplitude and $\mathcal{F}$ is a factor required to ensure that the scaling behaviour of $\overline{\mathcal{M}}$ is respected. This factor is sampled from the solution set obtained by solving the system[2] of

---

[2]We allow for negative power coefficients in our solution set, so generically $\mathcal{F}$ is not a simple monomial of square and angle brackets. Instead, it can also have brackets raised to a negative power.

Eqs. (12-13) using [48]. In our analysis, multiplication by unity and addition of zero will count as a single scrambling step since a single identity is sufficient to undo them.

One could be concerned that the backward generation procedure introduces some bias in the training samples. While the target $\overline{\mathcal{M}}$ are chosen to match with simple amplitudes expressions (or at least part of them), it is not immediately clear whether all possible $\mathcal{M}$ can be reached from this generation process. It will thus be important to test our models on amplitude expressions that one encounters in practical settings, as we do in Sec 4.4, to ensure that we have adequate generalization beyond the training set.

## 2.4 Analytic simplification

Before moving on, it is worthwhile to comment briefly on various analytic approaches to amplitudes simplification that have been developed over the years. Since the ambiguities in representing a given spinor-helicity expression stem from the Schouten identity and momentum conservation, it is natural to try to devise kinematic variables which trivialize these identities. For example, in projective coordinates, we have that $\lambda_i^\alpha = (1, z_i)$ and $\langle ij \rangle = z_i - z_j$, so the Schouten identity is algebraically satisfied. However, momentum conservation is not manifest, so in these variables, a given spinor helicity expression can still be expressed in many distinct ways.

Alternatively, one can trivially manifest momentum conservation using momentum twistor variables [49,50]. Here one defines twistors residing in dual spacetime coordinates for which $x_i^{\alpha\dot\alpha} - x_{i+1}^{\alpha\dot\alpha} = \lambda_i^\alpha \bar\lambda_i^{\dot\alpha}$. These variables are natural for planar amplitudes exhibiting dual conformal invariance, as in maximally supersymmetric Yang-Mills theory. However, such simplifications are certainly not generic. More importantly, since momentum twistors are four-component objects, five or more of them are linearly dependent due to the higher-dimensional analogue of the Schouten identity. Indeed, any finite-dimensional representation of the kinematics will exhibit this ambiguity. Thus, irrespective of the analytic approach taken, there is no general way to trivialize the simplification task.

# 3 One-shot learning

Armed with a training set of spinor-helicity expressions $\mathcal{M}$ and their simplified target counterparts $\overline{\mathcal{M}}$, we can now apply ML. Concretely, we are interested in reducing complicated input expressions like

$$\mathcal{M} = \frac{\left(-\langle 12 \rangle^2 [12][15] - \langle 13 \rangle \langle 24 \rangle [13][45] + \langle 13 \rangle \langle 24 \rangle [14][35] - \langle 13 \rangle \langle 24 \rangle [15][34]\right) \langle 12 \rangle}{\langle 15 \rangle \langle 23 \rangle \langle 34 \rangle \langle 45 \rangle [12][15]}, \quad (20)$$

down to simplified target expressions like

$$\overline{\mathcal{M}} = -\frac{\langle 12 \rangle^3}{\langle 15 \rangle \langle 23 \rangle \langle 34 \rangle \langle 45 \rangle}. \quad (21)$$

By hand, the simplification of spinor-helicity expressions proceeds by successive applications of well-chosen identities. For instance, the jump from Eq. (20) to Eq. (21) would be achieved using a single Schouten identity, $[13][45] = [15][43] + [14][35]$. We would instead like to create a ML algorithm that performs this simplification automatically.

The task of simplifying expressions is similar to theorem proving, where a program learns to apply a set of axioms or tactics to reach a desired goal. In this context, reinforcement learning [51,52] and Monte Carlo tree search [30] have already successfully reconstructed lengthy proofs. However, these approaches become increasingly difficult to implement for larger expressions and more mathematical identities. In this paper, rather than trying to train models

Table 1: Transformer architecture and training hyperparameters used for the one-shot simplification of spinor-helicity amplitudes.

| Hyperparameter Type | Parameter Description | Value |
|---|---|---|
| Network architecture | Encoder layers | 3 |
| | Decoder layers | 3 |
| | Attention heads | 8 |
| | Embedding dimension | 512 |
| | Maximum input length | 2560 |
| Training parameters | Batch size | 16 |
| | Epoch size | 50000 |
| | Epoch number | 1500 |
| | Learning Rate | $10^{-4}$ |

to learn a sequence of simplification steps, we instead focus on models that can generate a list of guesses for what the simple form can be without explicitly listing the intermediate steps in between. In this way, going from $\mathcal{M} \to \overline{\mathcal{M}}$ can be viewed as a one-shot translation task for which transformer networks have demonstrated excellent performance on analogous tasks.

From their conception, transformer models have excelled at translation tasks [20]. More recently, these architectures have been deployed to integrate functions [29] and simplify poly-logarithmic expressions [32], which are mathematical problems that share common features with language translation. In particular, one exploits the fact that any mathematical expression possesses a tree-like structure. Using prefix notation, where operators precede operands, this tree-like structure is represented as an ordered set of tokens, akin to a regular sentence, yielding an input that can be passed through a transformer. See Appendix A for a detailed description of this structure in a concrete example.

As an initial experiment, we restrict our training data to expressions composed of at most 1k tokens, guaranteeing a reasonable memory requirement and training time. The structure of the training data is discussed in detail in Appendix B. Since scrambling identities will swiftly increase the size of an expression, we initially restrict to $\mathcal{M}$ that are related to $\overline{\mathcal{M}}$ by at most three scrambling steps. With more scrambling steps, the typical size of an amplitude can exceed thousands of tokens and one-shot simplification is no longer suitable. We discuss the simplification of more complicated expressions such as this in the subsequent section.

## 3.1 Network architecture

In this paper we closely follow the implementation of [29] and employ an encoder-decoder transformer architecture[3] defined with the hyperparameters detailed in Tab. 1. As detailed in Appendix A, an input expression like $\mathcal{M} = \langle 12 \rangle [34]$ is first converted to a prefix notation $\mathcal{P}(\mathcal{M}) = [\text{'mul'}, \text{'ab12'}, \text{'sb34'}]$ where the tokens are either binary operators, integers, or angle and square brackets. This set of ordered tokens is fed through an embedding layer with positional encoding before passing through a set of self-attention layers. These layers ensure that the encoding of each token is conditioned on the embedding of all of the other tokens making up the input sentence. The resulting embedded sentence is then passed to the decoder, which is composed of self-attention layers and a final projection layer that together with a softmax function is responsible for assigning a probability distribution over the set of allowed tokens.

---

[3]Our implementation and analysis use the *PyTorch* [53], *Sympy* [54], *Scikit-learn* [55], *Numpy* [56] and *Matplotlib* [57] libraries.

Following the procedure outlined in Sec 2.2, we generate training data for spinor-helicity amplitudes with four, five, or six external momenta. In each case, we train a separate transformer network on a single A100 GPU using the parameters summarized in Tab 1 and the *Adam* optimizer [58]. To reach 1500 epochs we have training times of 45h, 65h and 80h for four, five and six-point amplitudes respectively. Each transformer is honed on training data for which three or fewer scrambling steps have been applied to produce the input amplitude, corresponding to a total of 10M unique amplitude pairs. We additionally retain 10k examples for testing our trained models, where the associated input amplitudes have not been encountered previously during training. To characterize the performance of our networks, we do not rely on the naive in-training measure of accuracy. Indeed, since our models are trained using a cross-entropy loss on the predicted tokens, the measure of accuracy that one has access to during training is based on whether the model can exactly reproduce the ordered set of tokens that corresponds to the target amplitude. However, in some cases, the same target amplitude can be written in different equivalent ways. For example, this can easily occur in four-point amplitudes, where the target expression defined in Eq. (9) can often be further simplified or expressed more compactly.[4] With the help of the tools developed in [13], we instead verify whether the numerical evaluation of the input amplitude matches that of the predicted output. We ask for a numerical equivalence at 9 digits of precision using two independent sets of phase space points, which proves to be sufficient for the amplitudes considered during training.

At inference time we implement a *beam search* [60] that automatically generates a multiplicity of distinct predictions for the simple form of the original spinor-helicity expression. Rather than restricting ourselves to *greedy decoding*, which simply outputs the tokens with the highest probability, this accommodates candidate amplitudes with lower probability tokens. For a beam search of size $N$, we retain the candidate amplitudes with the $N$ lowest scores, where the latter are calculated by summing the log-likelihood probability of each token and normalizing by the sequence length. Since the numerical evaluation of the candidate amplitudes provides us with an unambiguous criterion for identifying valid solutions, we can use large beam sizes at inference time in the hope that at least one candidate proves to be correct.

To boost the performance at inference time we also consider an alternative to beam search known as *nucleus sampling* [61]. Nucleus sampling is a stochastic decoding technique whereby the model output is constructed by sampling subsequent tokens according to their probability distributions, whereas, in contrast, beam search selects subsequent tokens based on their highest probabilities. To avoid sampling over irrelevant tokens, nucleus sampling only considers the most promising tokens by selecting the minimal set whose combined probability distribution exceeds a threshold $p_n$. This guarantees that only promising expressions are sampled and that tokens with low scores are generically ignored. Our implementation is further detailed in Appendix C.

## 3.2 Results

To characterize the performance of our trained models we compare evaluation results for a beam search of size 1, 5, 10, 20, and 50 in Fig. 2. In each case, we compare the complexity of the target amplitudes $\overline{\mathcal{M}}$ to the complexity of the best hypothesis in the beam. As described in Appendix B, we define the complexity to be the number of distinct square and angle brackets that compose the amplitude, which is a proxy for the compactness of the resulting expression. For beams of size greater than five, our models perform well, recovering a valid simplified form

---

[4]When manipulating four-point amplitudes, the momentum conservation identities typically only involve two distinct groups of terms. This implies that factorization is not unique and that the least common denominator in a four-point amplitude is not uniquely fixed [59]. Therefore, one generically has many ways of rewriting the same amplitude without increasing its complexity. For five- and six-point amplitudes this is much less likely, as factorization is conjectured to be unique.

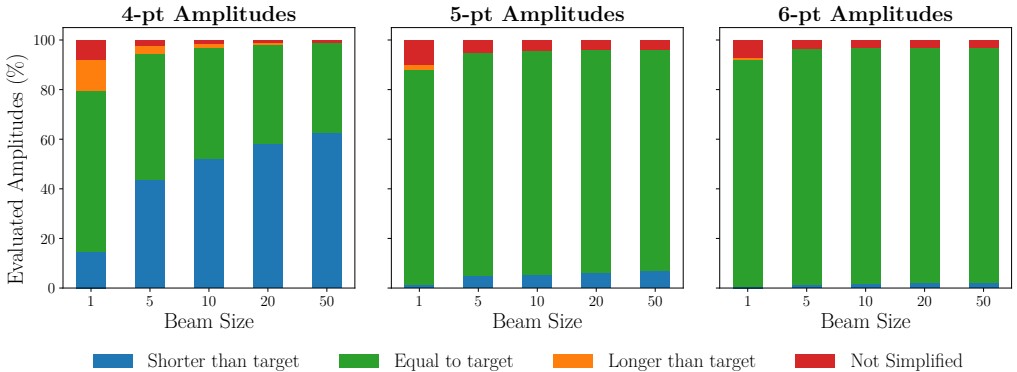

Figure 2: Performance on the held-out test set for the one-shot simplification of complicated $n$-point spinor-helicity amplitudes of up to 1k tokens. We compare the length of the model prediction against the length of the target. Green shows when the network reduces an expression to the correct target length while blue shows when the network simplifies beyond the target, which is possible for four-point amplitudes since they are highly redundant. The results are reported for different beam sizes used at inference, where only the shortest hypothesis is retained.

Table 2: Overall accuracy on the held-out test set for the one-shot simplification of complicated $n$-point spinor-helicity amplitudes. A model-generated amplitude is deemed accurate if it is both numerically equivalent and simpler than the input amplitude. Different inference techniques are compared and we indicate in each case the total number of generated candidate amplitudes.

| Technique | Beam Search | | | Nucleus Sampling | | | Beam + Nucleus | | |
|---|---|---|---|---|---|---|---|---|---|
| # Candidates | 1 | 10 | 20 | 1 | 10 | 20 | 10 | 20 | 100 |
| 4-pt amplitudes | 92.0% | 98.6% | 98.8% | 88.8% | 98.3% | 99.0% | 98.7% | 99.2% | 99.6% |
| 5-pt amplitudes | 90.0% | 95.9% | 96.1% | 90.4% | 95.9% | 96.8% | 96.4% | 97.4% | 98.7% |
| 6-pt amplitudes | 93.1% | 96.9% | 97.0% | 93.9% | 97.2% | 98.0% | 97.6% | 98.4% | 99.1% |

of the input expression in over 95% of cases, irrespective of the number of external momenta. Remarkably, for four-point amplitudes, the models recover shorter versions of the original target expression and the different hypotheses generated in the beam correspond to distinct valid amplitudes. In fact, for a beam of size 50, we find that 63% of the target four-point amplitudes admit a more compact rewriting that is recovered by our model. For five-point and six-point amplitudes, this occurs in only 7% and 2.5% of cases respectively, indicating that the target amplitudes are typically generated in their most compact form.

Tab. 2 contrasts the performance of our networks that use nucleus sampling versus beam search at inference time. We deem an amplitude accurate if it corresponds to a valid simplified form of the input amplitude, even if it is not as short as the intended target.[5] Both methods display comparable accuracy, but nucleus sampling is better when generating many candidate amplitudes. Here we can see that it is also beneficial to combine both inference methods to create a wider variety of model outputs. When generating ten total candidates, split evenly between each technique, we already reach or exceed the performance that is obtained by generating ten candidate amplitudes from a single inference method.

---

[5]This corresponds to combining all performance categories in Fig. 2 barring from "Not Simplified".

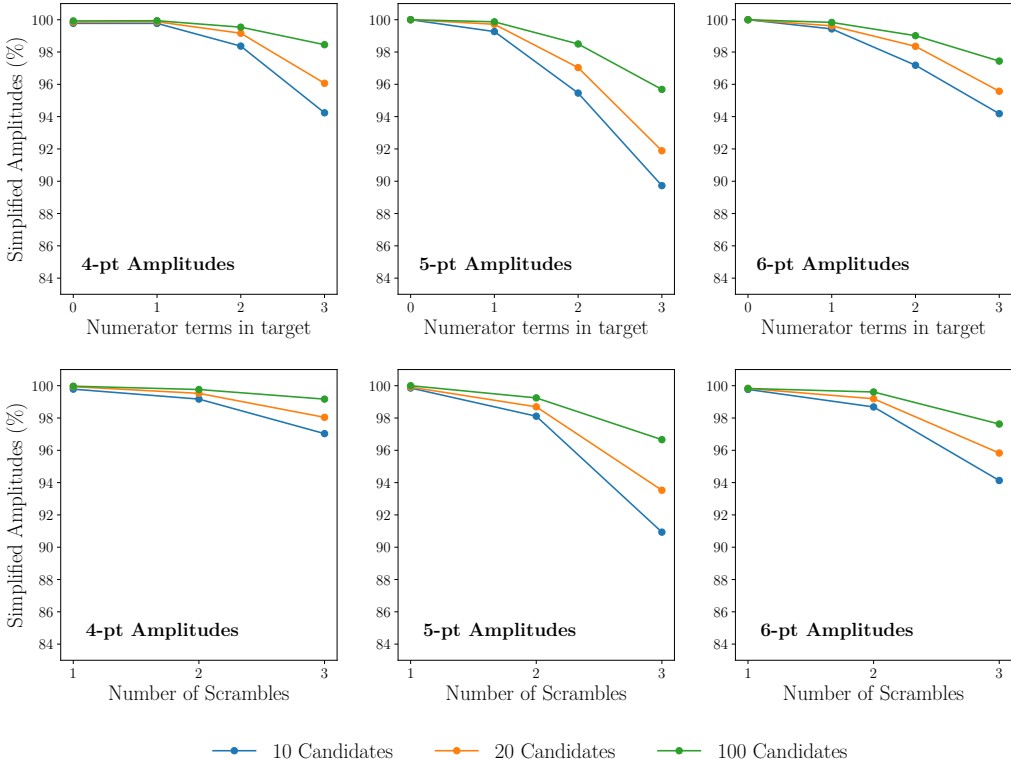

Figure 3: Accuracy on the held-out test set for the one-shot simplification of complicated $n$-point spinor-helicity amplitudes. A model-generated amplitude is deemed accurate if it is both numerically equivalent and simpler than the input amplitude. We compare the accuracy based on the number of distinct terms in the numerator of the target amplitude (top row) and based on the number of identities used to scramble it (bottom row).

Fig. 3 evaluates the performance of our models based on the number of identities used to scramble the target $\overline{\mathcal{M}}$ and the number of numerator terms $N_{\text{terms}}$ in $\overline{\mathcal{M}}$. Here a clear trend emerges in which all models perform slightly worse as the complexity of the target or the number of scrambling steps increases. This is of course expected, since the space of possible input amplitudes $\mathcal{M}$ grows swiftly with the number of identities applied. During training, however, our transformer models are exposed to amplitudes generated using one to three scrambles in equal proportion. Therefore, they have seen a larger fraction of the amplitude space $\mathcal{M}$ obtained by applying a single identity and are more successful in recovering the associated simple representations. Nevertheless, the relative drop in performance is minor, with models still recovering a simplified form with over 95% accuracy, even when three scrambling steps are used. To ascertain whether this performance generalizes well, we test our trained model for five-point amplitudes on data that is generated with up to five scrambling steps. The result is the blue curve in Fig. 4, which showcases a lack of proper generalization. Even with 100 candidate answers, we only recover a correct amplitude with a 60% success rate for five scrambling steps. This drop in performance is mitigated when training a model directly on amplitudes that are up to five scrambles away, as seen from the orange curve in Fig. 4, but, as we further explore in Appendix D, recovering a performance at the 99% level is unattainable. This implies that our models require additional training data to tackle longer amplitudes successfully and cannot generalize directly to longer sequence lengths. Nevertheless, for input amplitudes that are a small number of scrambling steps away from the target, the simplification

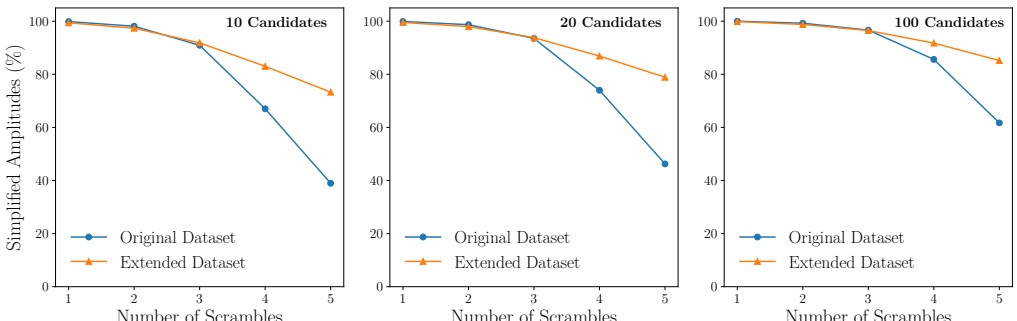

Figure 4: Accuracy on the held-out test set for the one-shot simplification of complicated five-point spinor-helicity amplitudes. We compare models that have seen amplitudes scrambled up to three times at most during training (blue) to models trained on up to five scrambles (orange).

task is essentially solved. Indeed, even when generating as few as ten candidate expressions, close to 99% of those input amplitudes are appropriately simplified by our models.

## 3.3 Embedding analysis

Our trained transformer models are successful at performing a one-shot simplification for input amplitudes of reasonable complexity, which are up to three scrambling steps away from their minimal form. To gain additional insight into the inner workings of these models we study their learned embeddings. This can be done, for instance, at the individual token level, and in Appendix E we analyze how the structure of the integers is learned. However, we can also probe how entire amplitudes are represented in our transformer models. To study how an input amplitude $\mathcal{M}$ is embedded, we pass it through the encoder layer of our transformer network. This initial amplitude is parsed in prefix notation as a sequence of words $\mathcal{P}(\mathcal{M})$ and the effect of the encoder layer is to project each word $w$ into the 512-dimensional embedding space. The resulting embedding vector $e(w)$ is conditioned on all other words, so that $e(w) = E(w|\mathcal{P}(\mathcal{M}))$ where $E$ is the encoder layer function. We then define our embedding for the full amplitude $e(\mathcal{M})$ to be the average over all of its word embeddings

$$e(\mathcal{M}) = \frac{1}{|\mathcal{P}(\mathcal{M})|} \sum_{w \in \mathcal{P}(\mathcal{M})} E(w|\mathcal{P}(\mathcal{M})), \qquad (22)$$

with $|\mathcal{P}(\mathcal{M})|$ being the cardinality of the set of words.

To analyze these amplitude embeddings we first normalize them, projecting the vectors on the unit hypersphere in $\mathbb{R}^{512}$. Then, we employ a two-dimensional t-SNE visualization [62] where the distances between the input vectors are computed using a cosine metric as in Eq. (E.1). In Fig. 5 the resulting visualization is color-coded based on the number of distinct terms in the numerator of the input amplitude, whereas in Fig. 6 the color scheme follows the mass dimension of the input amplitude. We can observe some structure in Fig. 5, where amplitudes are grouped based on the number of input terms.[6] This is particularly apparent for the four-point amplitudes, where the input amplitudes with only one or two terms in the numerator are well separated. Within those groups, there appears to be an additional ordering based on the mass dimension of the amplitude, as can be deduced from Fig. 6. For higher point amplitudes the ordering is not as evident. In particular, for the six-point visualization, we have

---

[6]A similar analysis of the embeddings derived solely from the token embedding layer and not the full encoder layer does not yield such a clear separation. This indicates that the attention mechanism is crucial in deriving this non-trivial structure.

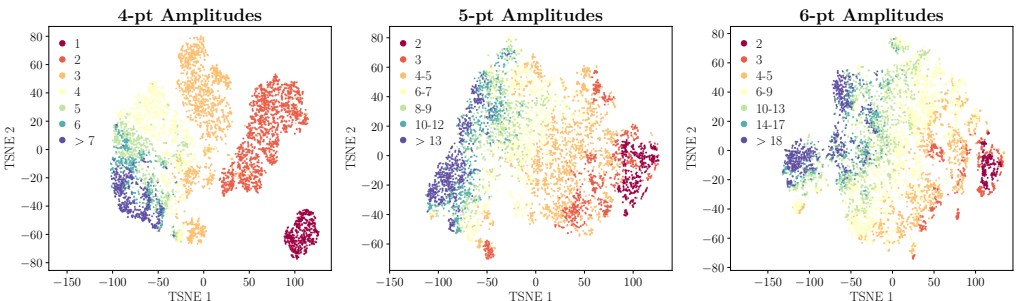

Figure 5: t-SNE visualization of 5k input amplitude embeddings. Each amplitude embedding is obtained using the transformer encoder, averaging over all constituent word embeddings. The points are color-coded according to the number of distinct terms in the numerator of the corresponding input amplitude.

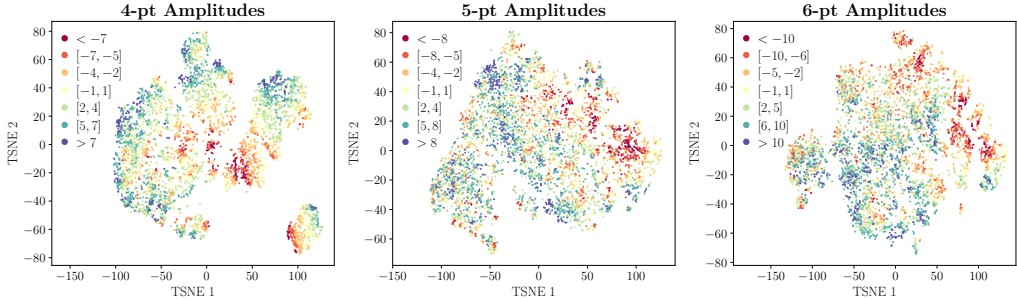

Figure 6: t-SNE visualization of 5k input amplitude embeddings, following the procedure detailed in the caption of Fig. (5). Each point is color-coded according to the mass dimension of the corresponding input amplitude.

a cluster of amplitudes which does not seem to be described by our color scheme (in the left of the rightmost panel of Fig. 5). Upon further investigation, we found that all of the input amplitudes there have at least one numerator coefficient which is ±2, instead of ±1 as for most other amplitudes. This suggests that the internal representation of the transformer encoder layer is increasingly complex for higher-point amplitudes.

## 4 Sequential simplification

The transformer models trained in Section 3 perform quite well on expressions of reasonable complexity, where the simplified form is three or fewer scrambling steps away.[7] However, as expected, performance drops precipitiously with increasing complexity of the input (more scrambles) and increasing complexity of the target (more irreducible terms). Rather than trying to brute-force the problem by training larger networks on bigger datasets, we instead consider an alternative route, performing a *sequential simplification* of spinor-helicity amplitudes. Akin to how we as humans simplify long mathematical expressions, we train a model to carefully select subsets of terms from the full amplitude and iteratively simplify those.

---

[7]As described in Appendix B, this corresponds to input amplitudes that have up to 10, 20 and 30 numerator terms for four, five and six-point amplitudes respectively.

## 4.1 Contrastive learning

In order to efficiently identify components of the amplitude that are likely to simplify, we train transformer networks to discern when individual numerator terms are *similar*. More precisely, we deem two distinct terms to be similar if they enter the same spinor-helicity identity, such as those presented in Section. 2.3. For a given numerator term $\mathcal{N}$ and its prefix representation $\mathcal{P}(\mathcal{N})$, we want our networks to construct a mapping $f : \mathcal{P}(\mathcal{N}) \to \mathbb{R}^{d_{\text{emb}}}$ where similar numerators are to be grouped close to one another in the embedding space. To learn this embedding we will rely on *contrastive learning*.

Contrastive learning has been widely used to construct data embeddings that are useful for downstream tasks [35–39], including in high-energy physics. For example, self-supervised contrastive learning was employed in [63] to ensure the invariance of data representations for top and QCD jets under translations and rotations in the $\eta - \phi$ plane. Those representations were then shown to yield superior performance when used as inputs in a downstream classification task. Similarly, supervised contrastive learning was utilized in [64], where embeddings of particle decay tree structures were learned, resulting in improved performance in subsequent reconstruction tasks.

Here the central idea is that for a sample indexed by $i$, we consider its given embedding, $\boldsymbol{z}_i$, called the *anchor*, and associate it with a similar sample $\boldsymbol{z}_p$. Together they form a *positive pair* that will be pulled towards each other within the embedding space. One also forms *negative pairs* by associating the anchor with other non-similar embeddings $\boldsymbol{z}_a$. Taking $I$ to be a batch of samples, the loss function for a positive pair of examples is then given by[8]

$$\mathcal{L}_{i,p} = -\log \frac{\exp(s(\boldsymbol{z}_i, \boldsymbol{z}_p)/\tau)}{\sum_{a \in A(i)} \exp(s(\boldsymbol{z}_i, \boldsymbol{z}_a)/\tau)}, \tag{23}$$

where $A(i) = I \setminus \{i\}$ is the set of all embeddings that are different than $\boldsymbol{z}_i$ and $\tau$ is a temperature parameter. We note here that we use the *cosine similarity*

$$s(\boldsymbol{z}_1, \boldsymbol{z}_2) = \frac{\boldsymbol{z}_1 \cdot \boldsymbol{z}_2}{||\boldsymbol{z}_1|| \, ||\boldsymbol{z}_2||} \tag{24}$$

to characterize the distance between samples in the embedding space. When the anchor is associated with a single positive example, typically obtained via data augmentation, we refer to $\mathcal{L}_{i,p}$ as a self-supervised contrastive loss.

For our purposes, however, we will be interested in the case where a given anchor is associated with different positives. Indeed, since a spinor-helicity identity can involve more than two distinct terms, multiple different terms may be considered similar. For instance, the momentum conservation identity of six-point amplitudes is a linear combination of either four or five terms. Therefore, we associate with each embedding $\boldsymbol{z}_i$ a label $y_i$ and the set of all positives is defined as $P(i) = \{p \in A(i) | y_p = y_i\}$. In our setting, terms that participate in the same identity will thus share the same label. The supervised contrastive loss is then generalized from Eq. (23) as

$$\mathcal{L}_{\text{con}} = -\frac{1}{|I|} \sum_{i \in I} \frac{1}{|P(i)|} \sum_{p \in P(i)} \log \frac{\exp(s(\boldsymbol{z}_i, \boldsymbol{z}_p)/\tau)}{\sum_{a \in A(i)} \exp(s(\boldsymbol{z}_i, \boldsymbol{z}_a)/\tau)}, \tag{25}$$

where we average the log term across all positive pairs.[9]

---

[8]In the literature this loss is known as the Normalized Temperature-scaled Cross Entropy Loss [37], and is derived from the $N$-pair loss [35] by the addition of the temperature parameter.

[9]We note that performing the average over positives *inside* the logarithm is discouraged, having been shown to generically lead to a drop in performance [38].

Table 3: Network architectures and training hyperparameters used for the learning of numerator embeddings via supervised contrastive learning.

| Hyperparameter Type | Parameter Description | Value |
|---|---|---|
| Encoder architecture | Num layers | 2 |
| | Attention heads | 8 |
| | Embedding dimension | 512 |
| | Maximum input length | 256 |
| Feed-forward architecture | Num layers | 2 |
| | Hidden dimension | 512 |
| Training parameters | Batch size | 128 |
| | Epoch size | 10,000 |
| | Epoch number | 500 |
| | Learning Rate | $10^{-4}$ |
| | Temperature | 0.15 |

## 4.2 Grouping terms

To obtain the embeddings in $\mathbb{R}^{d_{\text{emb}}}$ we construct our mapping $f$ by composing a transformer encoder layer and a projection layer. Akin to the procedure described in Section 3.3, following Eq. (22), we obtain an embedding $e(\mathcal{N})$ for a numerator term $\mathcal{N}$ by averaging over all of the word embeddings coming from the encoder layer. This output is then passed through a simple feed-forward network $h(e(\mathcal{N}))$ to yield the final embedding $z$. The network architecture and hyperparameters used are listed in Tab. 3.

To generate relevant training data, we take the pairs of input and output amplitudes $\{\mathcal{M}, \overline{\mathcal{M}}\}$ created in Section 2.2. We isolate the pairs where $\mathcal{M}$ has been obtained by a single scrambling identity and where $\overline{\mathcal{M}}$ is either a constant or an amplitude with a single numerator term. For each $\mathcal{M}$ and $\overline{\mathcal{M}}$ pair we construct a set of all of the numerator terms[10] $S_{\mathcal{N}} = \{\mathcal{N}_1, \mathcal{N}_2, \cdots, \mathcal{N}_g\}$, where $g$ is typically around two to six. All terms in each of these sets are considered similar. Each different set of numerator terms constitutes one data entry of equivalent positives. Following this procedure, we create 635k unique sets of numerator terms for four-point amplitudes. For five-point and six-point amplitudes we have a larger number of unique $S_{\mathcal{N}}$, with 1.15M and 1.38M entries respectively. In each case, we also isolate 10k different $S_{\mathcal{N}}$ sets to generate a validation and a held-out test set.

When forming the validation and test sets we specifically select different $S_{\mathcal{N}}$ that share the same little group scaling. In particular, when adding a new $S_{\mathcal{N}}$ to the validation or test set, we either draw it randomly from the complete entry pool or from the sets of $S_{\mathcal{N}}$ that share the same little group scaling. Similarly, during training, we construct the batches of training samples $I$ by following an analogous procedure, grouping various $S_{\mathcal{N}}$ based on their little group scaling. A batch is composed of 128 different numerator sets, where all numerator terms $\mathcal{N}$ inside a given $S_{\mathcal{N}}^{(h)}$ are attributed the same label $y_h$. The total number of $z$ samples that compose a batch is thus $\sum_{h=1}^{128} |S_{\mathcal{N}}^{(h)}|$, averaging 330, 480, and 700 samples per batch for four, five, and six-point amplitudes, respectively. By including $S_{\mathcal{N}}$ with identical little group scaling in the same batch, we ensure that a given anchor $z_i$ will form a negative pair with examples $z_a$ that share the same scaling, without belonging to the same group of positives. This strategy prevents our networks from learning the trivial embedding where all the numerators that share the same

---

[10]We put $\mathcal{M}$ and $\overline{\mathcal{M}}$ under a common denominator before extracting the numerator terms. Additionally, the numerator of $\overline{\mathcal{M}}$ is multiplied by -1 to guarantee similarity.

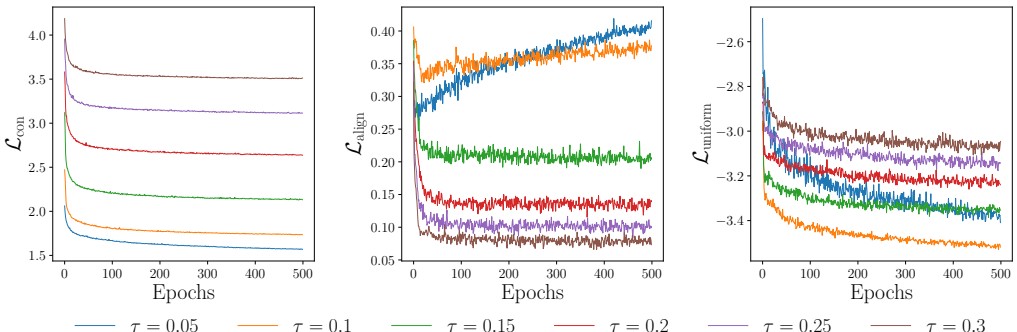

Figure 7: Contrastive, alignment and uniformity losses evaluated on the validation set for five-point amplitudes as a function of the training epoch. The temperature parameter $\tau$ enters the supervised contrastive loss of Eq. (25). A new network is trained for each distinct value of $\tau$.

little group scaling are pulled together in the embedding space.[11]

We apply this batch construction on the fly during training and optimize using the supervised contrastive loss of Eq. (25). Training is done over 500 epochs, lasting around 2h, where each epoch involves 10k different $S_{\mathcal{N}}$. To determine the temperature parameter we sweep for $\tau \in [0.05, 0.1, 0.15, 0.2, 0.25, 0.3]$ and retain the optimal $\tau^\star$ by tracking the alignment and uniformity losses [39, 63] on the validation set. We define those losses as

$$\mathcal{L}_{\text{align}} = \frac{1}{|I|} \sum_{i \in I} \frac{2}{|P(i)|} \sum_{p \in P(i)} 1 - s(\boldsymbol{z}_i, \boldsymbol{z}_p), \tag{26}$$

$$\mathcal{L}_{\text{uniform}} = \log \left[ \frac{1}{|I|} \sum_{i \in I} \frac{1}{|A(i)|} \sum_{a \in A(i)} \exp\left( -4\left(1 - s(\boldsymbol{z}_i, \boldsymbol{z}_a)\right) \right) \right], \tag{27}$$

where low values of $\mathcal{L}_{\text{align}}$ indicate that similar pairs are aligned, while low values of $\mathcal{L}_{\text{uniform}}$ are achieved if the normalized $\boldsymbol{z}_i$ are uniformly distributed on the unit hypersphere in $\mathbb{R}^{d_{\text{emb}}}$. In Fig. 7 we sweep through $\tau$ and display the resulting contrastive, alignment and uniformity losses for five-point amplitudes as a function of the training epochs. We notice on the left panel that the supervised contrastive loss is stable across temperatures but has widely different ranges, as a consequence of $\tau$ directly entering Eq. (25). To compare different $\tau$ values we thus refer to the middle and right panel since $\tau$ does not enter Eq. (26) and Eq. (27). As expected, higher values of $\tau$ lead to a stronger alignment between positive pairs, while lower values of $\tau$ typically indicate a more uniform embedding. We decide to retain $\tau^\star = 0.15$ as the final value of the temperature parameter, prioritizing a more uniform distribution, but other choices would be equally valid.

To assess whether the network learns a useful representation, we can compare the cosine-similarly metric in the embedding space between two terms and the number of simplification steps by which they are related. We start by generating 500 simple random amplitude expressions that have a single numerator term, in the form of Eq. (9). We then apply a scrambling step and retain the numerator terms of the resulting expression. Iterating this procedure ten

---

[11]When training networks where batches are formed by randomly sampling $S_{\mathcal{N}}$ naively, we observe that the embedding space exhibits a trivial disposition, with terms sharing the same little group scaling pulled close to one another, irrespective of their actual similarity. This occurs because, during training, the networks rarely encounter different $S_{\mathcal{N}}$ instances with the same scaling within the same batch, resulting in very few relevant negative pair examples.

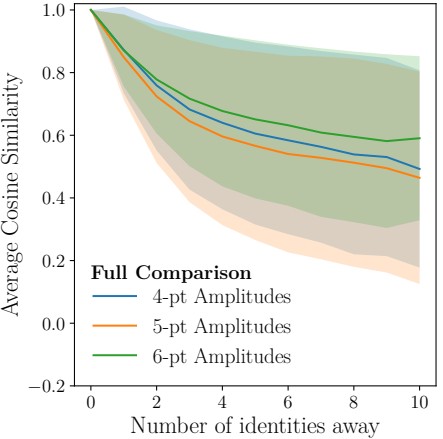
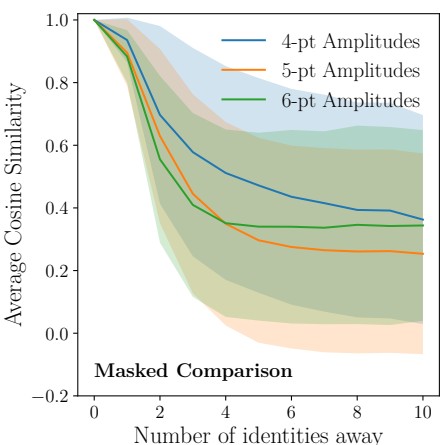

Figure 8: To assess whether the network is learning to group similar terms, we compare the average cosine similarity in the embedding space to the number of identities by which two terms are separated. We represent the one standard deviation using shaded uncertainty bands. We compute the cosine similarity by either looking at the full numerator terms (left panel) or by excluding any common bracket terms (right panel). This confirms that the network is learning to group similar terms.

times gives us sets of numerators that are $n \leq 10$ identities away from the original numerator term. We can then compute the cosine similarity between the original numerator term and a numerator term that is $n$ identities away.[12] To guarantee an accurate comparison, we take care to put both the original amplitude and its scrambled version under a common denominator such that both numerator terms share the same little group scaling. Fig 8 depicts the average cosine similarity as a function of the number of identities. On the left panel, we compare the numerator terms as they are, while on the right panel, we compute the cosine similarity between numerator terms where common brackets are excluded. Lower cosine similarity values are obtained when excluding common brackets, indicating that without this masking our networks are prone to overestimate the similarity between terms that share common factors. Crucially, the learned embedding is well suited for practical purposes since there is a direct correlation between the cosine similarity and the number of identities required to relate two terms. We also stress that this mapping was learned only by forming positive pairs between terms that are a single identity away. At no point did we explicitly add information about terms that were multiple identities away. We note that since one of our identities, momentum squared, can be recovered by a combination of the others, we expect a large standard deviation at more than 3 identities away. We comment on this point further in Appendix F where we further explore the structure of the embedding space, making sure that sufficiently dissimilar terms are not being clustered together.

## 4.3 Simplifying long expressions

Our strategy for lengthy spinor-helicity amplitudes relies on selecting subsets of terms in the full expression that are likely to simplify. Using the trained embedding maps from Section 4.2, we implement an algorithm that is far more efficient than attempting simplification on all possible groupings. From Fig. 8 it is apparent that it suffices to consider numerator terms that have a high cosine similarity with one another, as those are likely to be related by a handful of iden-

---

[12]After applying a scrambling step, we randomly sample a numerator term of the resulting expression to form the next amplitude to be scrambled. This ensures that every time an identity is applied, all of the numerator terms in the resulting expression share the same similarity.

tities. In practice, for a given numerator term $\mathcal{N}_1$, we compute its cosine similarity with all of the other numerator terms in the amplitude and retain those terms $\mathcal{N}_b$ for which $s(\mathbf{z}_1, \mathbf{z}_b) > c$, where $c$ is a cutoff to be fixed. This gives a reduced amplitude that can then be fed through the trained models of Section 3. If a valid simplified form of the reduced amplitude is found, we set it aside, and focus on the remaining numerator terms in the original amplitude. After performing a similar grouping and simplification procedure over all remaining numerator terms, we consider having done a single *pass* over the original amplitude, yielding an equivalent simplified amplitude. We then iterate our simplification algorithm using this new amplitude as input. If no simplification is achieved when parsing through the numerator terms of the input amplitude, we deem it to be maximally simplified and return the expression.[13]

We leave the choice of cutoff $c$ as a parameter in our algorithm but allow for its dynamic update as we iterate the simplification algorithm. Indeed, it is more efficient to start with a higher value of the cutoff, making sure that any groupings we consider will indeed be reducible, before relaxing the cutoff value as we perform more passes through the amplitude. In practice, after $t$ passes, we take our dynamical cutoff to be

$$c(t) = c_0^{1+t\alpha}, \tag{28}$$

where both $c_0$, the initial cutoff, and $\alpha$, the decay parameter, are to be chosen. As we increase the number of passes, we reduce the overall number of numerator terms and hence our simplification algorithm can still run in a reasonable time, even as $c(t)$ decreases. By default, we will consider only the masked comparison scheme when computing the cosine similarity between terms, as it is more representative of the actual similarity distance between terms. Additionally, as described in Section 2.2, our simplifier models are mainly trained on expressions where the numerator terms are dressed with an overall numerical coefficient of $\pm 1$. Therefore, when simplifying an expression, we choose by default to blind all constants.[14]

## 4.4 Physical amplitudes

To benchmark the performance of our models we test them on explicit four- and five-point scattering amplitudes that appear in physical theories like Yang-Mills theory and gravity. All of these amplitudes can be computed using known methods, which we summarize now. Some example amplitudes and their simplified forms are given in Appendix G.

We compute the tree-level gluon amplitudes of Yang-Mills theory using standard planar Feynman diagrams. In their raw form, these amplitudes are complicated explicit functions of four-momenta and polarizations which are then mechanically recast in terms of spinor-helicity variables. As usual, polarizations are ambiguously defined, so expressing them in terms of spinor-helicity variables requires arbitrary choices for certain reference spinors. For this analysis we consider all possible choices of reference—that is, every possible assignment of the polarization reference spinor to an arbitrary external leg. Since all four- and five-point

---

[13]We deem an amplitude to be corresponding to a simpler form if it has either fewer numerator terms, or if the numerical coefficients in front of those have been reduced. We check the validity of the reduction numerically on two sets of light-like momenta. If we find no simplifications in the amplitude, then we perform "shuffling" passes, where we also allow for valid solutions with the same number of terms and numerical coefficients. Considering these equivalent rewriting proves to be useful for the simplifier model.

[14]Practically this implies that when reducing $\mathcal{M} = a_1 \mathcal{N}_1 + a_2 \mathcal{N}_2$, with $|a_1| < |a_2|$ we first reduce $\mathcal{N}_1 + \mathcal{N}_2 \to \mathcal{N}_r$ before returning $\overline{\mathcal{M}} = (a_2 - a_1)\mathcal{N}_2 + a_1\mathcal{N}_r$. This choice is well suited for the maximal reduction of the numerical coefficients in front of the numerator terms. Other procedures could also be considered. One could for instance isolate numerator terms with proportional coefficients when searching for similar terms, or even generate new training data with non-trivial coefficients. When numerator coefficients arise from expanding a linear combination of terms to a power, one might instead consider first factorizing the amplitude before attempting to simplify each factor independently.



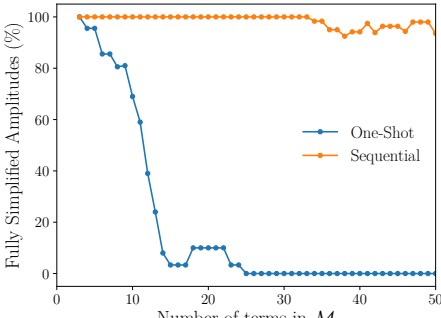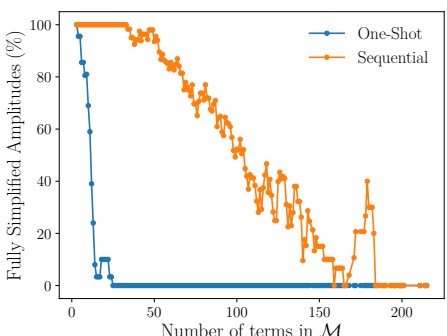

Figure 9: Performance of our models in simplifying four- and five-point scattering amplitudes in physical theories. The left panel focuses on 315 amplitudes with up to 50 terms distinct terms. The right panel includes some additional 850 amplitudes that have up to 200 numerator terms. The one-shot simplification (blue) generates 100 candidate amplitudes for the final result in one go, while the sequential simplification (orange) focuses on small subexpressions, iteratively simplifying those. Our results are displayed using a rolling window of size five.

amplitudes in Yang-Mills theory are maximally helicity violating, the optimally simplified target amplitudes are the monomial Parke-Taylor expressions.

For the case of gravity, we compute the tree-level graviton amplitudes by inserting the complicated Feynman diagram expressions for the gluon amplitudes into the KLT formula [65]. Here the optimally simple target amplitude is the expression obtained by inserting the Parke-Taylor form of the gluon amplitudes into KLT. To obtain amplitudes involving scalars and gravitons together, we employ the differential operators defined in [66].

To begin, we analyze the maximally helicity violating tree amplitudes of four- and five-point gluon scattering and four-point graviton scattering.[15] We organize the amplitudes based on their number of numerator terms, attempt a one-shot simplification and contrast the resulting performance with sequential simplification. For the one-shot simplification, we generate 100 candidates using both beam search and nucleus sampling, deeming an amplitude to be simplified if it has decreased in complexity. For sequential simplification, we consider the cutoff parameters $(c_0, \alpha) \in \{(0.9, 0.25), (0.9, 1), (0.95, 1), (0.95, 2)\}$, run the algorithm multiple times, and retain the most simplified result. The resulting amplitude is deemed valid only if it has been successfully reduced to the single Parke-Taylor monomial term. In this setting, we increase the execution speed of our algorithm by requiring the internal simplifier model to output only 10 total candidate expressions when attempting a simplification.

Fig. 9 demonstrates the performance of both the one-shot simplification approach and the sequential simplification approach using contrastive learning. As soon as the input amplitudes are more than about ten terms in length, the one-shot models struggle to find a valid simplified form. This is expected because our four-point amplitude training data only contained expressions with fewer than ten terms, while only 15% of the five-point data had more than ten terms. Additionally, we suspect that most of the lengthier physical amplitudes are only reachable through greater than three spinor-helicity identities. Therefore, these expressions also fall outside of the scope of our trained models in the one-shot simplification scheme. However, as inferred from the left panel in Fig. 9, the sequential simplification allows us to alleviate this issue, retaining near-perfect accuracy for inputs of up to 40 terms. Notably, 99% of four-point amplitudes are successfully fully simplified following this procedure, while 97%

---

[15]We do not consider five-point graviton amplitudes since the corresponding inputs can reach many thousands of terms.

of the five-point amplitudes with fewer than 50 terms get correctly reduced to a single term. Even in instances where the iterative scheme does not find the most simplified form, it still reduces the input amplitudes to about 18% of their original length. From the right panel in Fig. 9 we observe that with our current settings (beam size and cutoff parameters), our iterative algorithm is less successful as the number of numerator terms increases. As we consider inputs with more than 100 terms in the numerator, less than half of those amplitudes are fully reduced to the Parke-Taylor formula. Nevertheless, we have that, on average, the amplitudes are reduced to about 13% of their original length, showcasing a drastic reduction in complexity. We expect that increasing the beam size at inference along with an additional tuning of the cutoff parameters could further enhance the performance of the iterative scheme.[16]

As a final test, we consider physical five-point scattering amplitudes involving scalars and gravitons, which are given explicitly in Appendix G. The theory under consideration describes a massless scalar that is minimally coupled to gravity, with a cubic self-interaction vertex. Notably, using the sequential simplification scheme we are able to reduce an input amplitude for three scalars and two same-helicity gravitons with 298 terms (13,256 tokens) down to its simplest form of two terms. The resulting expression is

$$\overline{\mathcal{M}}(\phi\phi\phi h^+ h^+) = \frac{\langle 12\rangle\langle 13\rangle\langle 23\rangle}{\langle 24\rangle\langle 25\rangle\langle 45\rangle}\left(\frac{[14][35]}{\langle 14\rangle\langle 35\rangle} - \frac{[15][34]}{\langle 15\rangle\langle 34\rangle}\right). \tag{29}$$

The entire simplification procedure takes about 30 minutes for this example, although the efficiency and speed of our algorithm could be further optimized. Sequential simplification can also reduce amplitudes for four scalars and one graviton[17] with 29 terms (661 tokens) to the far simpler result

$$\overline{\mathcal{M}}(\phi\phi\phi\phi h^+) = \frac{[25][45]}{\langle 15\rangle\langle 35\rangle[14][23]}\left(1 + \frac{\langle 14\rangle\langle 34\rangle[14]}{\langle 23\rangle\langle 45\rangle[25]} - \frac{\langle 12\rangle\langle 23\rangle[23]}{\langle 14\rangle\langle 25\rangle[45]}\right). \tag{30}$$

To our knowledge, these simplified expressions have not appeared before in the literature. The existence of these compact expressions is perhaps not so surprising, since these amplitudes can be computed by inserting the self-dual one-point function of the graviton into the propagators of a pure scalar amplitude. Nevertheless, it is quite nice that the transformer model can hone in on these results mechanically.

## 5 Conclusion

We have used ML to simplify scattering amplitudes written in terms of spinor-helicity variables. Physical processes in gauge theories and gravity typically correspond to amplitude expressions with many hundreds or even thousands of terms, at least when computed using textbook Feynman diagrams. The transformer models presented here have the capacity to drastically reduce such expressions. For example, when fed four and five-point gluon tree amplitudes with hundreds of terms, our models can arrive at the Parke-Taylor formula, which is a single monomial term.

---

[16]Further algorithmic improvements can naturally be considered. For instance, the iterative scheme can halt if terms only simplify through some non-trivial identity or with an intermediate increase in complexity. We could therefore imagine either training networks on a wider set of identities, or allowing for them to output equivalent, but more complex intermediate forms. Alternatively, it is also possible to imagine integrating an MCTS-inspired search, akin to [30], where intermediate groupings and simplifications can be viewed as nodes of a simplification tree. We leave the exploration of these improvements for future work.

[17]Here we have chosen flavour structure for the scalars so that they only self-interact through the exchange of a scalar in a single channel.

The training set for our transformer models was built by randomly generating simple target spinor-helicity expressions $\overline{\mathcal{M}}$ and then scrambling them into a more complicated form $\mathcal{M}$ using the mathematical identities of momentum conservation and the Schouten relation. Our models were then trained to deduce $\mathcal{M}$ from $\overline{\mathcal{M}}$. Networks which implemented one-shot simplification were able to simplify expressions containing up to around ten terms. Considering that amplitudes arising in physical calculations typically have many more than this, we also introduced a sequential simplification algorithm to tackle those longer expressions. Using contrastive learning, we trained embedding networks capable of grouping terms likely to simplify. Enhanced by this contrastive learning step, the sequential simplification networks are then able to reduce amplitudes containing hundreds of terms. For example, an amplitude arising from the scattering of three scalars and two gravitons is correctly simplified from 298 terms down to just two.

The simplification of spinor-helicity amplitudes using ML showcases the efficacy of this approach when explicit analytical algorithms are challenging to design. More generally, this work demonstrates how a practical ML tool can be implemented for calculations in high-energy physics, yielding novel amplitude formulas, and paving the way for more automated symbolic manipulations. Crucially, our results are verifiable as we obtain exact formulas which can be checked through explicit numerical evaluations. Our framework is flexible enough to be adapted to related problems. For instance, it should be possible to train networks to grapple with expressions written in terms of momentum twistor variables, or with factors other than two-particle spinor brackets. More generally, we expect transformer models to play an increasingly larger role in solving mathematical problems, where answers can be easily cross-checked and incoherent model hallucinations can be systemically avoided. For the approach advocated in this paper to be applicable, the primary requirement is the generation of synthetic training data. If data is available, then, instead of developing an entirely new classical algorithm for each simplification task, one can simply recycle the same ML framework.

# Acknowledgments

The computations in this paper were run on the FASRC Cannon cluster supported by the FAS Division of Science Research Computing Group at Harvard University.

**Funding information** AD and MDS are supported in part by the National Science Foundation under Cooperative Agreement PHY-2019786 (The NSF AI Institute for Artificial Intelligence and Fundamental Interactions). CC is supported by the Department of Energy (Grant No. DE-SC0011632) and by the Walter Burke Institute for Theoretical Physics. This work was initiated in part at the Aspen Center for Physics, which is supported by National Science Foundation grant PHY-2210452.

# A   Parsing amplitudes

Passing a spinor-helicity amplitude through a transformer network is only possible once it has been converted into a set of ordered tokens, mimicking the structure of words in sentences. To transform a mathematical expression into a set of tokens we utilize the tree-like structure

of amplitudes and parse them using prefix notation:

$$\frac{[12][34]}{\langle 12\rangle\langle 34\rangle} = \qquad\qquad = [\text{'mul'}, \text{'pow'}, \text{'ab12'}, \text{'-1'}, \text{'mul'}, \cdots]. \qquad (A.1)$$

The prefix representation of a given expression is obtained by reading the tree from left to right, starting with the topmost node. In Eq. (A.1) the orange nodes represent binary operators which in our case can be[18] $[\times, +, \text{pow}]$. The green nodes are the tree leaves and can contain either integers or square and angle brackets. In this paper, we choose to represent integers by following a base 10 decomposition. Explicitly, for an integer $c$ we have $c = \sum_j a_j 10^j$, so that, for example, the integer 12 gets parsed as ['add', '10', '2']. Adopting a base 10 decomposition scheme has already been proved useful when training transformers [67] and allows us to retain a dictionary with a small list of required words. We emphasize that we choose to represent each square and angle bracket with its own unique token instead of treating the opening and closing brackets as binary operators. This allows for a more compact representation of the amplitude and smaller sequence lengths. The total number of tokens required to describe the brackets for $n$ external legs is then $n(n-1)/2$ once we use antisymmetry to impose a canonical ordering for the external particle labels.

## B  Training data composition

Section 2.2 describes how we generate sample training pairs of input and output amplitudes $\{\mathcal{M}, \overline{\mathcal{M}}\}$. Our main experiments are run on training sets that are composed of one to three scrambles at most, where expressions longer than 1k tokens are discarded. This cutoff has little impact on four-point amplitudes, of which fewer than 0.1% are discarded, but becomes more relevant at higher-point. Indeed, our data generation procedure discards 2% of five-point amplitudes and 8% of six-point amplitudes. One might worry that we are throwing away relevant data by imposing this cut, but the majority of the amplitudes discarded in that way are associated with a higher number of scrambles[19] and can still be simplified with the techniques of Section 4. In addition to restricting the number of scrambles, we also only retain output amplitudes that have at most three distinct numerator terms. After scrambling, from the left panel of Fig. 10 we can infer that this translates in input amplitudes having at most 10, 20 and 30 numerator terms for four-, five-, and six-point amplitudes, respectively.

To characterize the resulting training data we also find it helpful to define a measure of complexity. We define the complexity of an amplitude as the number of distinct square and angle brackets that compose it, being agnostic about the power at which these brackets are raised. This measure of complexity is calculated for an amplitude that is rationalized, with its numerator terms written in a fully expanded form, and serves to indicate the compactness of the associated expression. For instance, the amplitudes of Eqs. (20-21) have complexities of 25 and 5 respectively. In the centre and right panels of Fig. 10 we estimate the complexity of

---

[18]In our word dictionary we also add the division operator for efficiently representing rational numbers, although we do not train on that data specifically.

[19]For example, 1% of the six-point amplitudes are discarded if we have a single scrambling move, whereas we throw away close to 15% of them when using three scrambling moves.

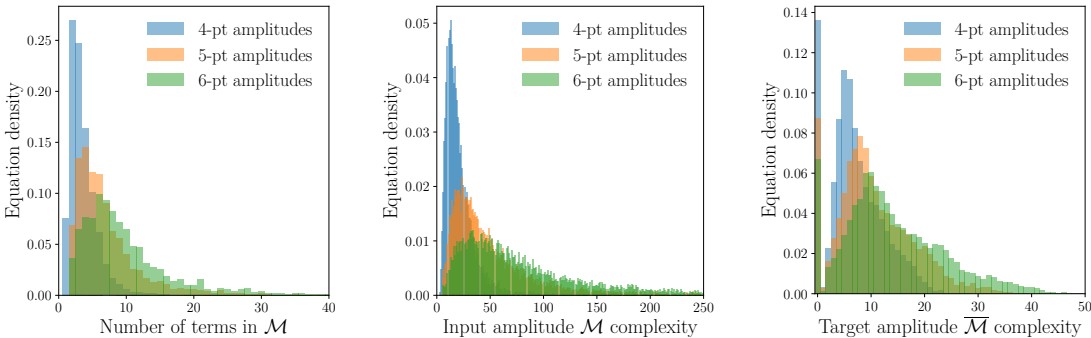

Figure 10: (Left panel) Number of distinct numerator terms in the input amplitudes $\mathcal{M}$. (Center and right panels) Complexity measure of the input and output amplitudes pairs $\{\mathcal{M}, \overline{\mathcal{M}}\}$ contained in the validation set. Our measure of complexity is defined as the number of distinct bracket terms contained in the amplitude.

Table 4: Samples of pairs of input and output amplitudes composing the training set for four, five and six-point amplitudes respectively. The samples selected are of median complexity.

| Input Amplitude $\mathcal{M}$ | Target Amplitude $\overline{\mathcal{M}}$ |
|---|---|
| $\dfrac{\langle 34 \rangle [13]^2 [34]^3}{\langle 23 \rangle [14]} + \dfrac{2\langle 34 \rangle [13]^2 [23][34]^2}{\langle 23 \rangle [12]} + \dfrac{\langle 34 \rangle [13]^2 [14][23]^2 [34]}{\langle 23 \rangle [12]^2}$ | $\dfrac{\langle 12 \rangle [13]^4 [24]^2}{\langle 23 \rangle [12][14]}$ |
| $\dfrac{\langle 12 \rangle^3 \langle 14 \rangle \langle 15 \rangle^2 [12][14]^2}{\langle 35 \rangle^3 [23][35]^3} - \dfrac{\langle 12 \rangle^3 \langle 14 \rangle \langle 15 \rangle [14]^2}{\langle 35 \rangle^2 [35]^3} + \dfrac{\langle 12 \rangle^3 \langle 15 \rangle^3 [12][14][15]}{\langle 35 \rangle^3 [23][35]^3}$ $- \dfrac{\langle 12 \rangle^3 \langle 15 \rangle^2 [14][15]}{\langle 35 \rangle^2 [35]^3} + \dfrac{\langle 12 \rangle^3 \langle 15 \rangle^2 \langle 45 \rangle [12][14][45]}{\langle 35 \rangle^3 [23][35]^3} - \dfrac{\langle 12 \rangle^3 \langle 15 \rangle \langle 45 \rangle [14][45]}{\langle 35 \rangle^2 [35]^3}$ | $\dfrac{\langle 12 \rangle^3 \langle 15 \rangle \langle 23 \rangle \langle 45 \rangle [14][24]}{\langle 35 \rangle^3 [35]^3}$ |
| $\dfrac{\langle 15 \rangle \langle 26 \rangle [14][46]^2}{\langle 13 \rangle \langle 45 \rangle^2 \langle 56 \rangle [12][15][16][56]} + \dfrac{\langle 25 \rangle [24][46]^2}{\langle 13 \rangle \langle 45 \rangle^2 [12][15][16][26]}$ $- \dfrac{\langle 25 \rangle [25][46]^3}{\langle 13 \rangle \langle 45 \rangle^2 [12][15][16][26][56]} + \dfrac{\langle 16 \rangle \langle 25 \rangle [13][14][46]^2}{\langle 12 \rangle \langle 45 \rangle^2 \langle 56 \rangle [12]^2 [15][16][56]}$ $- \dfrac{\langle 16 \rangle \langle 25 \rangle [14][45][46]^2}{\langle 12 \rangle \langle 13 \rangle \langle 45 \rangle \langle 56 \rangle [12]^2 [15][16][56]} + \dfrac{\langle 16 \rangle \langle 23 \rangle \langle 25 \rangle [14][23][46]^2}{\langle 12 \rangle \langle 13 \rangle \langle 45 \rangle^2 \langle 56 \rangle [12]^2 [15][16][56]}$ $- \dfrac{\langle 16 \rangle \langle 25 \rangle \langle 46 \rangle [14][46]^3}{\langle 12 \rangle \langle 13 \rangle \langle 45 \rangle^2 \langle 56 \rangle [12]^2 [15][16][56]} - \dfrac{\langle 16 \rangle \langle 25 \rangle [14][46]^2}{\langle 12 \rangle \langle 13 \rangle \langle 45 \rangle^2 [12]^2 [15][16]}$ | $\dfrac{\langle 12 \rangle [14][46]^2 - \langle 25 \rangle [45][46]^2}{\langle 13 \rangle \langle 45 \rangle^2 [12][15][16][56]}$ |

the amplitude pairs that our models train on, sorting them based on the number of distinct external momenta. Whereas the complexity of typical four-point amplitudes only doubles after scrambling, the six-point amplitude's complexities increase more than fourfold. Examples of median complexity for the input and output amplitudes composing the training set are given in Tab. 4.

## C   Nucleus sampling calibration

In Section 3.2 we utilize an alternative to beam search at inference time which is known as nucleus sampling [61]. When generating the predictions of a model, each new token is sampled from a set $V(p_n)$ that is constructed so the summed probability distribution of the

tokens in that set exceeds $p_n$. In mathematical terms, this implies that

$$\sum_{x \in V(p_n)} P(x|\text{context}) \geq p_n \,, \tag{C.1}$$

where $P(x|\text{context})$ is the probability assigned by the model to the token $x$. In our case, this probability is conditioned on a context given by the input amplitude fed through the encoder network and the output tokens that have already been generated. The reason for creating the set $V(p_n)$ is that we entirely forbid the sampling of irrelevant tokens and greatly reduce the generation of false outputs. For our purposes, we also utilize *temperature* to first shape the probability distributions over tokens [68]. Given the outputs (logits), $u_j$, of the transformer decoder's final projection layer, the probability distribution over tokens $x_j$ is described by

$$P(x_j|\text{context}) = \frac{\exp(u_j/\tau)}{\sum_k \exp(u_k/\tau)} \,, \tag{C.2}$$

where $\tau$ is the temperature parameter and the sum over $k$ is to be taken over all possible words. Using a small temperature value $\tau < 1$ skews the distribution to favour high probability tokens compared to the $\tau = 1$ regime which is the usual softmax function. In the opposite limit, at high temperatures $\tau > 1$, the probability distribution starts to even out. Allowing for high temperatures in nucleus sampling enlarges the size of the set $V(p_n)$ and diversifies the outputs of the model. To calibrate our nucleus sampling inference we can tune both $p_n$ and $\tau$. We do so by looking at the accuracy of our models on a held-out validation set for the one-shot simplification of five-point amplitudes. We sweep across parameters $\tau$ and $p_n$ using nucleus sampling at inference, sampling a total of ten model predictions. If any one of those predictions is a valid simplified form of the input amplitude, then we deem the input amplitude simplified accurately. In the left panel of Fig. 11 we observe that the temperature parameter is the one with the highest impact, whereas changing $p_n \in [0.9, 0.975]$ has minimal effects. We select the optimal temperature parameter $\tau^\star = 1.25$ and the nucleus probability cutoff $p_n^\star = 0.925$. In the right panel of Fig. 11 we plot the combined accuracy with a beam search of size ten where we can assert that the integration of both techniques yields an increased performance. Nucleus sampling with higher temperatures is then able to generate some of the model outputs that beam search is not able to recover.

## D    Training on intricate amplitudes

In Section 3.2 we limited our experiments to models trained on amplitudes at most three scrambling identities away from their optimally simple form. We asserted that the performance of our models drops as we consider amplitudes with an increasing number of scrambles. For instance, using both beam search and nucleus sampling to generate 20 candidate solutions we have 99.9%, 98.7% and 93.6% accuracy on five-point amplitudes generated with one, two, and three scrambles respectively. However, when testing our model on amplitudes generated with four or five scrambles our performance drops to 74.0% and 46.2% respectively. Thus our models cannot generalize easily to more intricate amplitudes. Instead, we consider training on five-point amplitudes that are up to five identities away in Fig. 12. The blue and orange curves correspond to models that have seen 10M different training examples, compared to 20M for the green curve. We can see that doubling the size of the training set only results in a minor boost in performance, irrespective of the beam size used at inference. The fact that the boost in performance is so minor leads us to believe that seeking improvement by further increasing the size of the training set is not an optimal strategy. We also note that, for amplitudes that are

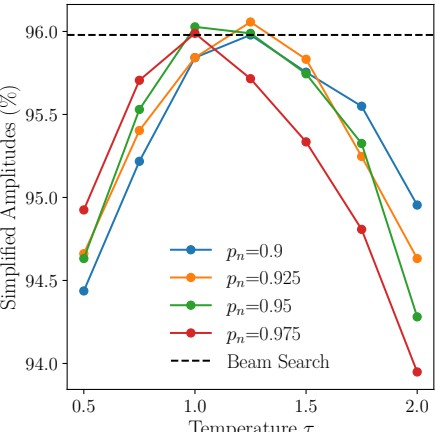
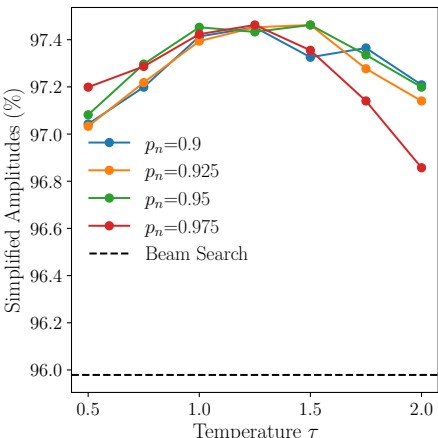

Figure 11: Accuracy on the held-out validation set for the one-shot simplification of complicated five-point spinor-helicity amplitudes. On the left panel, we only use nucleus sampling, generating ten different candidates and retaining the most promising one. On the right panel, we combine nucleus sampling with a beam search of size ten. In both cases we sweep across the probability cutoff $p_n$ of Eq. (C.1) and the temperature parameter $\tau$ of Eq. (C.2). The black dashed line indicates the accuracy when only using a beam search of size ten.

one or two scrambles away, the best performance is still achieved when the training data only contains amplitudes that are at most three identities away. Since that is the domain of interest for the sequential simplification of amplitudes, we do not push further the development of the one-shot simplification of more intricate amplitudes.

# E  Integer embeddings

To visualize the embeddings learned by our transformer models we follow [69] and calculate the cosine similarity between the learned integer embeddings. After passing through the transformer's initial embedding layer, each token $t$ is represented in the 512-dimensional embedding space by a vector $e(t)$. The cosine similarity between two tokens $t_1$ and $t_2$ is the dot product between their respective vectors

$$s(t_1, t_2) = \frac{e(t_1) \cdot e(t_2)}{||e(t_1)|| \, ||e(t_2)||}, \tag{E.1}$$

taking values in $[-1, 1]$. Higher similarity values indicate that the associated integer embeddings are closely aligned in the embedding space. In Fig. 13 we represent the similarity matrices for the models trained on four, five and six-point amplitude data. We can observe that the embedding of any integer $i$ is closely aligned with the embedding of $i \pm 1$, especially for the four-point model, indicating that our models have learned some of the sequential nature of integers. The features are not as striking as in [69] though, which is expected as most of our integers only appear as power exponents when representing amplitudes, rather than overall numerical factors. Nonetheless, we can also notice that the models have started to learn common divisors between integers. In particular, the integers four and eight have a high similarity even though they are nonconsecutive integers.

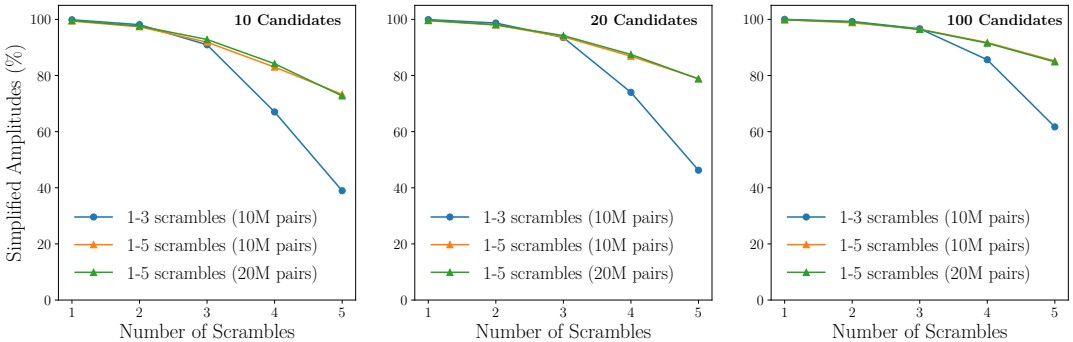

Figure 12: Accuracy on the held-out test set for the one-shot simplification of compli­cated five-point spinor-helicity amplitudes. We compare models trained on datasets with 10M training example pairs to models trained on a larger dataset of 20M exam­ple pairs. Circle markers indicate models that have seen amplitudes scrambled up to three times at most during training while triangle markers indicate models training on up to five scrambles.

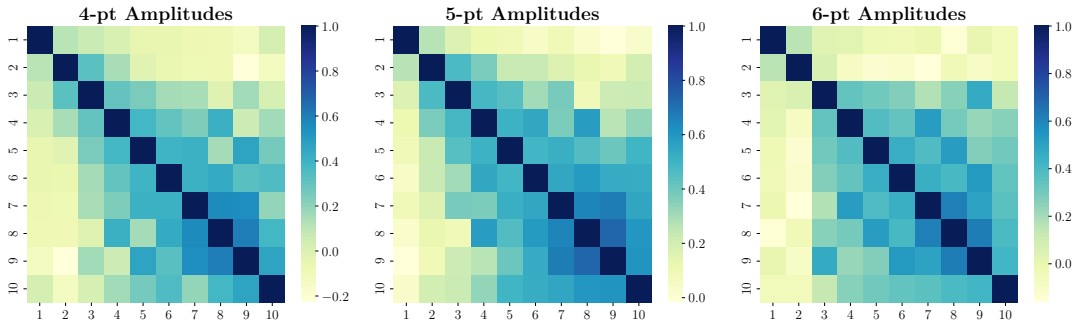

Figure 13: Cosine similarity between the integer embeddings learned by our models when doing a one-shot simplification of $n$-point spinor helicity amplitudes.

## F  Cosine similarity for dissimilar terms

In Section 4.2 we constructed a mapping $f$ for projecting numerator terms in an embedding space $\mathbb{R}^{d_{\mathrm{emb}}}$ where similar terms (involved in the same spinor-helicity identity) are close to one another. We further checked that the distance between numerator terms was correlated with the number of scrambling steps that separated them. For instance, taking into account the standard deviations, for five point amplitudes at a single identity away we have an average cosine similarity of $0.85 \pm 0.14$ in the full comparison case and of $0.90 \pm 0.11$ in the masked case. As we reach 4 identities away we drop to $0.60 \pm 0.26$ and $0.35 \pm 0.32$ respectively. We observe that in all cases the standard deviation levels off at around $\pm 0.3$ for more than 3 identities. We believe this to be an artifact of our testing procedure. Indeed, since one of the identities we have considered (momentum squared) can be recovered by a successive application of the others (Schouten and momentum conservation), we can recover the same amplitude terms through a varying number of identities. Crucially, the high cosine similarity regime remains well correlated with numerator terms that are separated by less than 3 identities.

It remains to check, however, that any numerator term is sufficiently far away in the em­bedding space from completely dissimilar terms. For instance, we expect that numerators that do not share the same mass dimension should not be close to one another. This is verified in Fig. 14 where we represent the average cosine similarity between numerator terms as a

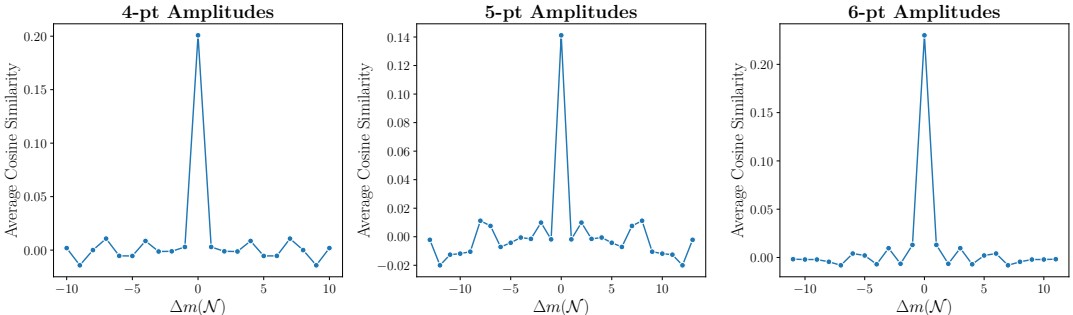

Figure 14: Average cosine similarity across different numerator terms as a function of their mass dimension difference $\Delta m(\mathcal{N})$. The numerator terms are sampled from the respective test sets.

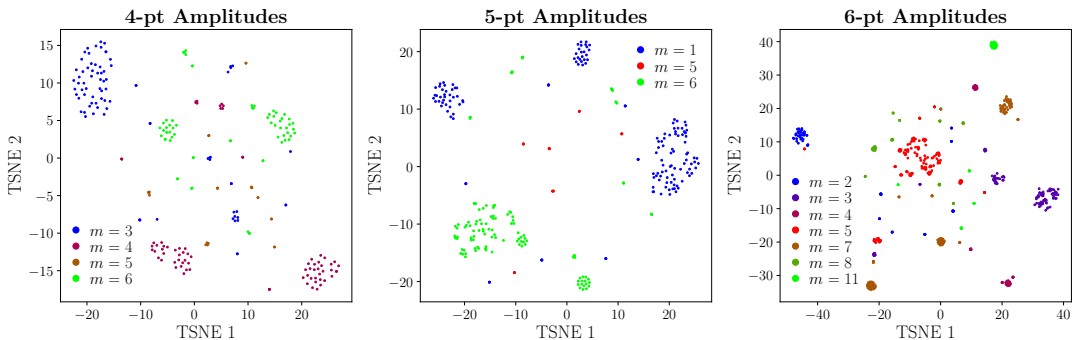

Figure 15: t-SNE visualization of learned numerator embeddings obtained following the mapping described in Section 4.2. The numerator terms are sampled from the respective test sets and are color-coded according to their mass dimension.

function of their difference in mass dimension. The numerator terms are extracted from 100 randomly sample $\mathcal{S}_{\mathcal{N}}$ sets from the test set. Whenever $\Delta m(\mathcal{N}) > 0$ we can observe that the cosine similarity sharply drops to zero, indicating that on average terms with different mass dimensions are uniformly distributed on the hypersphere in $\mathbb{R}^{d_{\text{emb}}}$. Terms with the same mass dimension still have low cosine similarities on average, reaching up to 0.3 at most for six-point amplitudes. This is expected as terms with the same dimension are not always similar and could either be many identities away or potentially unrelated. This intuition is verified in Fig. 15 where we display a t-SNE visualization of various numerator embeddings. Terms with the same mass dimension tend to form distinct clusters, but we also observe that different clusters can form even across terms sharing the same mass dimension. Within a unique cluster terms typically share the same little group scaling and are related via spinor helicity identities.

# G Physical amplitudes

To benchmark the sequential simplification scheme developed in Section 4.3 we feed in complicated input amplitudes that arise in physical theories involving gluons, gravitons, and scalars. In the following, we give some example input amplitudes that are successfully simplified by our model.

• Four gluons: $\mathcal{M}(g^-g^-g^+g^+)$. When expressed in terms of polarization vectors, the color-stripped four-point amplitude is

$$\mathcal{M} = -(\epsilon_1\cdot\epsilon_4)(\epsilon_2\cdot\epsilon_3) + (\epsilon_1\cdot\epsilon_3)(\epsilon_2\cdot\epsilon_4) - (\epsilon_1\cdot\epsilon_2)(\epsilon_3\cdot\epsilon_4) + \frac{(\epsilon_1\cdot\epsilon_4)(p_1\cdot\epsilon_2)(p_1\cdot\epsilon_3)}{(p_1\cdot p_2)} \tag{G.1}$$

$$- \frac{(\epsilon_1\cdot\epsilon_3)(p_1\cdot\epsilon_2)(p_1\cdot\epsilon_4)}{(p_1\cdot p_2)} - \frac{(\epsilon_2\cdot\epsilon_4)(p_1\cdot\epsilon_3)(p_2\cdot\epsilon_1)}{(p_1\cdot p_2)} + \frac{(\epsilon_2\cdot\epsilon_3)(p_1\cdot\epsilon_4)(p_2\cdot\epsilon_1)}{(p_1\cdot p_2)} + \frac{(\epsilon_1\cdot\epsilon_4)(p_1\cdot\epsilon_2)(p_2\cdot\epsilon_3)}{(p_1\cdot p_2)}$$

$$- \frac{(\epsilon_1\cdot\epsilon_2)(p_1\cdot\epsilon_4)(p_2\cdot\epsilon_3)}{(p_1\cdot p_2)} - \frac{(\epsilon_2\cdot\epsilon_4)(p_2\cdot\epsilon_1)(p_2\cdot\epsilon_3)}{(p_1\cdot p_2)} - \frac{(\epsilon_1\cdot\epsilon_3)(p_1\cdot\epsilon_2)(p_2\cdot\epsilon_4)}{(p_1\cdot p_2)} + \frac{(\epsilon_1\cdot\epsilon_2)(p_1\cdot\epsilon_3)(p_2\cdot\epsilon_4)}{(p_1\cdot p_2)}$$

$$+ \frac{(\epsilon_2\cdot\epsilon_3)(p_2\cdot\epsilon_1)(p_2\cdot\epsilon_4)}{(p_1\cdot p_2)} - \frac{(\epsilon_3\cdot\epsilon_4)(p_1\cdot\epsilon_2)(p_3\cdot\epsilon_1)}{(p_1\cdot p_2)} + \frac{(\epsilon_3\cdot\epsilon_4)(p_2\cdot\epsilon_1)(p_3\cdot\epsilon_2)}{(p_1\cdot p_2)} + \frac{(\epsilon_2\cdot\epsilon_3)(p_1\cdot\epsilon_4)(p_2\cdot\epsilon_1)}{(p_2\cdot p_3)}$$

$$+ \frac{(\epsilon_1\cdot\epsilon_4)(p_1\cdot\epsilon_2)(p_2\cdot\epsilon_3)}{(p_2\cdot p_3)} - \frac{(\epsilon_1\cdot\epsilon_2)(p_1\cdot\epsilon_4)(p_2\cdot\epsilon_3)}{(p_2\cdot p_3)} - \frac{(\epsilon_2\cdot\epsilon_4)(p_2\cdot\epsilon_1)(p_2\cdot\epsilon_3)}{(p_2\cdot p_3)} + \frac{(\epsilon_2\cdot\epsilon_3)(p_2\cdot\epsilon_1)(p_2\cdot\epsilon_4)}{(p_2\cdot p_3)}$$

$$- \frac{(\epsilon_2\cdot\epsilon_4)(p_2\cdot\epsilon_3)(p_3\cdot\epsilon_1)}{(p_2\cdot p_3)} + \frac{(\epsilon_2\cdot\epsilon_3)(p_2\cdot\epsilon_4)(p_3\cdot\epsilon_1)}{(p_2\cdot p_3)} - \frac{(\epsilon_1\cdot\epsilon_4)(p_1\cdot\epsilon_3)(p_3\cdot\epsilon_2)}{(p_2\cdot p_3)} + \frac{(\epsilon_1\cdot\epsilon_3)(p_1\cdot\epsilon_4)(p_3\cdot\epsilon_2)}{(p_2\cdot p_3)}$$

$$+ \frac{(\epsilon_3\cdot\epsilon_4)(p_2\cdot\epsilon_1)(p_3\cdot\epsilon_2)}{(p_2\cdot p_3)} + \frac{(\epsilon_3\cdot\epsilon_4)(p_3\cdot\epsilon_1)(p_3\cdot\epsilon_2)}{(p_2\cdot p_3)} - \frac{(\epsilon_1\cdot\epsilon_4)(\epsilon_2\cdot\epsilon_3)(p_1\cdot p_2)}{(p_2\cdot p_3)} - \frac{(\epsilon_1\cdot\epsilon_2)(\epsilon_3\cdot\epsilon_4)(p_2\cdot p_3)}{(p_1\cdot p_2)}.$$

In terms of spinor helicity variables with reference vectors $r_1^\mu = p_4^\mu, r_2^\mu = p_1^\mu, r_3^\mu = p_4^\mu$ and $r_4^\mu = p_3^\mu$, this reduces to

$$\mathcal{M} = - \frac{\langle 12\rangle\langle 13\rangle\langle 24\rangle[13][24]}{\langle 23\rangle\langle 34\rangle^2[12][23]} + \frac{\langle 12\rangle\langle 13\rangle\langle 24\rangle[14]}{\langle 23\rangle\langle 34\rangle^2[12]} - \frac{\langle 12\rangle\langle 24\rangle[13][24]^2}{\langle 34\rangle^2[12][14][23]} + \frac{\langle 12\rangle\langle 24\rangle[24]}{\langle 34\rangle^2[12]}$$

$$+ \frac{\langle 12\rangle[13][24][34]}{\langle 34\rangle[12][14][23]} - \frac{\langle 12\rangle[34]}{\langle 34\rangle[12]} - \frac{\langle 13\rangle\langle 14\rangle[13][34]}{\langle 34\rangle^2[12][23]} - \frac{\langle 13\rangle\langle 24\rangle[13][24][34]}{\langle 34\rangle^2[12][14][23]}$$

$$+ \frac{\langle 13\rangle\langle 24\rangle[34]}{\langle 34\rangle^2[12]} - \frac{\langle 13\rangle\langle 24\rangle[13][24]}{\langle 34\rangle^2[12]^2} + \frac{\langle 13\rangle\langle 24\rangle[14][23]}{\langle 34\rangle^2[12]^2} + \frac{\langle 13\rangle[13][34]^2}{\langle 34\rangle[12][14][23]}$$

$$- \frac{\langle 14\rangle\langle 23\rangle[34]}{\langle 34\rangle^2[12]} - \frac{\langle 23\rangle\langle 24\rangle[13][24]^2}{\langle 34\rangle^2[12]^2[14]} + \frac{\langle 23\rangle\langle 24\rangle[23][24]}{\langle 34\rangle^2[12]^2}$$

$$+ \frac{\langle 23\rangle[13][24][34]}{\langle 34\rangle[12]^2[14]} - \frac{\langle 23\rangle[23][34]}{\langle 34\rangle[12]^2}. \tag{G.2}$$

Our model reduces this 17 term amplitude to an equivalent rewriting of the $n=4$ version of Eq. (1):

$$\overline{\mathcal{M}} = - \frac{\langle 12\rangle[34]^2}{\langle 34\rangle[14][23]}.$$

• Four gravitons: $\mathcal{M}(h^-h^-h^+h^+)$

$$\mathcal{M} = \frac{\langle 12\rangle^4[13][24][34]}{\langle 23\rangle\langle 34\rangle^2[12][23]} - \frac{\langle 12\rangle^4[14][34]}{\langle 23\rangle\langle 34\rangle^2[12]} - \frac{\langle 12\rangle^4[13]^2[24]^2}{\langle 23\rangle\langle 34\rangle^2[12]^2[23]} + \frac{2\langle 12\rangle^4[13][14][24]}{\langle 23\rangle\langle 34\rangle^2[12]^2}$$

$$- \frac{\langle 12\rangle^4[14]^2[23]}{\langle 23\rangle\langle 34\rangle^2[12]^2} + \frac{\langle 12\rangle^4\langle 14\rangle[13]^2[24]^2}{\langle 13\rangle\langle 24\rangle\langle 34\rangle^2[12]^2[23]} - \frac{2\langle 12\rangle^4\langle 14\rangle[13][14][24]}{\langle 13\rangle\langle 24\rangle\langle 34\rangle^2[12]^2} + \frac{\langle 12\rangle^4\langle 14\rangle[14]^2[23]}{\langle 13\rangle\langle 24\rangle\langle 34\rangle^2[12]^2}$$

$$+ \frac{\langle 12\rangle^3\langle 13\rangle[13]^2[24][34]}{\langle 23\rangle\langle 34\rangle^2[12]^2[23]} - \frac{\langle 12\rangle^3\langle 13\rangle[13][14][34]}{\langle 23\rangle\langle 34\rangle^2[12]^2} + \frac{\langle 12\rangle^3\langle 14\rangle[13]^2[24][34]}{\langle 24\rangle\langle 34\rangle^2[12]^2[23]}$$

$$- \frac{\langle 12\rangle^3\langle 14\rangle[13][14][34]}{\langle 24\rangle\langle 34\rangle^2[12]^2} - \frac{\langle 12\rangle^3\langle 14\rangle[13]^3[24]^2}{\langle 24\rangle\langle 34\rangle^2[12]^3[23]} + \frac{2\langle 12\rangle^3\langle 14\rangle[13]^2[14][24]}{\langle 24\rangle\langle 34\rangle^2[12]^3}$$

$$- \frac{\langle 12\rangle^3\langle 14\rangle[13][14]^2[23]}{\langle 24\rangle\langle 34\rangle^2[12]^3} + \frac{\langle 12\rangle^3[34]^2}{\langle 34\rangle^2[12]} + \frac{\langle 12\rangle^3[13][24][34]}{\langle 34\rangle^2[12]^2}$$

$$- \frac{\langle 12\rangle^3 [14][23][34]}{\langle 34\rangle^2 [12]^2} - \frac{\langle 12\rangle^3 [13]^2 [24]^2}{\langle 34\rangle^2 [12]^3} + \frac{2\langle 12\rangle^3 [13][14][23][24]}{\langle 34\rangle^2 [12]^3}$$

$$- \frac{\langle 12\rangle^3 [14]^2 [23]^2}{\langle 34\rangle^2 [12]^3} + \frac{\langle 12\rangle^3 \langle 14\rangle^2 \langle 23\rangle [13]^3 [24]^2}{\langle 13\rangle \langle 24\rangle^2 \langle 34\rangle^2 [12]^3 [23]} - \frac{2\langle 12\rangle^3 \langle 14\rangle^2 \langle 23\rangle [13]^2 [14][24]}{\langle 13\rangle \langle 24\rangle^2 \langle 34\rangle^2 [12]^3}$$

$$+ \frac{\langle 12\rangle^3 \langle 14\rangle^2 \langle 23\rangle [13][14]^2 [23]}{\langle 13\rangle \langle 24\rangle^2 \langle 34\rangle^2 [12]^3} + \frac{\langle 12\rangle^3 \langle 14\rangle \langle 23\rangle [13][24][34]}{\langle 13\rangle \langle 24\rangle \langle 34\rangle^2 [12]^2} - \frac{\langle 12\rangle^3 \langle 14\rangle \langle 23\rangle [14][23][34]}{\langle 13\rangle \langle 24\rangle \langle 34\rangle^2 [12]^2}$$

$$+ \frac{\langle 12\rangle^3 \langle 14\rangle \langle 23\rangle [13]^2 [24]^2}{\langle 13\rangle \langle 24\rangle \langle 34\rangle^2 [12]^3} - \frac{2\langle 12\rangle^3 \langle 14\rangle \langle 23\rangle [13][14][23][24]}{\langle 13\rangle \langle 24\rangle \langle 34\rangle^2 [12]^3} + \frac{\langle 12\rangle^3 \langle 14\rangle \langle 23\rangle [14]^2 [23]^2}{\langle 13\rangle \langle 24\rangle \langle 34\rangle^2 [12]^3}$$

$$+ \frac{\langle 12\rangle^2 \langle 13\rangle \langle 14\rangle [13]^3 [24][34]}{\langle 24\rangle \langle 34\rangle^2 [12]^3 [23]} - \frac{\langle 12\rangle^2 \langle 13\rangle \langle 14\rangle [13]^2 [14][34]}{\langle 24\rangle \langle 34\rangle^2 [12]^3} + \frac{2\langle 12\rangle^2 \langle 13\rangle [13][34]^2}{\langle 34\rangle^2 [12]^2}$$

$$+ \frac{2\langle 12\rangle^2 \langle 14\rangle \langle 23\rangle [13]^2 [24][34]}{\langle 24\rangle \langle 34\rangle^2 [12]^3} - \frac{2\langle 12\rangle^2 \langle 14\rangle \langle 23\rangle [13][14][23][34]}{\langle 24\rangle \langle 34\rangle^2 [12]^3} + \frac{2\langle 12\rangle^2 \langle 23\rangle [23][34]^2}{\langle 34\rangle^2 [12]^2}$$

$$+ \frac{\langle 12\rangle^2 \langle 14\rangle \langle 23\rangle^2 [13][23][24][34]}{\langle 13\rangle \langle 24\rangle \langle 34\rangle^2 [12]^3} - \frac{\langle 12\rangle^2 \langle 14\rangle \langle 23\rangle^2 [14][23]^2 [34]}{\langle 13\rangle \langle 24\rangle \langle 34\rangle^2 [12]^3}$$

$$+ \frac{\langle 12\rangle \langle 13\rangle^2 [13]^2 [34]^2}{\langle 34\rangle^2 [12]^3} + \frac{2\langle 12\rangle \langle 13\rangle \langle 23\rangle [13][23][34]^2}{\langle 34\rangle^2 [12]^3} + \frac{\langle 12\rangle \langle 23\rangle^2 [23]^2 [34]^2}{\langle 34\rangle^2 [12]^3} .$$

Our model reduces this 40 term amplitude to:

$$\overline{\mathcal{M}} = - \frac{\langle 12\rangle^5 [34]^2}{\langle 13\rangle \langle 23\rangle \langle 24\rangle \langle 34\rangle [23]} .$$

• Five gluons: $\mathcal{M}(g^- g^- g^+ g^+ g^+)$

$$\mathcal{M} = \frac{\langle 12\rangle^3 [13]}{\langle 23\rangle \langle 24\rangle \langle 35\rangle \langle 45\rangle [23]} + \frac{\langle 12\rangle^3 [14][25]}{\langle 13\rangle \langle 24\rangle \langle 35\rangle \langle 45\rangle [12][45]} - \frac{\langle 12\rangle^3 [15][24]}{\langle 13\rangle \langle 24\rangle \langle 35\rangle \langle 45\rangle [12][45]}$$

$$- \frac{\langle 12\rangle^2 [13][34]}{\langle 24\rangle \langle 35\rangle \langle 45\rangle [14][23]} + \frac{\langle 12\rangle^2 [13]}{\langle 24\rangle \langle 35\rangle \langle 45\rangle [12]} - \frac{\langle 12\rangle^2 [13][24][35]}{\langle 24\rangle \langle 35\rangle \langle 45\rangle [12][23][45]}$$

$$+ \frac{\langle 12\rangle^2 [13][25][34]}{\langle 24\rangle \langle 35\rangle \langle 45\rangle [12][23][45]} + \frac{\langle 12\rangle^2 [14][35]}{\langle 24\rangle \langle 35\rangle \langle 45\rangle [12][45]} - \frac{\langle 12\rangle^2 [15][34]}{\langle 24\rangle \langle 35\rangle \langle 45\rangle [12][45]}$$

$$- \frac{\langle 12\rangle^2 [13][45]}{\langle 23\rangle \langle 24\rangle \langle 35\rangle [12][23]} - \frac{\langle 12\rangle^2 \langle 23\rangle [13][45]}{\langle 15\rangle \langle 24\rangle \langle 34\rangle \langle 35\rangle [14][15]} - \frac{\langle 12\rangle^2 [45]}{\langle 15\rangle \langle 34\rangle \langle 35\rangle [15]}$$

$$- \frac{\langle 12\rangle^2 [13][34][45]}{\langle 15\rangle \langle 24\rangle \langle 35\rangle [14][15][23]} + \frac{\langle 12\rangle^2 [13][45]}{\langle 15\rangle \langle 24\rangle \langle 35\rangle [12][15]} + \frac{\langle 12\rangle^2 [14][45]}{\langle 15\rangle \langle 23\rangle \langle 35\rangle [12][15]}$$

$$+ \frac{\langle 12\rangle^2 \langle 34\rangle [13][34][45]}{\langle 15\rangle \langle 23\rangle \langle 24\rangle \langle 35\rangle [12][15][23]} - \frac{\langle 12\rangle^2 \langle 15\rangle [15]}{\langle 13\rangle \langle 24\rangle \langle 35\rangle \langle 45\rangle [12]} + \frac{\langle 12\rangle^2 \langle 23\rangle [23]}{\langle 13\rangle \langle 24\rangle \langle 35\rangle \langle 45\rangle [12]}$$

$$- \frac{\langle 12\rangle^2 \langle 23\rangle [25][34]}{\langle 13\rangle \langle 24\rangle \langle 35\rangle \langle 45\rangle [12][45]} + \frac{\langle 12\rangle^2 \langle 23\rangle [15][24][34]}{\langle 13\rangle \langle 24\rangle \langle 35\rangle \langle 45\rangle [12][14][45]}$$

$$+ \frac{\langle 12\rangle^2 \langle 23\rangle [14][23][25]}{\langle 13\rangle \langle 24\rangle \langle 35\rangle \langle 45\rangle [12]^2 [45]} - \frac{\langle 12\rangle^2 \langle 23\rangle [15][23][24]}{\langle 13\rangle \langle 24\rangle \langle 35\rangle \langle 45\rangle [12]^2 [45]} - \frac{\langle 12\rangle^2 \langle 23\rangle [25]}{\langle 13\rangle \langle 24\rangle \langle 34\rangle \langle 35\rangle [12]}$$

$$+ \frac{\langle 12\rangle^2 \langle 23\rangle [15][24]}{\langle 13\rangle \langle 24\rangle \langle 34\rangle \langle 35\rangle [12][14]} + \frac{\langle 12\rangle^2 [24]}{\langle 13\rangle \langle 35\rangle \langle 45\rangle [12]} + \frac{\langle 12\rangle^2 \langle 34\rangle [13][24][34]}{\langle 13\rangle \langle 24\rangle \langle 35\rangle \langle 45\rangle [12][14][23]}$$

$$- \frac{\langle 12\rangle^2 [45]}{\langle 13\rangle \langle 24\rangle \langle 35\rangle [12]} + \frac{\langle 12\rangle^2 \langle 23\rangle [13][24][45]}{\langle 13\rangle \langle 15\rangle \langle 24\rangle \langle 35\rangle [12][14][15]} + \frac{\langle 12\rangle^2 [24][45]}{\langle 13\rangle \langle 15\rangle \langle 35\rangle [12][15]}$$

$$+ \frac{\langle 12\rangle^2 \langle 34\rangle [13][24][34][45]}{\langle 13\rangle \langle 15\rangle \langle 24\rangle \langle 35\rangle [12][14][15][23]} - \frac{\langle 12\rangle \langle 13\rangle [13][34][35]}{\langle 24\rangle \langle 35\rangle \langle 45\rangle [12][23][45]}$$

$$+ \frac{\langle 12\rangle \langle 13\rangle [13][15][34]^2}{\langle 24\rangle \langle 35\rangle \langle 45\rangle [12][14][23][45]} + \frac{\langle 12\rangle \langle 13\rangle \langle 23\rangle [13][34]}{\langle 15\rangle \langle 24\rangle \langle 34\rangle \langle 35\rangle [12][14]} + \frac{\langle 12\rangle \langle 13\rangle [34]}{\langle 15\rangle \langle 34\rangle \langle 35\rangle [12]}$$

$$+ \frac{\langle 12\rangle \langle 13\rangle [13][34]^2}{\langle 15\rangle \langle 24\rangle \langle 35\rangle [12][14][23]} + \frac{\langle 12\rangle \langle 14\rangle \langle 23\rangle [13][45]}{\langle 15\rangle \langle 24\rangle \langle 34\rangle \langle 35\rangle [12][15]} + \frac{\langle 12\rangle \langle 14\rangle [14][45]}{\langle 15\rangle \langle 34\rangle \langle 35\rangle [12][15]}$$

$$+ \frac{\langle 12 \rangle \langle 14 \rangle [13][34][45]}{\langle 15 \rangle \langle 24 \rangle \langle 35 \rangle [12][15][23]} - \frac{\langle 12 \rangle \langle 15 \rangle [13][15][34]}{\langle 24 \rangle \langle 35 \rangle \langle 45 \rangle [12][14][23]} + \frac{\langle 12 \rangle \langle 23 \rangle [13][24][34][35]}{\langle 24 \rangle \langle 35 \rangle \langle 45 \rangle [12][14][23][45]}$$

$$- \frac{\langle 12 \rangle \langle 23 \rangle [34][35]}{\langle 24 \rangle \langle 35 \rangle \langle 45 \rangle [12][45]} + \frac{\langle 12 \rangle \langle 23 \rangle [15][34]^2}{\langle 24 \rangle \langle 35 \rangle \langle 45 \rangle [12][14][45]} - \frac{\langle 12 \rangle \langle 23 \rangle [13][24][35]}{\langle 24 \rangle \langle 35 \rangle \langle 45 \rangle [12]^2 [45]}$$

$$+ \frac{\langle 12 \rangle \langle 23 \rangle [14][23][35]}{\langle 24 \rangle \langle 35 \rangle \langle 45 \rangle [12]^2 [45]} - \frac{\langle 12 \rangle \langle 23 \rangle [35]}{\langle 24 \rangle \langle 34 \rangle \langle 35 \rangle [12]} + \frac{\langle 12 \rangle \langle 23 \rangle [15][34]}{\langle 24 \rangle \langle 34 \rangle \langle 35 \rangle [12][14]} + \frac{\langle 12 \rangle [34]}{\langle 35 \rangle \langle 45 \rangle [12]}$$

$$+ \frac{\langle 12 \rangle \langle 34 \rangle [13][34]^2}{\langle 24 \rangle \langle 35 \rangle \langle 45 \rangle [12][14][23]} - \frac{\langle 12 \rangle \langle 23 \rangle^2 [13][23][45]}{\langle 15 \rangle \langle 24 \rangle \langle 34 \rangle \langle 35 \rangle [12][14][15]}$$

$$+ \frac{\langle 12 \rangle \langle 23 \rangle^2 [13][24][35]}{\langle 15 \rangle \langle 24 \rangle \langle 34 \rangle \langle 35 \rangle [12][14][15]} - \frac{\langle 12 \rangle \langle 23 \rangle [23][45]}{\langle 15 \rangle \langle 34 \rangle \langle 35 \rangle [12][15]} + \frac{\langle 12 \rangle \langle 23 \rangle [24][35]}{\langle 15 \rangle \langle 34 \rangle \langle 35 \rangle [12][15]}$$

$$+ \frac{\langle 12 \rangle \langle 23 \rangle [13][24][34][35]}{\langle 15 \rangle \langle 24 \rangle \langle 35 \rangle [12][14][15][23]} + \frac{\langle 12 \rangle \langle 23 \rangle \langle 45 \rangle [13][45]^2}{\langle 15 \rangle \langle 24 \rangle \langle 34 \rangle \langle 35 \rangle [12][14][15]} + \frac{\langle 12 \rangle [34][45]}{\langle 15 \rangle \langle 35 \rangle [12][15]}$$

$$+ \frac{\langle 12 \rangle \langle 45 \rangle [45]^2}{\langle 15 \rangle \langle 34 \rangle \langle 35 \rangle [12][15]} + \frac{\langle 12 \rangle \langle 34 \rangle [13][34]^2 [45]}{\langle 15 \rangle \langle 24 \rangle \langle 35 \rangle [12][14][15][23]}$$

$$+ \frac{\langle 12 \rangle \langle 45 \rangle [13][34][45]^2}{\langle 15 \rangle \langle 24 \rangle \langle 35 \rangle [12][14][15][23]} + \frac{\langle 12 \rangle \langle 14 \rangle \langle 23 \rangle [13][24][34]}{\langle 13 \rangle \langle 24 \rangle \langle 35 \rangle \langle 45 \rangle [12][14][23]} - \frac{\langle 12 \rangle \langle 14 \rangle \langle 23 \rangle [34]}{\langle 13 \rangle \langle 24 \rangle \langle 35 \rangle \langle 45 \rangle [12]}$$

$$- \frac{\langle 12 \rangle \langle 14 \rangle \langle 23 \rangle [45]}{\langle 13 \rangle \langle 24 \rangle \langle 34 \rangle \langle 35 \rangle [12]} + \frac{\langle 12 \rangle \langle 14 \rangle \langle 23 \rangle [14][25]}{\langle 13 \rangle \langle 24 \rangle \langle 34 \rangle \langle 35 \rangle [12]^2} - \frac{\langle 12 \rangle \langle 14 \rangle \langle 23 \rangle [15][24]}{\langle 13 \rangle \langle 24 \rangle \langle 34 \rangle \langle 35 \rangle [12]^2}$$

$$+ \frac{\langle 12 \rangle \langle 14 \rangle \langle 23 \rangle^2 [13][24][45]}{\langle 13 \rangle \langle 15 \rangle \langle 24 \rangle \langle 34 \rangle \langle 35 \rangle [12][14][15]} + \frac{\langle 12 \rangle \langle 14 \rangle \langle 23 \rangle [24][45]}{\langle 13 \rangle \langle 15 \rangle \langle 34 \rangle \langle 35 \rangle [12][15]}$$

$$+ \frac{\langle 12 \rangle \langle 14 \rangle \langle 23 \rangle [13][24][34][45]}{\langle 13 \rangle \langle 15 \rangle \langle 24 \rangle \langle 35 \rangle [12][14][15][23]} - \frac{\langle 12 \rangle \langle 15 \rangle \langle 23 \rangle [15][23]}{\langle 13 \rangle \langle 24 \rangle \langle 35 \rangle \langle 45 \rangle [12]^2} + \frac{\langle 12 \rangle \langle 23 \rangle^2 [23]^2}{\langle 13 \rangle \langle 24 \rangle \langle 35 \rangle \langle 45 \rangle [12]^2}$$

$$- \frac{\langle 12 \rangle \langle 23 \rangle^2 [23][25][34]}{\langle 13 \rangle \langle 24 \rangle \langle 35 \rangle \langle 45 \rangle [12]^2 [45]} + \frac{\langle 12 \rangle \langle 23 \rangle^2 [15][23][24][34]}{\langle 13 \rangle \langle 24 \rangle \langle 35 \rangle \langle 45 \rangle [12]^2 [14][45]} - \frac{\langle 12 \rangle \langle 23 \rangle^2 [23][25]}{\langle 13 \rangle \langle 24 \rangle \langle 34 \rangle \langle 35 \rangle [12]^2}$$

$$+ \frac{\langle 12 \rangle \langle 23 \rangle^2 [15][23][24]}{\langle 13 \rangle \langle 24 \rangle \langle 34 \rangle \langle 35 \rangle [12]^2 [14]} + \frac{\langle 12 \rangle \langle 23 \rangle [23][24]}{\langle 13 \rangle \langle 35 \rangle \langle 45 \rangle [12]^2} + \frac{\langle 12 \rangle \langle 23 \rangle \langle 34 \rangle [13][24][34]}{\langle 13 \rangle \langle 24 \rangle \langle 35 \rangle \langle 45 \rangle [12]^2 [14]}$$

$$+ \frac{\langle 12 \rangle \langle 23 \rangle [13][24][45]}{\langle 13 \rangle \langle 24 \rangle \langle 35 \rangle [12]^2 [14]} - \frac{\langle 12 \rangle \langle 23 \rangle [23][45]}{\langle 13 \rangle \langle 24 \rangle \langle 35 \rangle [12]^2} + \frac{\langle 12 \rangle \langle 23 \rangle [25][34]}{\langle 13 \rangle \langle 24 \rangle \langle 35 \rangle [12]^2} - \frac{\langle 12 \rangle \langle 23 \rangle [15][24][34]}{\langle 13 \rangle \langle 24 \rangle \langle 35 \rangle [12]^2 [14]}$$

$$+ \frac{\langle 12 \rangle \langle 23 \rangle \langle 45 \rangle [25][45]}{\langle 13 \rangle \langle 24 \rangle \langle 34 \rangle \langle 35 \rangle [12]^2} - \frac{\langle 12 \rangle \langle 23 \rangle \langle 45 \rangle [15][24][45]}{\langle 13 \rangle \langle 24 \rangle \langle 34 \rangle \langle 35 \rangle [12]^2 [14]} + \frac{\langle 13 \rangle \langle 23 \rangle [13][34]^2 [35]}{\langle 24 \rangle \langle 35 \rangle \langle 45 \rangle [12][14][23][45]}$$

$$+ \frac{\langle 13 \rangle \langle 23 \rangle^2 [13][34][35]}{\langle 15 \rangle \langle 24 \rangle \langle 34 \rangle \langle 35 \rangle [12][14][15]} + \frac{\langle 13 \rangle \langle 23 \rangle [34][35]}{\langle 15 \rangle \langle 34 \rangle \langle 35 \rangle [12][15]} + \frac{\langle 13 \rangle \langle 23 \rangle [13][34]^2 [35]}{\langle 15 \rangle \langle 24 \rangle \langle 35 \rangle [12][14][15][23]}$$

$$+ \frac{\langle 14 \rangle \langle 23 \rangle [13][34]^2}{\langle 24 \rangle \langle 35 \rangle \langle 45 \rangle [12][14][23]} + \frac{\langle 14 \rangle \langle 23 \rangle^2 [13][34][45]}{\langle 15 \rangle \langle 24 \rangle \langle 34 \rangle \langle 35 \rangle [12][14][15]} + \frac{\langle 14 \rangle \langle 23 \rangle [34][45]}{\langle 15 \rangle \langle 34 \rangle \langle 35 \rangle [12][15]}$$

$$+ \frac{\langle 14 \rangle \langle 23 \rangle [13][34]^2 [45]}{\langle 15 \rangle \langle 24 \rangle \langle 35 \rangle [12][14][15][23]} - \frac{\langle 15 \rangle \langle 23 \rangle [15][35]}{\langle 24 \rangle \langle 34 \rangle \langle 35 \rangle [12]^2} + \frac{\langle 23 \rangle^2 [13][24][34][35]}{\langle 24 \rangle \langle 35 \rangle \langle 45 \rangle [12]^2 [14][45]}$$

$$- \frac{\langle 23 \rangle^2 [23][34][35]}{\langle 24 \rangle \langle 35 \rangle \langle 45 \rangle [12]^2 [45]} + \frac{\langle 23 \rangle^2 [13][24][35]}{\langle 24 \rangle \langle 34 \rangle \langle 35 \rangle [12]^2 [14]} + \frac{\langle 23 \rangle [24][35]}{\langle 34 \rangle \langle 35 \rangle [12]^2} + \frac{\langle 23 \rangle [34][35]}{\langle 24 \rangle \langle 35 \rangle [12]^2}$$

$$- \frac{\langle 14 \rangle \langle 15 \rangle \langle 23 \rangle [15][45]}{\langle 13 \rangle \langle 24 \rangle \langle 34 \rangle \langle 35 \rangle [12]^2} + \frac{\langle 14 \rangle \langle 23 \rangle^2 [13][24][34]}{\langle 13 \rangle \langle 24 \rangle \langle 35 \rangle \langle 45 \rangle [12]^2 [14]} - \frac{\langle 14 \rangle \langle 23 \rangle^2 [23][34]}{\langle 13 \rangle \langle 24 \rangle \langle 35 \rangle \langle 45 \rangle [12]^2}$$

$$+ \frac{\langle 14 \rangle \langle 23 \rangle^2 [13][24][45]}{\langle 13 \rangle \langle 24 \rangle \langle 34 \rangle \langle 35 \rangle [12]^2 [14]} + \frac{\langle 14 \rangle \langle 23 \rangle [24][45]}{\langle 13 \rangle \langle 34 \rangle \langle 35 \rangle [12]^2} + \frac{\langle 14 \rangle \langle 23 \rangle [34][45]}{\langle 13 \rangle \langle 24 \rangle \langle 35 \rangle [12]^2} \, .$$

Our model reduces this 100 term amplitude to the $n = 5$ version of Eq. (1):

$$\overline{\mathcal{M}} = - \frac{\langle 12 \rangle^3}{\langle 15 \rangle \langle 23 \rangle \langle 34 \rangle \langle 45 \rangle} \, .$$

• Three scalars and two same-helicity gravitons: $\mathcal{M}(\phi\phi\phi h^+ h^+)$

$$
\begin{aligned}
\mathcal{M} =\ & \frac{\langle 12\rangle\langle 13\rangle^2\langle 15\rangle\langle 25\rangle\langle 34\rangle [12][15]^2[24][34]}{\langle 14\rangle\langle 23\rangle\langle 35\rangle^3\langle 45\rangle^3[23][35][45]} + \frac{\langle 12\rangle\langle 13\rangle^2\langle 15\rangle\langle 34\rangle [12][15]^2[34]^2}{\langle 14\rangle\langle 23\rangle\langle 35\rangle^2\langle 45\rangle^3[23][35][45]} \\
& + \frac{\langle 12\rangle\langle 13\rangle^2 [12][15][34]}{\langle 14\rangle\langle 35\rangle^2\langle 45\rangle^2[35]} + \frac{\langle 12\rangle\langle 13\rangle^2\langle 25\rangle\langle 34\rangle [12][15][24][34]}{\langle 14\rangle\langle 23\rangle\langle 35\rangle^3\langle 45\rangle^2[23][35]} \\
& + \frac{\langle 12\rangle\langle 13\rangle^2\langle 34\rangle [12][15][34]^2}{\langle 14\rangle\langle 23\rangle\langle 35\rangle^2\langle 45\rangle^2[23][35]} - \frac{\langle 12\rangle\langle 13\rangle [12][15][24][34]}{\langle 35\rangle\langle 45\rangle^3[23][45]} - \frac{\langle 12\rangle\langle 13\rangle [12][15][34]^2}{\langle 25\rangle\langle 45\rangle^3[23][45]} \\
& - \frac{\langle 12\rangle\langle 13\rangle\langle 23\rangle [12][34]}{\langle 15\rangle\langle 25\rangle\langle 34\rangle\langle 45\rangle^2} - \frac{\langle 12\rangle\langle 13\rangle [12][24][34]}{\langle 15\rangle\langle 35\rangle\langle 45\rangle^2[23]} - \frac{\langle 12\rangle\langle 13\rangle [12][34]^2}{\langle 15\rangle\langle 25\rangle\langle 45\rangle^2[23]} \\
& - \frac{\langle 12\rangle\langle 13\rangle\langle 15\rangle^2\langle 34\rangle [12][14][15][25][34]}{\langle 14\rangle\langle 35\rangle^3\langle 45\rangle^3[23][35][45]} - \frac{\langle 12\rangle\langle 13\rangle\langle 15\rangle^2\langle 34\rangle [12][15]^2[34]}{\langle 14\rangle\langle 23\rangle\langle 35\rangle^2\langle 45\rangle^3[23][35]} \\
& + \frac{\langle 12\rangle\langle 13\rangle\langle 15\rangle\langle 25\rangle\langle 34\rangle [12][15][24][25][34]}{\langle 14\rangle\langle 35\rangle^3\langle 45\rangle^3[23][35][45]} + \frac{\langle 12\rangle\langle 13\rangle\langle 15\rangle\langle 34\rangle [12][15][34]}{\langle 14\rangle\langle 35\rangle^2\langle 45\rangle^3[35]} \\
& + \frac{\langle 12\rangle\langle 13\rangle\langle 15\rangle\langle 34\rangle [12][15][24][34]}{\langle 14\rangle\langle 35\rangle^2\langle 45\rangle^3[23][45]} + \frac{\langle 12\rangle\langle 13\rangle\langle 15\rangle\langle 34\rangle [12][15][25][34]^2}{\langle 14\rangle\langle 35\rangle^2\langle 45\rangle^3[23][35][45]} \\
& + \frac{\langle 12\rangle\langle 13\rangle\langle 15\rangle\langle 34\rangle [12][15][34]^2}{\langle 14\rangle\langle 25\rangle\langle 35\rangle\langle 45\rangle^3[23][45]} - \frac{\langle 12\rangle\langle 13\rangle\langle 15\rangle\langle 25\rangle\langle 34\rangle^2 [12][15][24][34]}{\langle 14\rangle\langle 23\rangle\langle 35\rangle^3\langle 45\rangle^3[23][35]} \\
& - \frac{\langle 12\rangle\langle 13\rangle\langle 15\rangle\langle 34\rangle^2 [12][15][34]^2}{\langle 14\rangle\langle 23\rangle\langle 35\rangle^2\langle 45\rangle^3[23][35]} - \frac{\langle 12\rangle\langle 13\rangle\langle 15\rangle\langle 34\rangle [12][15][34][45]}{\langle 14\rangle\langle 23\rangle\langle 35\rangle^2\langle 45\rangle^2[23][35]} \\
& + \frac{\langle 12\rangle\langle 13\rangle\langle 23\rangle [12][25][34]}{\langle 14\rangle\langle 35\rangle^2\langle 45\rangle^2[35]} + \frac{\langle 12\rangle\langle 13\rangle\langle 23\rangle [12][34]}{\langle 14\rangle\langle 25\rangle\langle 35\rangle\langle 45\rangle^2} + \frac{\langle 12\rangle\langle 13\rangle\langle 25\rangle\langle 34\rangle [12][24][25][34]}{\langle 14\rangle\langle 35\rangle^3\langle 45\rangle^2[23][35]} \\
& - \frac{\langle 12\rangle\langle 13\rangle\langle 34\rangle [12][34][45]}{\langle 14\rangle\langle 35\rangle^2\langle 45\rangle^2[35]} + \frac{\langle 12\rangle\langle 13\rangle\langle 34\rangle [12][24][34]}{\langle 14\rangle\langle 35\rangle^2\langle 45\rangle^2[23]} + \frac{\langle 12\rangle\langle 13\rangle\langle 34\rangle [12][25][34]^2}{\langle 14\rangle\langle 35\rangle^2\langle 45\rangle^2[23][35]} \\
& + \frac{\langle 12\rangle\langle 13\rangle\langle 34\rangle [12][34]^2}{\langle 14\rangle\langle 25\rangle\langle 35\rangle\langle 45\rangle^2[23]} - \frac{\langle 12\rangle\langle 13\rangle\langle 25\rangle\langle 34\rangle^2 [12][24][34][45]}{\langle 14\rangle\langle 23\rangle\langle 35\rangle^3\langle 45\rangle^2[23][35]} \\
& - \frac{\langle 12\rangle\langle 13\rangle\langle 34\rangle^2 [12][34]^2[45]}{\langle 14\rangle\langle 23\rangle\langle 35\rangle^2\langle 45\rangle^2[23][35]} + \frac{\langle 12\rangle\langle 15\rangle\langle 23\rangle [12][14][25][34]}{\langle 25\rangle\langle 35\rangle\langle 45\rangle^3[23][45]} + \frac{\langle 12\rangle\langle 15\rangle [12][15][34]}{\langle 25\rangle\langle 45\rangle^3[23]} \\
& - \frac{\langle 12\rangle\langle 23\rangle [12][34]}{\langle 25\rangle\langle 45\rangle^3} + \frac{\langle 12\rangle [12][34][45]}{\langle 25\rangle\langle 45\rangle^2[23]} - \frac{\langle 12\rangle\langle 15\rangle^2\langle 23\rangle\langle 34\rangle [12][14][25]^2[34]}{\langle 14\rangle\langle 35\rangle^3\langle 45\rangle^3[23][35][45]} \\
& - \frac{\langle 12\rangle\langle 15\rangle^2\langle 23\rangle\langle 34\rangle [12][14][25][34]}{\langle 14\rangle\langle 25\rangle\langle 35\rangle^2\langle 45\rangle^3[23][45]} + \frac{\langle 12\rangle\langle 15\rangle^2\langle 34\rangle^2 [12][14][25][34]}{\langle 14\rangle\langle 35\rangle^3\langle 45\rangle^3[23][35]} \\
& - \frac{\langle 12\rangle\langle 15\rangle^2\langle 34\rangle [12][15][25][34]}{\langle 14\rangle\langle 35\rangle^2\langle 45\rangle^3[23][35]} - \frac{\langle 12\rangle\langle 15\rangle^2\langle 34\rangle [12][15][34]}{\langle 14\rangle\langle 25\rangle\langle 35\rangle\langle 45\rangle^3[23]} \\
& + \frac{\langle 12\rangle\langle 15\rangle^2\langle 34\rangle^2 [12][15][34][45]}{\langle 14\rangle\langle 23\rangle\langle 35\rangle^2\langle 45\rangle^3[23][35]} + \frac{\langle 12\rangle\langle 15\rangle\langle 23\rangle\langle 34\rangle [12][25][34]}{\langle 14\rangle\langle 35\rangle^2\langle 45\rangle^3[35]} \\
& + \frac{\langle 12\rangle\langle 15\rangle\langle 23\rangle\langle 34\rangle [12][34]}{\langle 14\rangle\langle 25\rangle\langle 35\rangle\langle 45\rangle^3} - \frac{\langle 12\rangle\langle 15\rangle\langle 34\rangle^2 [12][34][45]}{\langle 14\rangle\langle 35\rangle^2\langle 45\rangle^3[35]} - \frac{\langle 12\rangle\langle 15\rangle\langle 34\rangle [12][25][34][45]}{\langle 14\rangle\langle 35\rangle^2\langle 45\rangle^2[23][35]} \\
& - \frac{\langle 12\rangle\langle 15\rangle\langle 34\rangle [12][34][45]}{\langle 14\rangle\langle 25\rangle\langle 35\rangle\langle 45\rangle^2[23]} + \frac{\langle 12\rangle\langle 15\rangle\langle 34\rangle^2 [12][34][45]^2}{\langle 14\rangle\langle 23\rangle\langle 35\rangle^2\langle 45\rangle^2[23][35]} \\
& + \frac{\langle 13\rangle^3\langle 15\rangle\langle 24\rangle [13][15]^2[24]^2}{\langle 14\rangle\langle 23\rangle\langle 35\rangle^2\langle 45\rangle^3[23][25][45]} + \frac{\langle 13\rangle^3\langle 15\rangle\langle 24\rangle [13][15]^2[24][34]}{\langle 14\rangle\langle 23\rangle\langle 25\rangle\langle 35\rangle\langle 45\rangle^3[23][25][45]} \\
& + \frac{\langle 13\rangle^3 [13][15][24]}{\langle 14\rangle\langle 35\rangle^2\langle 45\rangle^2[25]} + \frac{\langle 13\rangle^3\langle 24\rangle [13][15][24]^2}{\langle 14\rangle\langle 23\rangle\langle 35\rangle^2\langle 45\rangle^2[23][25]} + \frac{\langle 13\rangle^3\langle 24\rangle [13][15][24][34]}{\langle 14\rangle\langle 23\rangle\langle 25\rangle\langle 35\rangle\langle 45\rangle^2[23][25]} \\
& + \frac{\langle 13\rangle^2\langle 15\rangle\langle 24\rangle [14][15]^2[24]^2}{\langle 23\rangle\langle 35\rangle^2\langle 45\rangle^3[23][25][45]} + \frac{\langle 13\rangle^2\langle 15\rangle\langle 24\rangle [14][15]^2[24][34]}{\langle 23\rangle\langle 25\rangle\langle 35\rangle\langle 45\rangle^3[23][25][45]} \\
& + \frac{\langle 13\rangle^2\langle 15\rangle\langle 25\rangle\langle 34\rangle [14][15]^2[24][34]}{\langle 23\rangle\langle 35\rangle^3\langle 45\rangle^3[23][35][45]} + \frac{\langle 13\rangle^2\langle 15\rangle\langle 34\rangle [14][15]^2[34]^2}{\langle 23\rangle\langle 35\rangle^2\langle 45\rangle^3[23][35][45]} + \frac{\langle 13\rangle^2 [14][15][24]}{\langle 35\rangle^2\langle 45\rangle^2[25]}
\end{aligned}
$$

$$+ \frac{\langle 13\rangle^2 [14][15][34]}{\langle 35\rangle^2 \langle 45\rangle^2 [35]} + \frac{\langle 13\rangle^2 \langle 24\rangle [15]^2 [24]^2 [34]}{\langle 23\rangle \langle 35\rangle \langle 45\rangle^3 [23][25][45]}$$

$$+ \frac{\langle 13\rangle^2 \langle 24\rangle [14][15][24]^2}{\langle 23\rangle \langle 35\rangle^2 \langle 45\rangle^2 [23][25]} + \frac{\langle 13\rangle^2 \langle 24\rangle [15]^2 [24][34]^2}{\langle 23\rangle \langle 25\rangle \langle 45\rangle^3 [23][25][45]} + \frac{\langle 13\rangle^2 \langle 24\rangle [14][15][24][34]}{\langle 23\rangle \langle 25\rangle \langle 35\rangle \langle 45\rangle^2 [23][25]}$$

$$+ \frac{\langle 13\rangle^2 \langle 25\rangle^2 \langle 34\rangle [15]^2 [24]^2 [34]}{\langle 23\rangle \langle 35\rangle^3 \langle 45\rangle^3 [23][35][45]} - \frac{\langle 13\rangle^2 \langle 25\rangle^2 [15]^2 [24]^2}{\langle 23\rangle \langle 35\rangle^2 \langle 45\rangle^3 [23][45]} + \frac{\langle 13\rangle^2 \langle 25\rangle \langle 34\rangle [15]^2 [24][34]^2}{\langle 23\rangle \langle 35\rangle^2 \langle 45\rangle^3 [23][35][45]}$$

$$+ \frac{\langle 13\rangle^2 \langle 25\rangle \langle 34\rangle [14][15][24][34]}{\langle 23\rangle \langle 35\rangle^3 \langle 45\rangle^2 [23][35]} - \frac{2\langle 13\rangle^2 \langle 25\rangle [15]^2 [24][34]}{\langle 23\rangle \langle 35\rangle \langle 45\rangle^3 [23][45]}$$

$$+ \frac{\langle 13\rangle^2 \langle 34\rangle [14][15][34]^2}{\langle 23\rangle \langle 35\rangle^2 \langle 45\rangle^2 [23][35]} - \frac{\langle 13\rangle^2 [15]^2 [34]^2}{\langle 23\rangle \langle 45\rangle^3 [23][45]} + \frac{\langle 13\rangle^2 \langle 25\rangle [15][24][34]}{\langle 15\rangle \langle 35\rangle^2 \langle 45\rangle^2 [35]}$$

$$+ \frac{\langle 13\rangle^2 [15][24][34]}{\langle 15\rangle \langle 35\rangle \langle 45\rangle^2 [25]} - \frac{\langle 13\rangle^2 [15][34]}{\langle 15\rangle \langle 34\rangle \langle 45\rangle^2} - \frac{\langle 13\rangle^2 \langle 25\rangle^2 [15][24]}{\langle 15\rangle \langle 24\rangle \langle 35\rangle^2 \langle 45\rangle^2} + \frac{\langle 13\rangle^2 \langle 24\rangle [15][24]^2 [34]}{\langle 15\rangle \langle 23\rangle \langle 35\rangle \langle 45\rangle^2 [23][25]}$$

$$+ \frac{\langle 13\rangle^2 \langle 24\rangle [15][24][34]^2}{\langle 15\rangle \langle 23\rangle \langle 25\rangle \langle 45\rangle^2 [23][25]} + \frac{\langle 13\rangle^2 \langle 25\rangle^2 \langle 34\rangle [15][24]^2 [34]}{\langle 15\rangle \langle 23\rangle \langle 35\rangle^3 \langle 45\rangle^2 [23][35]} - \frac{\langle 13\rangle^2 \langle 25\rangle^2 [15][24]^2}{\langle 15\rangle \langle 23\rangle \langle 35\rangle^2 \langle 45\rangle^2 [23]}$$

$$+ \frac{\langle 13\rangle^2 \langle 25\rangle \langle 34\rangle [15][24][34]^2}{\langle 15\rangle \langle 23\rangle \langle 35\rangle^2 \langle 45\rangle^2 [23][35]} - \frac{2\langle 13\rangle^2 \langle 25\rangle [15][24][34]}{\langle 15\rangle \langle 23\rangle \langle 35\rangle \langle 45\rangle^2 [23]} - \frac{\langle 13\rangle^2 [15][34]^2}{\langle 15\rangle \langle 23\rangle \langle 45\rangle^2 [23]}$$

$$- \frac{\langle 13\rangle^2 \langle 15\rangle^2 \langle 24\rangle [13][14][15][24]}{\langle 14\rangle \langle 25\rangle \langle 35\rangle^2 \langle 45\rangle^3 [23][45]} - \frac{\langle 13\rangle^2 \langle 15\rangle^2 \langle 24\rangle [13][15]^2 [24]}{\langle 14\rangle \langle 23\rangle \langle 25\rangle \langle 35\rangle \langle 45\rangle^3 [23][25]}$$

$$+ \frac{\langle 13\rangle^2 \langle 15\rangle \langle 24\rangle [15]^2 [24]^2}{\langle 14\rangle \langle 35\rangle^2 \langle 45\rangle^3 [25][45]} + \frac{\langle 13\rangle^2 \langle 15\rangle \langle 24\rangle [13][15][24]}{\langle 14\rangle \langle 25\rangle \langle 35\rangle \langle 45\rangle^3 [25]} + \frac{\langle 13\rangle^2 \langle 15\rangle \langle 34\rangle [15]^2 [34]^2}{\langle 14\rangle \langle 35\rangle^2 \langle 45\rangle^3 [35][45]}$$

$$+ \frac{\langle 13\rangle^2 \langle 15\rangle [15]^2 [24]}{\langle 14\rangle \langle 35\rangle^2 \langle 45\rangle^2 [25]} + \frac{\langle 13\rangle^2 \langle 15\rangle [15]^2 [34]}{\langle 14\rangle \langle 35\rangle^2 \langle 45\rangle^2 [35]} - \frac{\langle 13\rangle^2 \langle 15\rangle \langle 24\rangle \langle 34\rangle [13][15][24]^2}{\langle 14\rangle \langle 23\rangle \langle 35\rangle^2 \langle 45\rangle^3 [23][25]}$$

$$- \frac{\langle 13\rangle^2 \langle 15\rangle \langle 24\rangle \langle 34\rangle [13][15][24][34]}{\langle 14\rangle \langle 23\rangle \langle 25\rangle \langle 35\rangle \langle 45\rangle^3 [23][25]} - \frac{\langle 13\rangle^2 \langle 15\rangle \langle 24\rangle [13][15][24][45]}{\langle 14\rangle \langle 23\rangle \langle 25\rangle \langle 35\rangle \langle 45\rangle^2 [23][25]}$$

$$- \frac{\langle 13\rangle^2 \langle 34\rangle [13][24][45]}{\langle 14\rangle \langle 35\rangle^2 \langle 45\rangle^2 [25]} - \frac{\langle 13\rangle^2 \langle 24\rangle \langle 34\rangle [13][24]^2 [45]}{\langle 14\rangle \langle 23\rangle \langle 35\rangle^2 \langle 45\rangle^2 [23][25]} - \frac{\langle 13\rangle^2 \langle 24\rangle \langle 34\rangle [13][24][34][45]}{\langle 14\rangle \langle 23\rangle \langle 25\rangle \langle 35\rangle \langle 45\rangle^2 [23][25]}$$

$$- \frac{\langle 13\rangle \langle 15\rangle^2 \langle 24\rangle [14]^2 [15][24]}{\langle 25\rangle \langle 35\rangle^2 \langle 45\rangle^3 [23][45]} - \frac{\langle 13\rangle \langle 15\rangle^2 \langle 34\rangle [14]^2 [15][25][34]}{\langle 35\rangle^3 \langle 45\rangle^3 [23][35][45]}$$

$$- \frac{\langle 13\rangle \langle 15\rangle^2 \langle 24\rangle [14][15]^2 [24]}{\langle 23\rangle \langle 25\rangle \langle 35\rangle \langle 45\rangle^3 [23][25]} - \frac{\langle 13\rangle \langle 15\rangle^2 \langle 34\rangle [14][15]^2 [34]}{\langle 23\rangle \langle 35\rangle^2 \langle 45\rangle^3 [23][35]} + \frac{\langle 13\rangle \langle 15\rangle \langle 24\rangle [14][15]^2 [24]^2}{\langle 35\rangle^2 \langle 45\rangle^3 [13][25][45]}$$

$$+ \frac{\langle 13\rangle \langle 15\rangle \langle 24\rangle [14][15][24]}{\langle 25\rangle \langle 35\rangle \langle 45\rangle^3 [25]} - \frac{\langle 13\rangle \langle 15\rangle \langle 24\rangle [14][15][24][34]}{\langle 25\rangle \langle 35\rangle \langle 45\rangle^3 [23][45]}$$

$$+ \frac{\langle 13\rangle \langle 15\rangle \langle 25\rangle [14][15][24][25]}{\langle 35\rangle^2 \langle 45\rangle^3 [23][45]} + \frac{\langle 13\rangle \langle 15\rangle \langle 34\rangle [14][15][34]}{\langle 35\rangle^2 \langle 45\rangle^3 [35]}$$

$$+ \frac{\langle 13\rangle \langle 15\rangle \langle 34\rangle [14][15][25][34]^2}{\langle 35\rangle^2 \langle 45\rangle^3 [23][35][45]} + \frac{\langle 13\rangle \langle 15\rangle [14][15][25][34]}{\langle 35\rangle \langle 45\rangle^3 [23][45]} + \frac{\langle 13\rangle \langle 15\rangle [14][15]^2 [24]}{\langle 35\rangle^2 \langle 45\rangle^2 [13][25]}$$

$$- \frac{\langle 13\rangle \langle 15\rangle \langle 24\rangle \langle 34\rangle [14][15][24]^2}{\langle 23\rangle \langle 35\rangle^2 \langle 45\rangle^3 [23][25]} - \frac{\langle 13\rangle \langle 15\rangle \langle 24\rangle \langle 34\rangle [14][15][24][34]}{\langle 23\rangle \langle 25\rangle \langle 35\rangle \langle 45\rangle^3 [23][25]}$$

$$- \frac{\langle 13\rangle \langle 15\rangle \langle 24\rangle [15]^2 [24][34]}{\langle 23\rangle \langle 25\rangle \langle 45\rangle^3 [23][25]} - \frac{\langle 13\rangle \langle 15\rangle \langle 24\rangle [14][15][24][45]}{\langle 23\rangle \langle 25\rangle \langle 35\rangle \langle 45\rangle^2 [23][25]}$$

$$- \frac{\langle 13\rangle \langle 15\rangle \langle 25\rangle \langle 34\rangle^2 [14][15][24][34]}{\langle 23\rangle \langle 35\rangle^3 \langle 45\rangle^3 [23][35]} - \frac{\langle 13\rangle \langle 15\rangle \langle 25\rangle \langle 34\rangle [15]^2 [24][34]}{\langle 23\rangle \langle 35\rangle^2 \langle 45\rangle^3 [23][35]}$$

$$+ \frac{\langle 13\rangle \langle 15\rangle \langle 25\rangle [15]^2 [24]}{\langle 23\rangle \langle 35\rangle \langle 45\rangle^3 [23]} - \frac{\langle 13\rangle \langle 15\rangle \langle 34\rangle^2 [14][15][34]^2}{\langle 23\rangle \langle 35\rangle^2 \langle 45\rangle^3 [23][35]} - \frac{\langle 13\rangle \langle 15\rangle \langle 34\rangle [14][15][34][45]}{\langle 23\rangle \langle 35\rangle^2 \langle 45\rangle^2 [23][35]}$$

$$+ \frac{\langle 13\rangle \langle 15\rangle [15]^2 [34]}{\langle 23\rangle \langle 45\rangle^3 [23]} + \frac{\langle 13\rangle \langle 23\rangle [14][25][34]}{\langle 35\rangle^2 \langle 45\rangle^2 [35]}$$

$$- \frac{\langle 13\rangle \langle 23\rangle [15][34]^2}{\langle 25\rangle \langle 45\rangle^3 [45]} - \frac{\langle 13\rangle \langle 23\rangle [15][34]}{\langle 25\rangle \langle 34\rangle \langle 45\rangle^2} + \frac{\langle 13\rangle \langle 24\rangle [15]^2 [24]^2 [34]}{\langle 35\rangle \langle 45\rangle^3 [13][25][45]}$$

$$+ \frac{\langle 13\rangle\langle 24\rangle[15][24][34]}{\langle 25\rangle\langle 45\rangle^3[25]} + \frac{\langle 13\rangle\langle 25\rangle^2\langle 34\rangle[15][24]^2[25][34]}{\langle 35\rangle^3\langle 45\rangle^3[23][35][45]} - \frac{\langle 13\rangle\langle 25\rangle^2[15]^2[24]^2}{\langle 35\rangle^2\langle 45\rangle^3[13][45]}$$

$$+ \frac{\langle 13\rangle\langle 25\rangle\langle 34\rangle[15][24][34]}{\langle 35\rangle^2\langle 45\rangle^3[35]} + \frac{\langle 13\rangle\langle 25\rangle\langle 34\rangle[15][24][25][34]^2}{\langle 35\rangle^2\langle 45\rangle^3[23][35][45]}$$

$$+ \frac{\langle 13\rangle\langle 25\rangle\langle 34\rangle[14][24][25][34]}{\langle 35\rangle^3\langle 45\rangle^2[23][35]} - \frac{\langle 13\rangle\langle 25\rangle[15][24]}{\langle 35\rangle\langle 45\rangle^3} - \frac{\langle 13\rangle\langle 25\rangle[15][24][25][34]}{\langle 35\rangle\langle 45\rangle^3[23][45]}$$

$$- \frac{\langle 13\rangle\langle 34\rangle[14][24][45]}{\langle 35\rangle^2\langle 45\rangle^2[25]} - \frac{\langle 13\rangle\langle 34\rangle[14][34][45]}{\langle 35\rangle^2\langle 45\rangle^2[35]} + \frac{\langle 13\rangle\langle 34\rangle[14][25][34]^2}{\langle 35\rangle^2\langle 45\rangle^2[23][35]} - \frac{\langle 13\rangle[15][34]}{\langle 45\rangle^3}$$

$$- \frac{\langle 13\rangle[15][25][34]^2}{\langle 45\rangle^3[23][45]} + \frac{\langle 13\rangle[15]^2[24][34]}{\langle 35\rangle\langle 45\rangle^2[13][25]} - \frac{\langle 13\rangle\langle 25\rangle^2[15]^2[24]}{\langle 24\rangle\langle 35\rangle^2\langle 45\rangle^2[13]} + \frac{\langle 13\rangle\langle 24\rangle\langle 25\rangle[15][24]^2}{\langle 23\rangle\langle 35\rangle\langle 45\rangle^3[23]}$$

$$- \frac{\langle 13\rangle\langle 24\rangle\langle 34\rangle[15][24]^2[34]}{\langle 23\rangle\langle 35\rangle\langle 45\rangle^3[23][25]} - \frac{\langle 13\rangle\langle 24\rangle\langle 34\rangle[14][24]^2[45]}{\langle 23\rangle\langle 35\rangle^2\langle 45\rangle^2[23][25]} + \frac{\langle 13\rangle\langle 24\rangle[15][24][34]}{\langle 23\rangle\langle 45\rangle^3[23]}$$

$$- \frac{\langle 13\rangle\langle 24\rangle\langle 34\rangle[15][24][34]^2}{\langle 23\rangle\langle 25\rangle\langle 45\rangle^3[23][25]} - \frac{\langle 13\rangle\langle 24\rangle\langle 34\rangle[14][24][34][45]}{\langle 23\rangle\langle 25\rangle\langle 35\rangle\langle 45\rangle^2[23][25]} - \frac{\langle 13\rangle\langle 24\rangle[15][24][34][45]}{\langle 23\rangle\langle 25\rangle\langle 45\rangle^2[23][25]}$$

$$- \frac{\langle 13\rangle\langle 25\rangle^2\langle 34\rangle^2[15][24]^2[34]}{\langle 23\rangle\langle 35\rangle^3\langle 45\rangle^3[23][35]} - \frac{\langle 13\rangle\langle 25\rangle\langle 34\rangle^2[15][24][34]^2}{\langle 23\rangle\langle 35\rangle^2\langle 45\rangle^3[23][35]}$$

$$- \frac{\langle 13\rangle\langle 25\rangle\langle 34\rangle^2[14][24][34][45]}{\langle 23\rangle\langle 35\rangle^3\langle 45\rangle^2[23][35]} + \frac{\langle 13\rangle\langle 25\rangle\langle 34\rangle[15][24][34]}{\langle 23\rangle\langle 35\rangle\langle 45\rangle^3[23]}$$

$$- \frac{\langle 13\rangle\langle 25\rangle\langle 34\rangle[15][24][34][45]}{\langle 23\rangle\langle 35\rangle^2\langle 45\rangle^2[23][35]} + \frac{\langle 13\rangle\langle 25\rangle[15][24][45]}{\langle 23\rangle\langle 35\rangle\langle 45\rangle^2[23]} - \frac{\langle 13\rangle\langle 34\rangle^2[14][34]^2[45]}{\langle 23\rangle\langle 35\rangle^2\langle 45\rangle^2[23][35]}$$

$$+ \frac{\langle 13\rangle\langle 34\rangle[15][34]^2}{\langle 23\rangle\langle 45\rangle^3[23]} + \frac{\langle 13\rangle[15][34][45]}{\langle 23\rangle\langle 45\rangle^2[23]} + \frac{\langle 13\rangle\langle 23\rangle\langle 25\rangle[24][25][34]}{\langle 15\rangle\langle 35\rangle^2\langle 45\rangle^2[35]}$$

$$- \frac{\langle 13\rangle\langle 23\rangle[25][34]}{\langle 15\rangle\langle 34\rangle\langle 45\rangle^2} + \frac{\langle 13\rangle\langle 25\rangle^2\langle 34\rangle[24]^2[25][34]}{\langle 15\rangle\langle 35\rangle^3\langle 45\rangle^2[23][35]} - \frac{\langle 13\rangle\langle 25\rangle\langle 34\rangle[24][34][45]}{\langle 15\rangle\langle 35\rangle^2\langle 45\rangle^2[35]}$$

$$+ \frac{\langle 13\rangle\langle 25\rangle\langle 34\rangle[24][25][34]^2}{\langle 15\rangle\langle 35\rangle^2\langle 45\rangle^2[23][35]} + \frac{\langle 13\rangle\langle 25\rangle[24][45]}{\langle 15\rangle\langle 35\rangle\langle 45\rangle^2} - \frac{\langle 13\rangle\langle 25\rangle[24][25][34]}{\langle 15\rangle\langle 35\rangle\langle 45\rangle^2[23]}$$

$$- \frac{\langle 13\rangle\langle 34\rangle[24][34][45]}{\langle 15\rangle\langle 35\rangle\langle 45\rangle^2[25]} + \frac{\langle 13\rangle[34][45]}{\langle 15\rangle\langle 45\rangle^2} - \frac{\langle 13\rangle[25][34]^2}{\langle 15\rangle\langle 45\rangle^2[23]}$$

$$+ \frac{\langle 13\rangle\langle 24\rangle\langle 25\rangle[24]^2[45]}{\langle 15\rangle\langle 23\rangle\langle 35\rangle\langle 45\rangle^2[23]} - \frac{\langle 13\rangle\langle 24\rangle\langle 34\rangle[24]^2[34][45]}{\langle 15\rangle\langle 23\rangle\langle 35\rangle\langle 45\rangle^2[23][25]} + \frac{\langle 13\rangle\langle 24\rangle[24][34][45]}{\langle 15\rangle\langle 23\rangle\langle 45\rangle^2[23]}$$

$$- \frac{\langle 13\rangle\langle 24\rangle\langle 34\rangle[24][34]^2[45]}{\langle 15\rangle\langle 23\rangle\langle 25\rangle\langle 45\rangle^2[23][25]} - \frac{\langle 13\rangle\langle 25\rangle^2\langle 34\rangle^2[24]^2[34][45]}{\langle 15\rangle\langle 23\rangle\langle 35\rangle^3\langle 45\rangle^2[23][35]} - \frac{\langle 13\rangle\langle 25\rangle\langle 34\rangle^2[24][34]^2[45]}{\langle 15\rangle\langle 23\rangle\langle 35\rangle^2\langle 45\rangle^2[23][35]}$$

$$+ \frac{\langle 13\rangle\langle 25\rangle\langle 34\rangle[24][34][45]}{\langle 15\rangle\langle 23\rangle\langle 35\rangle\langle 45\rangle^2[23]} + \frac{\langle 13\rangle\langle 34\rangle[34]^2[45]}{\langle 15\rangle\langle 23\rangle\langle 45\rangle^2[23]} - \frac{\langle 13\rangle\langle 15\rangle^2\langle 23\rangle\langle 24\rangle[14][15][24]}{\langle 14\rangle\langle 25\rangle\langle 35\rangle^2\langle 45\rangle^3[45]}$$

$$+ \frac{\langle 13\rangle\langle 15\rangle^2\langle 24\rangle\langle 34\rangle[13][14][24]}{\langle 14\rangle\langle 25\rangle\langle 35\rangle^2\langle 45\rangle^3[23]} + \frac{\langle 13\rangle\langle 15\rangle^2\langle 24\rangle\langle 34\rangle[13][15][24][45]}{\langle 14\rangle\langle 23\rangle\langle 25\rangle\langle 35\rangle\langle 45\rangle^3[23][25]}$$

$$- \frac{\langle 13\rangle\langle 15\rangle\langle 23\rangle\langle 24\rangle[15][24][34]}{\langle 14\rangle\langle 25\rangle\langle 35\rangle\langle 45\rangle^3[45]} + \frac{2\langle 13\rangle\langle 15\rangle\langle 23\rangle\langle 34\rangle[15][25][34]^2}{\langle 14\rangle\langle 35\rangle^2\langle 45\rangle^3[35][45]}$$

$$+ \frac{2\langle 13\rangle\langle 15\rangle\langle 23\rangle[15][25][34]}{\langle 14\rangle\langle 35\rangle^2\langle 45\rangle^2[35]} + \frac{\langle 13\rangle\langle 15\rangle\langle 23\rangle\langle 34\rangle[15][34]^2}{\langle 14\rangle\langle 25\rangle\langle 35\rangle\langle 45\rangle^3[45]} + \frac{\langle 13\rangle\langle 15\rangle\langle 23\rangle[15][34]}{\langle 14\rangle\langle 25\rangle\langle 35\rangle\langle 45\rangle^2}$$

$$- \frac{\langle 13\rangle\langle 15\rangle\langle 24\rangle^2[15][24]^2}{\langle 14\rangle\langle 25\rangle\langle 35\rangle\langle 45\rangle^3[25]} - \frac{\langle 13\rangle\langle 15\rangle\langle 24\rangle\langle 34\rangle[15][24]^2}{\langle 14\rangle\langle 35\rangle^2\langle 45\rangle^3[25]} - \frac{\langle 13\rangle\langle 15\rangle\langle 24\rangle\langle 34\rangle[13][24][45]}{\langle 14\rangle\langle 25\rangle\langle 35\rangle\langle 45\rangle^3[25]}$$

$$- \frac{\langle 13\rangle\langle 15\rangle\langle 24\rangle[15][24][45]}{\langle 14\rangle\langle 25\rangle\langle 35\rangle\langle 45\rangle^2[25]} - \frac{2\langle 13\rangle\langle 15\rangle\langle 34\rangle^2[15][34]^2}{\langle 14\rangle\langle 35\rangle^2\langle 45\rangle^3[35]} - \frac{\langle 13\rangle\langle 15\rangle\langle 34\rangle[15][24][45]}{\langle 14\rangle\langle 35\rangle^2\langle 45\rangle^2[25]}$$

$$- \frac{2\langle 13\rangle\langle 15\rangle\langle 34\rangle[15][34][45]}{\langle 14\rangle\langle 35\rangle^2\langle 45\rangle^2[35]} + \frac{\langle 13\rangle\langle 15\rangle\langle 24\rangle\langle 34\rangle[13][24][45]^2}{\langle 14\rangle\langle 23\rangle\langle 25\rangle\langle 35\rangle\langle 45\rangle^2[23][25]}$$

$$- \frac{\langle 15\rangle^2\langle 23\rangle\langle 24\rangle[14]^2[15][24]}{\langle 25\rangle\langle 35\rangle^2\langle 45\rangle^3[13][45]} - \frac{\langle 15\rangle^2\langle 23\rangle\langle 34\rangle[14]^2[25]^2[34]}{\langle 35\rangle^3\langle 45\rangle^3[23][35][45]} + \frac{\langle 15\rangle^2\langle 24\rangle\langle 34\rangle[14]^2[24]}{\langle 25\rangle\langle 35\rangle^2\langle 45\rangle^3[23]}$$

$$+ \frac{\langle 15 \rangle^2 \langle 34 \rangle^2 [14]^2 [25][34]}{\langle 35 \rangle^3 \langle 45 \rangle^3 [23][35]} - \frac{\langle 15 \rangle^2 \langle 34 \rangle [14][15][25][34]}{\langle 35 \rangle^2 \langle 45 \rangle^3 [23][35]} + \frac{\langle 15 \rangle^2 \langle 24 \rangle \langle 34 \rangle [14][15][24][45]}{\langle 23 \rangle \langle 25 \rangle \langle 35 \rangle \langle 45 \rangle^3 [23][25]}$$

$$+ \frac{\langle 15 \rangle^2 \langle 34 \rangle^2 [14][15][34][45]}{\langle 23 \rangle \langle 35 \rangle^2 \langle 45 \rangle^3 [23][35]} - \frac{2 \langle 15 \rangle \langle 23 \rangle \langle 24 \rangle [14][15][24][34]}{\langle 25 \rangle \langle 35 \rangle \langle 45 \rangle^3 [13][45]}$$

$$- \frac{\langle 15 \rangle \langle 23 \rangle \langle 25 \rangle \langle 34 \rangle [14][24][25]^2 [34]}{\langle 35 \rangle^3 \langle 45 \rangle^3 [23][35][45]} + \frac{\langle 15 \rangle \langle 23 \rangle \langle 25 \rangle [14][15][24][25]}{\langle 35 \rangle^2 \langle 45 \rangle^3 [13][45]}$$

$$+ \frac{\langle 15 \rangle \langle 23 \rangle \langle 34 \rangle [14][25][34]}{\langle 35 \rangle^2 \langle 45 \rangle^3 [35]} + \frac{\langle 15 \rangle \langle 23 \rangle [14][25]^2 [34]}{\langle 35 \rangle \langle 45 \rangle^3 [23][45]} - \frac{\langle 15 \rangle \langle 24 \rangle^2 [14][15][24]^2}{\langle 25 \rangle \langle 35 \rangle \langle 45 \rangle^3 [13][25]}$$

$$- \frac{\langle 15 \rangle \langle 24 \rangle \langle 34 \rangle [14][15][24]^2}{\langle 35 \rangle^2 \langle 45 \rangle^3 [13][25]} - \frac{\langle 15 \rangle \langle 24 \rangle [14][24][25]}{\langle 35 \rangle \langle 45 \rangle^3 [23]} - \frac{\langle 15 \rangle \langle 24 \rangle \langle 34 \rangle [14][24][45]}{\langle 25 \rangle \langle 35 \rangle \langle 45 \rangle^3 [25]}$$

$$+ \frac{\langle 15 \rangle \langle 24 \rangle \langle 34 \rangle [14][24][34]}{\langle 25 \rangle \langle 35 \rangle \langle 45 \rangle^3 [23]} - \frac{\langle 15 \rangle \langle 24 \rangle [14][15][24][45]}{\langle 25 \rangle \langle 35 \rangle \langle 45 \rangle^2 [13][25]} + \frac{\langle 15 \rangle \langle 25 \rangle \langle 34 \rangle^2 [14][24][25][34]}{\langle 35 \rangle^3 \langle 45 \rangle^3 [23][35]}$$

$$- \frac{\langle 15 \rangle \langle 25 \rangle \langle 34 \rangle [15][24][25][34]}{\langle 35 \rangle^2 \langle 45 \rangle^3 [23][35]} - \frac{\langle 15 \rangle \langle 34 \rangle^2 [14][34][45]}{\langle 35 \rangle^2 \langle 45 \rangle^3 [35]} - \frac{\langle 15 \rangle \langle 34 \rangle [14][25][34]}{\langle 35 \rangle \langle 45 \rangle^3 [23]}$$

$$- \frac{\langle 15 \rangle \langle 34 \rangle [14][25][34][45]}{\langle 35 \rangle^2 \langle 45 \rangle^2 [23][35]} - \frac{\langle 15 \rangle \langle 34 \rangle [14][15][24][45]}{\langle 35 \rangle^2 \langle 45 \rangle^2 [13][25]} + \frac{\langle 15 \rangle [15][25][34]}{\langle 45 \rangle^3 [23]}$$

$$- \frac{\langle 15 \rangle \langle 24 \rangle [15][24][45]}{\langle 23 \rangle \langle 45 \rangle^3 [23]} + \frac{\langle 15 \rangle \langle 24 \rangle \langle 34 \rangle [15][24][34][45]}{\langle 23 \rangle \langle 25 \rangle \langle 45 \rangle^3 [23][25]} + \frac{\langle 15 \rangle \langle 24 \rangle \langle 34 \rangle [14][24][45]^2}{\langle 23 \rangle \langle 25 \rangle \langle 35 \rangle \langle 45 \rangle^2 [23][25]}$$

$$+ \frac{\langle 15 \rangle \langle 25 \rangle \langle 34 \rangle^2 [15][24][34][45]}{\langle 23 \rangle \langle 35 \rangle^2 \langle 45 \rangle^3 [23][35]} + \frac{\langle 15 \rangle \langle 34 \rangle^2 [14][34][45]^2}{\langle 23 \rangle \langle 35 \rangle^2 \langle 45 \rangle^2 [23][35]} - \frac{\langle 15 \rangle \langle 34 \rangle [15][34][45]}{\langle 23 \rangle \langle 45 \rangle^3 [23]}$$

$$- \frac{\langle 23 \rangle^2 [25][34]^2}{\langle 25 \rangle \langle 45 \rangle^3 [45]} - \frac{\langle 23 \rangle^2 [25][34]}{\langle 25 \rangle \langle 34 \rangle \langle 45 \rangle^2} - \frac{\langle 23 \rangle \langle 24 \rangle [15][24][34]^2}{\langle 25 \rangle \langle 45 \rangle^3 [13][45]} + \frac{\langle 23 \rangle \langle 25 \rangle \langle 34 \rangle [24][25][34]}{\langle 35 \rangle^2 \langle 45 \rangle^3 [35]}$$

$$+ \frac{\langle 23 \rangle \langle 25 \rangle [15][24][25][34]}{\langle 35 \rangle \langle 45 \rangle^3 [13][45]} - \frac{\langle 23 \rangle [25][34]}{\langle 45 \rangle^3} + \frac{\langle 23 \rangle \langle 34 \rangle [34]^2}{\langle 25 \rangle \langle 45 \rangle^3} + \frac{\langle 23 \rangle [34][45]}{\langle 25 \rangle \langle 45 \rangle^2}$$

$$- \frac{\langle 24 \rangle^2 [15][24]^2 [34]}{\langle 25 \rangle \langle 45 \rangle^3 [13][25]} + \frac{2 \langle 24 \rangle \langle 25 \rangle [15][24]^2}{\langle 35 \rangle \langle 45 \rangle^3 [13]} - \frac{\langle 24 \rangle \langle 34 \rangle [15][24]^2 [34]}{\langle 35 \rangle \langle 45 \rangle^3 [13][25]} + \frac{\langle 24 \rangle [24][45]}{\langle 45 \rangle^3}$$

$$- \frac{\langle 24 \rangle \langle 34 \rangle [24][34][45]}{\langle 25 \rangle \langle 45 \rangle^3 [25]} - \frac{\langle 24 \rangle [15][24][34][45]}{\langle 25 \rangle \langle 45 \rangle^2 [13][25]} - \frac{\langle 25 \rangle \langle 34 \rangle^2 [24][34][45]}{\langle 35 \rangle^2 \langle 45 \rangle^3 [35]}$$

$$- \frac{\langle 25 \rangle \langle 34 \rangle [24][25][34][45]}{\langle 35 \rangle^2 \langle 45 \rangle^2 [23][35]} + \frac{2 \langle 25 \rangle [15][24][45]}{\langle 35 \rangle \langle 45 \rangle^2 [13]} + \frac{\langle 34 \rangle [34][45]}{\langle 45 \rangle^3} - \frac{\langle 34 \rangle [15][24][34][45]}{\langle 35 \rangle \langle 45 \rangle^2 [13][25]}$$

$$+ \frac{[25][34][45]}{\langle 45 \rangle^2 [23]} - \frac{\langle 24 \rangle [24][45]^2}{\langle 23 \rangle \langle 45 \rangle^2 [23]} + \frac{\langle 24 \rangle \langle 34 \rangle [24][34][45]^2}{\langle 23 \rangle \langle 25 \rangle \langle 45 \rangle^2 [23][25]} + \frac{\langle 25 \rangle \langle 34 \rangle^2 [24][34][45]^2}{\langle 23 \rangle \langle 35 \rangle^2 \langle 45 \rangle^2 [23][35]}$$

$$- \frac{\langle 34 \rangle [34][45]^2}{\langle 23 \rangle \langle 45 \rangle^2 [23]} + \frac{\langle 15 \rangle^2 \langle 23 \rangle \langle 24 \rangle \langle 34 \rangle [14][24]}{\langle 14 \rangle \langle 25 \rangle \langle 35 \rangle^2 \langle 45 \rangle^3} + \frac{\langle 15 \rangle \langle 23 \rangle^2 \langle 34 \rangle [25]^2 [34]^2}{\langle 14 \rangle \langle 35 \rangle^2 \langle 45 \rangle^3 [35][45]}$$

$$+ \frac{\langle 15 \rangle \langle 23 \rangle^2 [25]^2 [34]}{\langle 14 \rangle \langle 35 \rangle^2 \langle 45 \rangle^2 [35]} + \frac{\langle 15 \rangle \langle 23 \rangle^2 \langle 34 \rangle [25][34]^2}{\langle 14 \rangle \langle 25 \rangle \langle 35 \rangle \langle 45 \rangle^3 [45]} + \frac{\langle 15 \rangle \langle 23 \rangle^2 [25][34]}{\langle 14 \rangle \langle 25 \rangle \langle 35 \rangle \langle 45 \rangle^2}$$

$$+ \frac{\langle 15 \rangle \langle 23 \rangle \langle 24 \rangle \langle 34 \rangle [24][34]}{\langle 14 \rangle \langle 25 \rangle \langle 35 \rangle \langle 45 \rangle^3} - \frac{2 \langle 15 \rangle \langle 23 \rangle \langle 34 \rangle^2 [25][34]^2}{\langle 14 \rangle \langle 35 \rangle^2 \langle 45 \rangle^3 [35]} - \frac{2 \langle 15 \rangle \langle 23 \rangle \langle 34 \rangle [25][34][45]}{\langle 14 \rangle \langle 35 \rangle^2 \langle 45 \rangle^2 [35]}$$

$$- \frac{\langle 15 \rangle \langle 23 \rangle \langle 34 \rangle^2 [34]^2}{\langle 14 \rangle \langle 25 \rangle \langle 35 \rangle \langle 45 \rangle^3} - \frac{\langle 15 \rangle \langle 23 \rangle \langle 34 \rangle [34][45]}{\langle 14 \rangle \langle 25 \rangle \langle 35 \rangle \langle 45 \rangle^2} + \frac{\langle 15 \rangle \langle 24 \rangle^2 \langle 34 \rangle [24]^2 [45]}{\langle 14 \rangle \langle 25 \rangle \langle 35 \rangle \langle 45 \rangle^3 [25]}$$

$$+ \frac{\langle 15 \rangle \langle 24 \rangle \langle 34 \rangle [24][45]^2}{\langle 14 \rangle \langle 25 \rangle \langle 35 \rangle \langle 45 \rangle^2 [25]} + \frac{\langle 15 \rangle \langle 34 \rangle^3 [34]^2 [45]}{\langle 14 \rangle \langle 35 \rangle^2 \langle 45 \rangle^3 [35]} + \frac{\langle 15 \rangle \langle 34 \rangle^2 [34][45]^2}{\langle 14 \rangle \langle 35 \rangle^2 \langle 45 \rangle^2 [35]}$$

$$+ \frac{\langle 15 \rangle^2 \langle 23 \rangle \langle 24 \rangle \langle 34 \rangle [14]^2 [24]}{\langle 13 \rangle \langle 25 \rangle \langle 35 \rangle^2 \langle 45 \rangle^3 [13]} - \frac{\langle 15 \rangle \langle 23 \rangle \langle 24 \rangle [14][24][25]}{\langle 13 \rangle \langle 35 \rangle \langle 45 \rangle^3 [13]}$$

$$+ \frac{2 \langle 15 \rangle \langle 23 \rangle \langle 24 \rangle \langle 34 \rangle [14][24][34]}{\langle 13 \rangle \langle 25 \rangle \langle 35 \rangle \langle 45 \rangle^3 [13]} + \frac{\langle 15 \rangle \langle 24 \rangle^2 \langle 34 \rangle [14][24]^2 [45]}{\langle 13 \rangle \langle 25 \rangle \langle 35 \rangle \langle 45 \rangle^3 [13][25]}$$

$$+ \frac{\langle 15 \rangle \langle 24 \rangle \langle 34 \rangle [14][24][45]^2}{\langle 13 \rangle \langle 25 \rangle \langle 35 \rangle \langle 45 \rangle^2 [13][25]} - \frac{\langle 23 \rangle \langle 24 \rangle [24][25][34]}{\langle 13 \rangle \langle 45 \rangle^3 [13]} + \frac{\langle 23 \rangle \langle 24 \rangle \langle 34 \rangle [24][34]^2}{\langle 13 \rangle \langle 25 \rangle \langle 45 \rangle^3 [13]}$$

$$
\begin{aligned}
&- \frac{\langle24\rangle^2[24]^2[45]}{\langle13\rangle\langle45\rangle^3[13]} + \frac{\langle24\rangle^2\langle34\rangle[24]^2[34][45]}{\langle13\rangle\langle25\rangle\langle45\rangle^3[13][25]} - \frac{\langle24\rangle[24][45]^2}{\langle13\rangle\langle45\rangle^2[13]} \\
&+ \frac{\langle24\rangle\langle34\rangle[24][34][45]^2}{\langle13\rangle\langle25\rangle\langle45\rangle^2[13][25]} + \frac{\langle13\rangle^2\langle15\rangle\langle34\rangle[14][15]^2[34]^2}{\langle12\rangle\langle35\rangle^2\langle45\rangle^3[12][35][45]} + \frac{\langle13\rangle^2\langle15\rangle[14][15]^2[34]}{\langle12\rangle\langle35\rangle^2\langle45\rangle^2[12][35]} \\
&+ \frac{\langle13\rangle^2\langle25\rangle\langle34\rangle[15]^2[24][34]^2}{\langle12\rangle\langle35\rangle^2\langle45\rangle^3[12][35][45]} + \frac{\langle13\rangle^2\langle25\rangle[15]^2[24][34]}{\langle12\rangle\langle35\rangle^2\langle45\rangle^2[12][35]} - \frac{\langle13\rangle^2[15]^2[34]^2}{\langle12\rangle\langle45\rangle^3[12][45]} \\
&- \frac{\langle13\rangle^2[15]^2[34]}{\langle12\rangle\langle34\rangle\langle45\rangle^2[12]} + \frac{2\langle13\rangle\langle15\rangle\langle23\rangle\langle34\rangle[14][15][25][34]^2}{\langle12\rangle\langle35\rangle^2\langle45\rangle^3[12][35][45]} \\
&+ \frac{2\langle13\rangle\langle15\rangle\langle23\rangle[14][15][25][34]}{\langle12\rangle\langle35\rangle^2\langle45\rangle^2[12][35]} - \frac{2\langle13\rangle\langle15\rangle\langle34\rangle^2[14][15][34]^2}{\langle12\rangle\langle35\rangle^2\langle45\rangle^3[12][35]} \\
&- \frac{2\langle13\rangle\langle15\rangle\langle34\rangle[14][15][34][45]}{\langle12\rangle\langle35\rangle^2\langle45\rangle^2[12][35]} + \frac{2\langle13\rangle\langle23\rangle\langle25\rangle\langle34\rangle[15][24][25][34]^2}{\langle12\rangle\langle35\rangle^2\langle45\rangle^3[12][35][45]} \\
&+ \frac{2\langle13\rangle\langle23\rangle\langle25\rangle[15][24][25][34]}{\langle12\rangle\langle35\rangle^2\langle45\rangle^2[12][35]} - \frac{2\langle13\rangle\langle23\rangle[15][25][34]^2}{\langle12\rangle\langle45\rangle^3[12][45]} - \frac{2\langle13\rangle\langle23\rangle[15][25][34]}{\langle12\rangle\langle34\rangle\langle45\rangle^2[12]} \\
&- \frac{2\langle13\rangle\langle25\rangle\langle34\rangle^2[15][24][34]^2}{\langle12\rangle\langle35\rangle^2\langle45\rangle^3[12][35]} - \frac{2\langle13\rangle\langle25\rangle\langle34\rangle[15][24][34][45]}{\langle12\rangle\langle35\rangle^2\langle45\rangle^2[12][35]} + \frac{2\langle13\rangle\langle34\rangle[15][34]^2}{\langle12\rangle\langle45\rangle^3[12]} \\
&+ \frac{2\langle13\rangle[15][34][45]}{\langle12\rangle\langle45\rangle^2[12]} + \frac{\langle15\rangle\langle23\rangle^2\langle34\rangle[14][25]^2[34]^2}{\langle12\rangle\langle35\rangle^2\langle45\rangle^3[12][35][45]} + \frac{\langle15\rangle\langle23\rangle^2[14][25]^2[34]}{\langle12\rangle\langle35\rangle^2\langle45\rangle^2[12][35]} \\
&- \frac{2\langle15\rangle\langle23\rangle\langle34\rangle^2[14][25][34]^2}{\langle12\rangle\langle35\rangle^2\langle45\rangle^3[12][35]} - \frac{2\langle15\rangle\langle23\rangle\langle34\rangle[14][25][34][45]}{\langle12\rangle\langle35\rangle^2\langle45\rangle^2[12][35]} \\
&+ \frac{\langle15\rangle\langle34\rangle^3[14][34]^2[45]}{\langle12\rangle\langle35\rangle^2\langle45\rangle^3[12][35]} + \frac{\langle15\rangle\langle34\rangle^2[14][34][45]^2}{\langle12\rangle\langle35\rangle^2\langle45\rangle^2[12][35]} + \frac{\langle23\rangle^2\langle25\rangle\langle34\rangle[24][25]^2[34]^2}{\langle12\rangle\langle35\rangle^2\langle45\rangle^3[12][35][45]} \\
&+ \frac{\langle23\rangle^2\langle25\rangle[24][25]^2[34]}{\langle12\rangle\langle35\rangle^2\langle45\rangle^2[12][35]} - \frac{\langle23\rangle^2[25]^2[34]^2}{\langle12\rangle\langle45\rangle^3[12][45]} - \frac{\langle23\rangle^2[25]^2[34]}{\langle12\rangle\langle34\rangle\langle45\rangle^2[12]} \\
&- \frac{2\langle23\rangle\langle25\rangle\langle34\rangle^2[24][25][34]^2}{\langle12\rangle\langle35\rangle^2\langle45\rangle^3[12][35]} - \frac{2\langle23\rangle\langle25\rangle\langle34\rangle[24][25][34][45]}{\langle12\rangle\langle35\rangle^2\langle45\rangle^2[12][35]} + \frac{2\langle23\rangle\langle34\rangle[25][34]^2}{\langle12\rangle\langle45\rangle^3[12]} \\
&+ \frac{2\langle23\rangle[25][34][45]}{\langle12\rangle\langle45\rangle^2[12]} + \frac{\langle25\rangle\langle34\rangle^3[24][34]^2[45]}{\langle12\rangle\langle35\rangle^2\langle45\rangle^3[12][35]} + \frac{\langle25\rangle\langle34\rangle^2[24][34][45]^2}{\langle12\rangle\langle35\rangle^2\langle45\rangle^2[12][35]} \\
&- \frac{\langle34\rangle^2[34]^2[45]}{\langle12\rangle\langle45\rangle^3[12]} - \frac{\langle34\rangle[34][45]^2}{\langle12\rangle\langle45\rangle^2[12]}\,.
\end{aligned}
$$

Our model reduces this 298 term amplitude to Eq. (29):

$$
\overline{\mathcal{M}} = \frac{\langle12\rangle\langle13\rangle\langle23\rangle}{\langle24\rangle\langle25\rangle\langle45\rangle}\left(\frac{[14][35]}{\langle14\rangle\langle35\rangle} - \frac{[15][34]}{\langle15\rangle\langle34\rangle}\right).
$$

• Four scalars and one graviton: $\mathcal{M}(\phi\phi\phi\phi h^+)$

$$
\begin{aligned}
\mathcal{M} = {}&\frac{\langle12\rangle^3[12][25]}{\langle15\rangle^2\langle23\rangle\langle25\rangle\langle45\rangle[23][45]} + \frac{\langle12\rangle^3\langle34\rangle[12][25]^2[34]}{\langle14\rangle\langle15\rangle^2\langle23\rangle\langle35\rangle\langle45\rangle[14][23][35][45]} \\
&+ \frac{\langle12\rangle^3\langle34\rangle[12][25][34]}{\langle14\rangle\langle15\rangle^2\langle23\rangle\langle25\rangle\langle45\rangle[14][23][45]} + \frac{\langle12\rangle^2\langle13\rangle[12][35]}{\langle15\rangle^2\langle23\rangle\langle25\rangle\langle45\rangle[23][45]} \\
&+ \frac{\langle12\rangle^2\langle13\rangle\langle34\rangle[12][25][34]}{\langle14\rangle\langle15\rangle^2\langle23\rangle\langle35\rangle\langle45\rangle[14][23][45]} + \frac{\langle12\rangle^2\langle13\rangle\langle34\rangle[12][34][35]}{\langle14\rangle\langle15\rangle^2\langle23\rangle\langle25\rangle\langle45\rangle[14][23][45]} \\
&+ \frac{\langle12\rangle^2\langle34\rangle[12][25][34]}{\langle15\rangle^2\langle23\rangle\langle35\rangle\langle45\rangle[14][23][35]} + \frac{\langle12\rangle^2\langle34\rangle[25]^2[34]}{\langle15\rangle^2\langle23\rangle\langle35\rangle\langle45\rangle[23][35][45]} \\
&+ \frac{\langle12\rangle^2[25]^2}{\langle15\rangle^2\langle23\rangle\langle45\rangle[23][45]} + \frac{\langle12\rangle\langle13\rangle^2\langle24\rangle[13][24][25]}{\langle14\rangle\langle15\rangle^2\langle23\rangle\langle35\rangle\langle45\rangle[14][23][45]} \\
&+ \frac{\langle12\rangle\langle13\rangle^2\langle24\rangle[13][24][35]}{\langle14\rangle\langle15\rangle^2\langle23\rangle\langle25\rangle\langle45\rangle[14][23][45]} + \frac{\langle12\rangle\langle13\rangle\langle24\rangle[13][24]}{\langle15\rangle^2\langle23\rangle\langle25\rangle\langle45\rangle[14][23]}
\end{aligned}
$$

$$+ \frac{\langle12\rangle\langle13\rangle\langle34\rangle[12][34]}{\langle15\rangle^2\langle23\rangle\langle35\rangle\langle45\rangle[14][23]} + \frac{\langle12\rangle\langle13\rangle\langle34\rangle[25][34]}{\langle15\rangle^2\langle23\rangle\langle35\rangle\langle45\rangle[23][45]}$$

$$+ \frac{\langle12\rangle\langle13\rangle[25][35]}{\langle15\rangle^2\langle23\rangle\langle45\rangle[23][45]} + \frac{\langle12\rangle\langle13\rangle\langle24\rangle[24][25]}{\langle14\rangle\langle15\rangle^2\langle35\rangle\langle45\rangle[14][45]} + \frac{\langle12\rangle\langle13\rangle\langle24\rangle[24][35]}{\langle14\rangle\langle15\rangle^2\langle25\rangle\langle45\rangle[14][45]}$$

$$+ \frac{\langle12\rangle\langle14\rangle\langle34\rangle[25][34]}{\langle15\rangle^2\langle23\rangle\langle35\rangle\langle45\rangle[23][35]} + \frac{\langle12\rangle\langle24\rangle[24]}{\langle15\rangle^2\langle25\rangle\langle45\rangle[14]} + \frac{\langle13\rangle^3\langle24\rangle[13][24][35]}{\langle14\rangle\langle15\rangle^2\langle23\rangle\langle35\rangle\langle45\rangle[14][23][45]}$$

$$+ \frac{\langle13\rangle^3\langle24\rangle[13][24][35]^2}{\langle14\rangle\langle15\rangle^2\langle23\rangle\langle25\rangle\langle45\rangle[14][23][25][45]} + \frac{\langle13\rangle^2\langle24\rangle[13][24][35]}{\langle15\rangle^2\langle23\rangle\langle25\rangle\langle45\rangle[14][23][25]}$$

$$+ \frac{\langle13\rangle^2\langle24\rangle[24][35]}{\langle14\rangle\langle15\rangle^2\langle35\rangle\langle45\rangle[14][45]} + \frac{\langle13\rangle^2\langle24\rangle[24][35]^2}{\langle14\rangle\langle15\rangle^2\langle25\rangle\langle45\rangle[14][25][45]} + \frac{\langle13\rangle^2[35]}{\langle14\rangle\langle15\rangle^2\langle35\rangle[14]}$$

$$+ \frac{\langle13\rangle^2[35]^2}{\langle14\rangle\langle15\rangle^2\langle25\rangle[14][25]} + \frac{\langle13\rangle\langle14\rangle\langle34\rangle[34]}{\langle15\rangle^2\langle23\rangle\langle35\rangle\langle45\rangle[23]}$$

$$+ \frac{\langle13\rangle\langle24\rangle[24][35]}{\langle15\rangle^2\langle25\rangle\langle45\rangle[14][25]} + \frac{\langle13\rangle[35][45]}{\langle15\rangle^2\langle25\rangle[14][25]} .$$

Our model reduces this 29 term amplitude to Eq. (30):

$$\overline{\mathcal{M}} = \frac{[25][45]}{\langle15\rangle\langle35\rangle[14][23]} \left( 1 + \frac{\langle14\rangle\langle34\rangle[14]}{\langle23\rangle\langle45\rangle[25]} - \frac{\langle12\rangle\langle23\rangle[23]}{\langle14\rangle\langle25\rangle[45]} \right) .$$

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
