# Peer review of "Learning the Simplicity of Scattering Amplitudes"

_SciPost Physics, doi:SciPost Phys. 18, 040 (2025)_

## Round 1 · Referee Report · Anonymous (Referee 1) · 2024-10-9

Report

In the present article, the authors consider the use of machine-learning techniques to simplify tree-level scattering amplitudes written in terms of spinor-helicity variables, a task of relevance to analytic calculations in theoretical high-energy physics.

While machine-learning techniques have in the past years revolutionized many different fields related to numeric and noisy data, comparably little work has been done on applying machine-learning techniques to analytic and exact subjects, such as theoretical physics and mathematics. The present paper is a very interesting specimen of such work. As such, it indeed provides a novel link between different research areas and opens a new pathway in an existing research field, with clear potential for multi-pronged follow-up work.

In an introduction, the authors summaries the challenge and prospective reward behind simplifying complicated expression in theoretical physics, and in particular those written in terms on spinor-helicity variables.
They sketch their machine-learning approach to this challenge and give an overview of related machine-learning approaches in the literature.

In the second section, the authors introduce the spinor-helicity formalism and describe how they construct their training data by applying a number a scrambling steps to randomly generated simple expressions to produce expressions that should be simplified in a supervised-learning approach.

In the third section, the authors describe a transformer model that is able to simplify an expression if the expression is related to the simple expression by three or less scrambling steps. However, the authors find that the accuracy drops with the number of scrambling steps and does not generalize beyond the number of steps seen at training.

In a fourth section, the authors address the problem in generalization by training a second transformer to identify subsets of terms that are likely to be simplified by the first transformer. This allows the combined model to reliably simplify expressions of up to 50 terms. They also demonstrate that their model is able to simplify complicated expressions arising from tree-level scattering amplitudes.

The authors conclude with a summary of their results and perspective on future work. In particular, the authors argue that a similar machine-learning approach could be used for a range of different simplification problems.

The authors give further more technical details on their approach in seven appendices. They have made their code available via a github repository, further facilitating the reproduction of their work.
Moreover, the authors provide a link to an interactive online demonstration of their model, allowing interested readers to apply the model and gauge its strengths and limitations themselves.

The paper is well written and presents very interesting results. I thus recommend it for publication provided the following minor comments are addressed in a satisfactory fashion:

1. The authors mention their interactive online demonstration only in the abstract. It might improve the impact of the paper to refer to this interactive demonstration also in other places, as well as to elaborate on its capabilities.

2. One case for machine-learning in this context is that there exists no clear algorithmic way to simplify expressions in spinor-helicity variables analytically. The authors make this case in section 2.4, but this seems to be so fundamental to their work that it might already be mentioned in the introduction.

3. A second case for machine-learning in this context is that the simplification is hard to achieve but that its correctness is easy to verify via numerics. The authors mention this numeric verification on page 11. But again, this aspect seems to be so fundamental to their work that it would benefit from being mentioned already in the introduction. (It is widely known that transformers are prone to hallucinations. The possibility to numerically verify their output is the reason that hallucinations are not a problem for the authors' approach.)

4. One of the motivations that the authors bring up for their work is the simplicity of the Parke-Taylor amplitude (1.1). The authors mention that their model successfully simplifies the corresponding expressions for four and five gluons. I could not find a corresponding statement for six gluons though. In contrast, Parke and Taylor successfully simplified their earlier results for six gluons in 1986, using slightly different but related variables. (Parke and Taylor came up with an educated guess that they checked numerically, but that is also how the author's model works.) Could the authors place their impressive(!) achievements using a machine-learning approach into perspective of what is currently happening in the field of scattering amplitudes using a more traditional approach?

5. On page 11, the authors give many relevant technical details on the training of their model. Could they also mention how long training took on the A100 GPUs that they used?

6. From figure 3, it seems like five-point amplitudes are harder to learn for the model than four- and six-point amplitudes. Do the authors have an explanation as to why?

7. In figure 8, the authors give the averages of cosine similarities. Would it be useful to give also standard deviations?

8. Below (4.6), the authors write ``even as c(t) increases''. Since $0<c_0<1$ and $\alpha>0$, doesn't c(t) decrease with t?

9. While the authors consider massless amplitudes, many interesting processes involve also massive particles. Could the authors comment on whether it is possible to extend their approach to the massive case, for which a variant of spinor-helicity variables exists as well?

10. As previously written, the paper is in general very well written. I have spotted only two typos that the authors might wish to correct. On page 25, ``structure words'' should likely read ``structure of words''. On page 31, ``away [...] form completely dissimilar terms'' should likely read ``away [...] from completely dissimilar terms''.

Recommendation

Ask for minor revision

  • validity: -
  • significance: -
  • originality: -
  • clarity: -
  • formatting: -
  • grammar: -

Author:  Aurélien Dersy  on 2024-11-20  [id 4970]

(in reply to Report 1 on 2024-10-09)
Category:
remark
answer to question

We thank the referee for their time and valuable insight. Please find below a detail response to each of their comments:

1) We add information in footnote 1 to refer to the presence of an interactive demonstration and highlight the possibility of a simple local download via GitHub. The interactive demonstration linked in the abstract is hosted online via Streamlit, making it much slower to use in practice, so we prefer to direct the reader to the more stable version hosted on Github. This app is able to showcase the abilities of both the one-shot simplifier and the full iterative simplification pipeline.

2) Following the referee's recommendation, we emphasize in the introduction the fact that we do not have access to a canonical algorithm for performing a standard analytical simplification of spinor-helicity amplitudes.

3) Referees 2 and 3 both correctly point out that accesses to numerical evaluations are crucial to avoid model hallucinations. We highlight this point in the conclusion and also bring it up in the introduction.

4) The traditional simplification of spinor-helicity amplitudes does not have a canonical approach. One of the routes that is considered in practice to this day is simple guesswork. Akin to how Parke-Taylor obtained their simple formula for 6-pt gluon scattering, much simplifications still require some educated guess and a numerical check. Another approach is the explicit algebraic simplification through the packages referred to in [13,14]. For instance in SpinorHelicity4D one can make use of the SchoutenSimplify routine which uses Mathematica's internal FullSimplify routine. The simplification power there is tied to FullSimplify's performance and does not always guarantee returning the simplest possible form. The final route to traditional simplification is to build a smart Ansatz for the answer (satisfying little group and mass dimension scalings) and to fix it via numerical evaluation, for instance leveraging singular kinematic limits (see arXiv:2010.14525 or arXiv:1904.04067). This method yields drastic simplifications but will also not guarantee the "simplest" possible form (according to our own working defintion of what is "simple").

5) The training of the simplifier models on one A100 GPU took around 45h, 65h and 80h of running time for 4,5,6 points respectively. The contrastive models are much faster to train, taking around 2h each. We include some of these time metrics in the manuscript.

6) 4-pt amplitudes are easier to learn as they are typically much shorter in length. As we describe in Appendix B, the average expression that the networks will see contain much fewer numerator terms than at 5 or 6pts. The apparent difference between the 5pt and 6pt is to be taken carefully. Indeed, it seems like the models perform better on 6pts with 3 scrambles than on 5 pts with 3 scrambles. However, as we described in Appendix B, our restriction to amplitudes with less than 1k tokens in the training set is more restrictive for 6pt. Indeed, we discard 2% of all 5-pts generated against 8% of all 6-pts. This means that we do not include potentially harder amplitudes at 6-pt in Fig.3, which can explain the apparent gap in performance. We note that those amplitudes can be appropriately simplified in the full iterative setup since they are typically associated with either 3 scrambling moves or more than 2 numerator terms.

7) For completeness, we now report the standard deviations for the cosine similarities in Figure 8. Since one of the identities that we have considered (momentum squared) can be recovered by applying different successions of momentum conservation and Schouten we see an uncertainty band of $\pm 0.3$ around the average cosine similarities for more than 3 identities away (irrespective of whether we are considering 4,5 or 6-pt amplitudes), which we believe is attributed to this relationship. For a single identity away, in the masked comparison, we instead have a tighter spread of under $\pm 0.1$ around the cosine similarities displayed on the plot, while for the full comparison it is around $\pm 0.15$. We decide to comment on these points explicitly in Appendix F (and guide the reader to the Appendix in the main text) as they are not crucial to the main story - which is that we can correlate high cosine similarity values with terms that are $<3$ identities away.

8) Indeed, this is a typo, thank you for pointing it out !

9) In general, we expect that the overall framework that we have developed can be extended to other non-trivial cases. For instance, if we focus on the spinor-helicity formalism for massive particles (say in arXiv:1709.04891 or arXiv:2304.01589), then we can anticipate that we would require additional "words" (symbolic tokens) for describing the amplitudes. Changing an amplitude's representation in this way does not critically impact the entire model's architecture (it just requires adapting the initial embedding layer), but imposes the need for a dedicated data generation procedure. That is, when tackling a new task, one needs to create training pairs $( \overline{\mathcal{M}} ,\mathcal{M})$ by defining relevant identities (or scrambling steps) that can be applied to $\overline{\mathcal{M}}$ to scramble it. Provided these training pairs are available, the rest of the training pipeline we have presented can be applied without further modifications.

10) Thank you for spotting these typos !

---

## Round 1 · Referee Report · Giuseppe De Laurentis (Referee 3) · 2024-10-11

Report

The article presented by the authors provides a novel and compelling
approach to simplifying expressions in spinor-helicity variables,
addressing challenges posed by redundancies from momentum conservation
and Schouten identities. By leveraging machine learning (ML)
techniques, the authors provide a fresh perspective on this complex
problem. The presentation is clear and detailed. First, the authors
present a one-shot simplification technique for expressions of
moderate size. Then, they expand on this by investigating a sequential
simplification approach where sub-expressions are simplified after
being grouped together based on their cosine similarity. This allows
the simplification of larger expressions. Their open-source code
available on GitHub adds further value to their contribution.

In summary, I believe this article will be a valuable contribution to
both the machine learning and the scattering amplitude literature. I
appreciate the effort made by the authors to connect these fields
while making the article accessible to both communities. I therefore
recommend its publication.

Beforehand, I suggest the following minor revisions to enhance clarity
and impact:

1. Towards the end of page 2, in the introduction, the authors review
ML applications to high-energy physics. It may be worthwhile to
include in this discussion prior studies aimed at reproducing the
numerical output of the amplitudes, rather than achieving exact
analytic simplifications. For example, consider referencing
arXiv:2002.07516 and arXiv:2107.06625, (and perhaps check
references therein). Mentioning these works would strengthen the
connection to existing literature.

2. Towards the end of page 5, the authors cite references [40-42] in
the context of little-group and dimensional analysis
constraints. While these are indeed relevant, the main
simplification in those works arises from analysing
singularities. Perhaps the wording could be rephrased as "These
constraints, together with information from singular kinematic
limits, can [...]" to more accurately reflect that
work. Additionally, arXiv:2203.04269 is a recent advancement in
this approach, which can simplify spinor-helicity expressions in
the roundabout way (complex analytic -> numeric -> simpler
analytic).

3. On page 9, in section 2.4, projective coordinates and twistors are
mentioned, but without any reference. In the context of momentum
twistors arXiv:0905.1473 comes to mind, and additional references
could help guide readers unfamiliar with these topics.

4. On page 11, the authors mention that a numerical check is performed
on candidate expressions generated by the one-shot simplification
approach to verify their validity. Looking in the code at
add_ons/numerical_evaluations.py, it appears they are using
double-precision (16-digit) phase space points, requiring 9-digit
precision to declare a value as zero (ZERO_ERROR_POW_LOCAL = 9). It
might be beneficial stating this in the paper.

In principle, one may be concerned that numerical similar, but not
analytically identical, simplified expressions could be erroneously
accepted, or, on the contrary, that valid simplifications could be
discarded due to precision loss. While this is probably unlikely
until expressions have hundreds or thousands of terms, it might be
worth commenting upon. Higher-precision and/or finite-field
evaluations would greatly reduce room for errors, if needed.

The authors may also wish to consider a native python
implementation of spinor-helicity evaluations, rather than using a
Mathematica link to S@M; the python package "lips" could be an
alternative.

5. The particular redundancy of four-point amplitudes is referred to
on multiple occasions. A more mathematically sound statement is
that at four point, factorization is not unique (see "unique
factorization domain", arXiv:2203.17170). While at n-point, n>4,
factorization is (conjecturally) unique. This implies that there
exists a unique least common denominator (LCD) for n>4, but not
n=4.

This is evident in the first two amplitudes in appendix G, which
admit representations with different denominators. The first
Parke-Taylor formula is more commonly written as
(⟨1|2⟩^3)/(⟨1|4⟩⟨2|3⟩⟨3|4⟩), while the second expression could be
written as (⟨1|2⟩^6[3|4])/(⟨1|3⟩⟨1|4⟩⟨2|3⟩⟨2|4⟩⟨3|4⟩). The authors
could comment on how this choice is made: does the ML model return
multiple candidate representations, and is one picked at random
among the valid ones? Or perhaps, are the denominator factors in
the simplified expression restricted to be a subset of those in the
original expression?

Similarly, for n>4, the authors could comment on the ability of the
model to identify the least common denominator. For instance, in
the last amplitude before the bibliography, the denominators
contain manifestly spurious factors [25] and [45]. I imagine this
is an artefact of the compact form chosen to write this expression
in the paper, even if the ML algorithm may return an expression
without those factors in the denominator. It is worth noting that a
clear and efficient algorithmic way to determine the LCD exists,
through univariate interpolation and factor matching.

6. On a related note to 5, it has been widely observed that since
rational functions in scattering amplitudes are sparse,
multivariate partial fraction decompositions are an important tool
to tame their complexity. The authors could comment on whether this
already affects their ML approach or how it could be included.

7. While the authors consider up to six-point amplitudes, it appears
that only maximally helicity violating trees are considered. It
might be worthwhile to comment on what changes would be required to
handle NMHV trees, which may include three-particle irreducible
denominator factors s_ijk, and potentially spurious spinor chains.

Similarly, I would imagine that a more compelling application of
this method would be to loop-integral rational coefficients, rather
than tree amplitudes. Like for NMHV trees, these may include
several more denominator factors, other than 2-particle spinor
brackets.

8. In loop amplitudes, the numerical constants in the rational
coefficients can be rational numbers with fairly large denominators
and numerators. In their work, the authors encounter mostly simple
numerical coefficients (like ±1 or ±2), and by default choose to
blind all constants (see page 22). They could comment on how their
method could be reconciled with the numbers observed in loop
coefficients. Perhaps a similarity could be defined among the
constants, on top of that among the spinor monomials?

Recommendation

Publish (easily meets expectations and criteria for this Journal; among top 50%)

  • validity: -
  • significance: -
  • originality: -
  • clarity: -
  • formatting: -
  • grammar: -

Author:  Aurélien Dersy  on 2024-11-20  [id 4972]

(in reply to Report 3 by Giuseppe De Laurentis on 2024-10-11)

We thank the referee for their time and valuable insight. Please find below a detail response to each of their comments:

1) We include the references mentioned by the referee in the introduction, along with the related work of arXiv:1912.11055. We also include the more recent work of arXiv:2112.09145, which discusses the application of novel ML techniques for numerically evaluating scattering amplitudes.

2) We thank the referee for pointing this out and implement the suggested rewording. We also include the reference to the relevant arXiv:2203.04269.

3) We include arXiv:0905.1473 and arXiv:1408.2459 as references for the momentum twistor variables.

4) As pointed out by the referee, when doing a numerical comparison one could in principle have numerically similar but not analytically similar amplitudes. In our implementation this is mitigated by the fact that we numerically evaluate the amplitudes on two independent sets of phase space points (we add a comment on this point in the manuscript). We have tested the numerical checks on our validation and test sets and observed that all of the amplitudes were correctly matched following that procedure. Evidently, we could make the procedure even more robust by picking out additional independent phase space points. We also note that when using the iterative simplification algorithm the numerical checks are done at each simplification step (typically only involving expressions with roughly 20 numerator terms) so that the precision loss isn't as concerning. As mentioned by the referee however, when doing a check on a full amplitude with 100's or 1000's of terms this issue would become important and we would need additional checks - for instance using the lips package, as indicated by the referee, which we will look to implement in a future version of the code.

5) We thank the referee for pointing us to relevant literature and we adapt our discussion regarding uniqueness in footnote 4. When generating $N$ solutions with beam search or nucleus sampling, the model can indeed generate different valid solutions. We choose to return the valid solution with the smallest number of distinct numerator terms. When different solutions have the same number of numerator terms (such as $-\frac{\langle 1 2 \rangle [34]^2}{\langle 3 4 \rangle [14][23]}$ and $\frac{\langle 1 2 \rangle^3}{\langle 1 4 \rangle \langle 2 3 \rangle \langle 3 4 \rangle}$) we return the solution generated by the model which is associated with the best score, that is, the smallest average value of the log probability $-\log(p(x_i))$ for each token $x_i$. We do not manually restrict the denominator factors or implement additional checks on those. The amplitude of Eq 4.8 has indeed been compactly rewritten, where the raw output form is

$$ \frac{- \langle 1 2 \rangle \langle 2 3 \rangle^2 \langle 4 5 \rangle \left[ 2 3 \right] \left[ 2 5 \right] + \langle 1 4 \rangle^2 \langle 2 5 \rangle \langle 3 4 \rangle \left[ 1 4 \right] \left[ 4 5 \right] + \langle 1 4 \rangle \langle 2 3 \rangle \langle 2 5 \rangle \langle 4 5 \rangle \left[ 2 5 \right] \left[ 4 5 \right]}{\langle 1 4 \rangle \langle 1 5 \rangle \langle 2 3 \rangle \langle 2 5 \rangle \langle 3 5 \rangle \langle 4 5 \rangle \left[ 1 4 \right] \left[ 2 3 \right]} $$
, and the denominator is free of the spurious factors of $[2 5],[4 5]$ . In general, the answers given by the model follow the structure of the target data set of amplitudes (from Section 2.2) - though we have not done a comprehensive study to verify whether the LCD form is always returned.

6) In our current implementation we have chosen to always feed input amplitudes by first putting them under a common denominator (following the $\textit{cancel}()$ routine in $\textit{sympy}$) and training has only been done on such amplitudes. One could naturally have trained on amplitudes in a partial fraction form but we have not explored that data representation. We note that this choice of representation is mainly for convenience purposes and that indeed other alternatives could be considered - there is no difference from an architecture point of view. It would also be interesting to learn a similarity metric that looks directly at factors coming from a partial fraction decomposition, as opposed to numerator terms only. But, from a practical point of view (and due to our implementation choices), generating relevant training data was more straightforward when focusing only on numerator terms.

7) To go beyond our current setup and include other variables, $s_{ijk}, \langle 1| 2 + 5|1 ],\cdots$, we do not need to drastically modify the model's architecture. At this level, the only change that would need to be implemented is with respect to the size of the total word vocabulary (which in turn impacts the dimension of the initial embedding layer), that is, the total number of symbolic tokens required to represent an amplitude. However, one would need to generate appropriate training data using these new variables. In particular, one has to define a procedure for sampling over a set of relevant simple amplitudes using these variables and coding the desired shuffling moves. The rest of the pipeline (training and iterative simplification) can proceed almost identically. We add a sentence to specify this in the conclusion.

8) Referees 2 and 3 both raise an important question about the role of constants. Indeed, the application of our method to amplitudes with different numerical constants poses a challenge. In the footnote 14 we comment on our algorithmic choice for dealing with this complication, which involves blinding the constants and performing the simplification which reduces the norm of the coefficients as much as possible. In that way we are still able to reduce amplitudes with non trivial coefficients - though we have not performed a detailed analysis. Also, as the referee suggests, one could in principle add another check on top, for instance prioritizing terms whose coefficients are multiple of each other. Alternatively, one can also consider training on a dataset where non trivial coefficients are considered - though generalizing a transformer for dealing with arbitrary numbers is not easy (see the robustness issue highlighted in arXiv:2109.13986 when dealing with integers sparsely covered in the training set) and so we did not consider this path. In cases where distinct coefficients are informative, say coming from expanding $2([12] [34]+ [14][23])^4$ , one could also consider attempting to first recover the factored form (focusing only on high similarity terms for instance) and then simplifying the factored expression , blinding the overall power or numerical factors. We added a comment on these alternative approaches in the footnote 14, but leave their implementation for future work.

---

## Round 1 · Referee Report · Gregor Kälin (Referee 2) · 2024-10-11

Report

The manuscript presents a machine learning approach for the simplification of scattering amplitudes, more precisely for the algebraic simplification of rational functions in spinor helicity variables. Such a simplification is indeed a challenge and one that many scientists in this field have stumbled upon.

The authors start by discussing the type of input and output amplitudes/expressions and the generation of the training data set via a backward generation. Two approaches, a one-shot simplification and a sequential version, are then discussed in depth. Both setups -- including transformer models, beam searches, and contrastive learning -- consist of state-of-the-art ML methods. The network architectures are carefully chosen for the problem at hand and are reasonably motivated by the authors.

For both approaches the manuscript includes a plethora of results for the performance that show high efficacy and success rates for many applications, but also point to limitations of these methods. I agree with the authors' conclusion that their approach has the power and flexibility to tackle real problems in theoretical high-energy physics (and beyond!).

The article is written in a clear and concise language and understandable, in most parts, also for non-experts in machine learning. I highly appreciate the authors' efforts in that respect, as the target group -- mostly theoretical physicists -- might not be very familiar with many of ML concepts.

Furthermore, the methods described here are a beautiful showcase of the application of ML to obtain exact results. I believe that these and similar methods are also applicable to other areas of theoretical physics where one seeks exact analytical data.

Finally, the submission is accompanied by a git repository containing the code and data set used in this project, which allows for reproductability of the results and which is a valuable resource for the community.

I, therefore, recommend this article for publication after a few minor things have been answered/corrected in an updated version:
1 - In Sec 2.2: I would like to see some justification for the choice of the target data set and/or how well it will work for more complicated terms and/or how is there any bias by doing a backward generation compared to a forward generation (see e.g. https://arxiv.org/pdf/1912.01412). It seems biased to me in the sense that it might work great for amplitudes where we expect such very simple final forms, but I wouldn't expect it to capture well more complicated final expressions. I am thinking here of amplitudes with a "simplest" form of 10-20 numerator terms or even an application of these ideas to intermediate expressions where one might be interested to simplify very complicated rational functions to slightly less complicated rational functions. Or formulated from a different perspective: Is the target data set optimal for the sequential simplification or is there some potential for improvement?
2 - In the end of section 2.2 the authors mention that their setup is restricted to unitary relative coefficients of terms. This seems quite a bit restrictive for "real-world" applications (e.g. not N=4 SYM). I have seen amplitudes where the (probably) "simplest" form contains relative coefficients like 3/7. There seems to be a partial resolution to this restriction in footnote 14, but I'd suggest to extend that discussion as it wasn't clear to me how this additional difficulty could be handled most efficiently.
3 - Similar to the point above, the restriction to Nterms <= 3 in section 2.2 might not be optimal for more complicated problems. In summary, some more care should be taken to the choice of data sets and a discussion of their validity/bias/potential for improvement would be useful to the reader.
4 - How does the approach described in this paper compare to existing software, as e.g. cited in [13,14]?

Further ideas for improvement:
5 - Section 2.4 seems out of place and most of its content might better fit into the introduction.
6 - The same holds for the last few sentences of section 4.1. These comments might fit better in thes introductory part of section 4 or even in the introduction section.
7 - Figure 12: It might be useful to also somehow mark the distinction of circle and triangle markers within the plots.
8 - It would be interesting to see a comparison of the approach described in this paper with existing software, as e.g. cited in [13,14].
9 - The conclusions might benefit from an extended discussion related to future developments in this research area. In particular it might be worth emphasizing that these methods produce exact analytical results and may be applicable to other problems where fast numerical checks of the answers are available. I can easily think of a handful of applications that fulfill the latter criterion.

Recommendation

Publish (easily meets expectations and criteria for this Journal; among top 50%)

  • validity: top
  • significance: high
  • originality: high
  • clarity: top
  • formatting: excellent
  • grammar: excellent

Author:  Aurélien Dersy  on 2024-11-20  [id 4971]

(in reply to Report 2 by Gregor Kälin on 2024-10-11)

We thank the referee for their time and valuable insight. Please find below a detail response to each of their comments:

1) The referee raises a valid point about the potential bias introduced by the forwards vs backward generation. As pointed out in arXiv:1912.01412 (or the more recent arXiv:2410.08304), one can bias the model when the training data is generated by sampling from the target (backward generation) vs from the input (forward generation). In our implementation we have a backward generation since we start from the target amplitude and generate complex expressions from there. Any bias introduced will be on the form of the complicated amplitudes (for instance there could be complicated amplitudes that are not captured by our data generation procedure). However, our final test cases in Section 4.4 are on amplitudes coming from Feynman diagrams directly, so more closely matching real world applications and not directly generated as part of our training data. Since our models perform reasonably well on those we believe that our backward data generation procedure (although not perfect !) should generalize reasonably well to actual applications. Clearly, when one knows the form of the complicated amplitude (say if focusing on a particular theory/problem) it would be helpful to have training data generated in a forward fashion - starting from the complicated amplitudes and reducing them with some alternative software to get the simplified form. This comes at the cost of 1) having to sample over some distribution of complicated amplitudes that is to be determined and 2) the requirement of having some alternate software or tool to perform the simplification. We add comments to that effect in sections 2.2 and 2.3. Regarding the structure of the very simple forms we have considered in our training data (at most 3 terms), we don't believe this to be a strong limiting factor when doing an iterative simplification. For example, if we consider an input amplitude with 10 terms that simplifies to a "simpler" form with 9 terms, then the hope for our simplification algorithm is to identify different groups that can reduce independently. For instance, 3 terms of the original 10 terms amplitude might combine to give 2 terms via a Schouten identity. In that case our models would only predict the 3 $\rightarrow$ 2 simplification, which is captured by our training data (and overall we have 10 $\rightarrow$ 9 terms). However, a failure mode could arise if we need an intermediate increase in complexity to see a simplification (for example, from combining multiple identities to get 10 $\rightarrow$ 9 terms). We comment on this point further in the footnote 16 in the main text.

2) Referees 2 and 3 both raise an important question about the role of constants. Indeed, the application of our method to amplitudes with different numerical constants poses a challenge. In the footnote 14 we comment on our algorithmic choice for dealing with this complication, which involves blinding the constants and performing the simplification which reduces the norm of the coefficients as much as possible. In that way we are still able to reduce amplitudes with non trivial coefficients - though we have not performed a detailed analysis. Also, as the referee suggests, one could in principle add another check on top, for instance prioritizing terms whose coefficients are multiple of each other. Alternatively, one can also consider training on a dataset where non trivial coefficients are considered - though generalizing a transformer for dealing with arbitrary numbers is not easy (see the robustness issue highlighted in arXiv:2109.13986 when dealing with integers sparsely covered in the training set) and so we did not consider this path. In cases where distinct coefficients are informative, say coming from expanding $2([12] [34]+ [14][23])^4$ , one could also consider attempting to first recover the factored form (focusing only on high similarity terms for instance) and then simplifying the factored expression , blinding the overall power or numerical factors. We added a comment on these alternative approaches in the footnote 14, but leave their implementation for future work.

3) We refer to our response in point 1) regarding the restriction $N_{\text{terms}}<3$. In short, we expect our approach to be reasonable when intermediate simplification steps (a single identity application for instance) can have a simple target state, which is a subset of the entire amplitude.

4) The traditional simplification of spinor-helicity amplitudes does not have a canonical approach. One of the routes that is considered in practice to this day is simple guesswork. Akin to how Parke-Taylor obtained their simple formula for 6-pt gluon scattering, much simplifications still require some educated guess and a numerical check. Another approach is the explicit algebraic simplification through the packages referred to in [13,14]. For instance in SpinorHelicity4D one can make use of the SchoutenSimplify routine which uses Mathematica's internal FullSimplify routine. The simplification power there is tied to FullSimplify's performance and does not always guarantee returning the simplest possible form. The final route to traditional simplification is to build a smart Ansatz for the answer (satisfying little group and mass dimension scalings) and to fix it via numerical evaluation, for instance leveraging singular kinematic limits (see arXiv:2010.14525 or arXiv:1904.04067). This method yields drastic simplifications but will also not guarantee the "simplest" possible form (according to our own working defintion of what is "simple").

5) Following referee's 3 comment we decide to broadly state in the introduction the fact that we do not have access to a clear algorithmic way for performing a standard analytical simplification of spinor-helicity amplitudes. We delay the technical details to section 2.4 as they might obscure the reading of the introduction.

6) Following the referee's recommendation, we move the last few sentences discussing the applications of contrastive learning in physical contexts closer to the start of section 4.1.

7) We have updated the figure's caption to more clearly reflect the difference between the circle and triangle markers.

8) We refer to our answer for point 4)

9) We add a few sentences in the conclusion to emphasize the point raised by the referee, which is that model hallucinations can be systematically avoided in a mathematical context, where numerical evaluations and cross-checks are available.

---

## Round 2 · Referee Report · Anonymous (Referee 1) · 2024-11-25

Report

The authors have addressed the points raised in my original report in a satisfactory manner. I thus recommend publication as is.

Recommendation

Publish (easily meets expectations and criteria for this Journal; among top 50%)

---

## Round 2 · Referee Report · Gregor Kälin (Referee 2) · 2024-12-9

Report

I would like to thank the authors for the care taken in improving the draft according to all the referee's reports, and in clarifying a few points that I have raised.
Since all my requests have been satisfactorily answered/implemented I recommend this manuscript for publication in its current form.

Recommendation

Publish (easily meets expectations and criteria for this Journal; among top 50%)

---

## Round 2 · Author Response

We thank the referees for their thorough review and insightful comments. We answered their comments or questions in the author replies and have adapted the manuscript to reflect their recommendations.

---

## Round 2 · List of Changes

• Modifications to the introduction: after Eq 1.3 ("and, to our ... expressions analytically")", on the top of p.3 ("In the field of ... purely symbolic"), near the end of the first paragraph on p.3 ("For those problems... model hallucinations")
  • Modifying footnote 1
  • In the last paragraph before 2.2 adding "together with information from singular kinematic limits"
  • First paragraph of 2.2, adding "Following the ... backward generation."
  • Fixed typos in 2.8/2.9/2.10
  • Added a paragraph before 2.4 "One could be ... training set."
  • Added references 49/50 in 2.4
  • In second paragraph of 3.1 added "To reach 1500 ... amplitudes respectively" and "We ask for .... during training"
  • Modified footnote 4
  • Minimal updates to Table 2 and Fig 2,3,5,6,10,13 to account for a small correction to the 6-pt training data (6-pt simplifier model is retrained and evaluated)
  • Moved the last paragraph of 4.1 upwards to the second paragraph of 4.1 "including in high-energy physics ... reconstruction tasks."
  • Fourth paragraph of 4.2 adding "lasting around 2h"
  • Adding standard deviations in Fig.8 and updating caption
  • Last paragraph before 4.3 adding "We note that ... We comment on this point further in"
  • Correcting typo below 4.6 "c(t) decreases"
  • Modifying footnote 14
  • Minimal updates to Fig.9,14,15 after correcting a small bug in the contrastive model (no qualitative changes noticed)
  • In the conclusion adding "Crucially, ... numerical evaluations" and "or with factors .... systematically avoided".
  • Updating the legend and caption in Fig 12
  • Adding a paragraph in Appendix F "For instance, taking ... less than 3 identities"

---

## Editorial Decision

published